# Rare phenomena of central rhythm and pattern generation in a case of complete spinal cord injury

Karen Minassian[1], Aymeric Bayart[1], Peter Lackner[2,3], Heinrich Binder[2], Brigitta Freundl[2] & Ursula S. Hofstoetter [1] ✉

Lumbar central pattern generators (CPGs) control the basic rhythm and coordinate muscle activation underlying hindlimb locomotion in quadrupedal mammals. The existence and function of CPGs in humans have remained controversial. Here, we investigated a case of a male individual with complete thoracic spinal cord injury who presented with a rare form of self-sustained rhythmic spinal myoclonus in the legs and rhythmic activities induced by epidural electrical stimulation (EES). Analysis of muscle activation patterns suggested that the myoclonus tapped into spinal circuits that generate muscle spasms, rather than reflecting locomotor CPG activity as previously thought. The EES-induced patterns were fundamentally different in that they included flexor-extensor and left-right alternations, hallmarks of locomotor CPGs, and showed spontaneous errors in rhythmicity. These motor deletions, with preserved cycle frequency and period when rhythmic activity resumed, were previously reported only in animal studies and suggest a separation between rhythm generation and pattern formation. Spinal myoclonus and the EES-induced activity demonstrate that the human lumbar spinal cord contains distinct mechanisms for generating rhythmic multi-muscle patterns.

In the motor control system for hindlimb locomotion in quadrupedal mammals, specialized neural circuits in the lumbar spinal cord represent the final network[1,2]. In reduced animal preparations, the neural control exerted by these circuits was shown to be necessary and sufficient to generate the basic motor patterns underlying locomotion[3–5]. This central pattern generator (CPG) capability of the lumbar spinal circuits was directly demonstrated in various animal models by the generation of so-called fictive locomotion, i.e., rhythmic motor output recorded from ventral roots or peripheral nerves with flexor-extensor and left-right alternations in the absence of descending neural inputs or peripheral feedback cues[6]. Fictive locomotion was expressed, inter alia, in the immobilized (neuromuscular block by curarization), acute spinal cat pretreated with L-Dopa by electrical dorsal root or dorsal column stimulation[7], in the immobilized decerebrate cat by electrical stimulation of brainstem structures[8], or in the isolated neonatal mouse spinal cord through the administration of excitatory amino acids and serotonin[9]. Such explicit evidence is sparse in non-human primates. Attempts to elicit fictive locomotion by L-Dopa and dorsal column stimulation failed in acutely spinalized and immobilized macaque monkeys[10]. In adult marmoset monkeys, electrical stimulation of the brainstem in the decerebrate preparation or administration of different pharmacological agents (clonidine, NMDA, serotonin) following spinalization generated various rhythmic activities that presented "component fictive patterns" with alternating as well as synchronous bursts in flexor and extensor nerves[11]. If combined, the component fictive patterns obtained under different experimental conditions would resemble a true fictive locomotor pattern. Yet, full fictive locomotion was not evoked by any single condition.

In humans, the experimental procedures necessary for the demonstration of fictive locomotion cannot be applied. Indirect

[1]Center for Medical Physics and Biomedical Engineering, Medical University of Vienna, Vienna, Austria. [2]Neurological Center, Clinic Penzing, Vienna, Austria. [3]Department of Neurology, Clinic Floridsdorf, Vienna, Austria. ✉e-mail: ursula.hofstoetter@meduniwien.ac.at

evidence of spinally generated rhythmic activity comes from specific motor phenomena in individuals with spinal cord injury (SCI)[6]. First, spontaneously emerging rhythmic spinal myoclonus at a highly reproducible rate of 0.3–0.6 Hz and involving multiple lower-limb muscles was described in six individuals with chronic, clinically complete SCI[12–14]. The electromyographic patterns largely involved synchronous bursting across muscles. Despite the lack of a locomotor pattern, it was suggested that this type of self-sustained rhythmic activity was due to a partial release of a CPG[12–14]. A common denominator in the individuals expressing this type of spinal myoclonus was the presence of additional musculoskeletal pathologies below the level of the lesion, particularly of the hip[12–15]. A second line of indirect evidence comes from induced rhythmic activities in paralyzed leg muscles by tonic epidural electrical stimulation (EES) of the lumbar spinal cord[16,17]. While the most common electromyographic pattern was synchronous bursting, locomotor-like patterns with a clear reciprocal relationship between antagonistic muscle groups were detected as well[18].

Assuming that locomotor CPGs exist in the human spinal cord, fundamental characteristics of their operation observed in animal experiments should also be detected in humans. Notable motor phenomena during otherwise robust fictive locomotion are so-called motor deletions, reflecting spontaneous errors in CPG operation. Motor deletions are an absence of activity in a set of synergistic motor pools during a time period when it would normally occur[7,19]. This failure to provide rhythmic drive can be accompanied by a failure to inactivate the antagonistic set of motoneurons, resulting in their sustained firing[9].

Here, we describe rare motor phenomena all occurring in a male individual who had sustained a thoracic sensory and motor complete SCI (Supplementary Table 1) and a luxation of the left hip joint in a traffic accident. The subject presented with severe, medication-resistant muscle spasms and was referred to a clinical program for the treatment of lower-limb spasticity by EES[20]. The clinical assessment battery included a standardized neurological examination documented by poly-electromyographic recordings[21]. The same examination was conducted multiple times over a period of three months to assess the manifestations of the subject's spasticity and, after the implantation with an epidural lead, to assess the effects of stimulation. The data analyzed and reported here were taken from these examinations. We observed muscle spasms, self-sustained spinal myoclonus, EES-induced rhythmic activity, and spontaneous motor deletions, all in the same subject. Based on the ability to directly compare these motor phenomena, we propose that the generation of spinal myoclonus is closely linked to the circuits underlying muscle spasms. Following the logic from animal studies[9,11,22], the component locomotor patterns evoked by EES support the activation of the CPG, and the specific type of motor deletions detected here indicates a separation of rhythm generation and pattern formation in the human lumbar spinal cord with a flexor-dominant operation.

## Results
### Self-sustained rhythmic spinal myoclonus in the paralyzed lower limbs
We observed an intriguing pattern of self-sustained rhythmic electromyographic activity in lower-limb muscles within six neurological examinations (Fig. 1). The rhythmic activities either followed brief manipulations of the lower limbs by the examiner to assess the subject's spasticity, or occurred spontaneously between assessment segments while the subject was lying relaxed with the legs extended. This phenomenon clearly differed from ankle clonus, the only common self-sustained rhythmic muscle activity seen after chronic SCI (Supplementary Figs. 1 and 2). We identified eleven examples of such self-sustained rhythmic activities (Supplementary Table 2) that lasted for a minimum of 10 s (median duration: 18.1 s, interquartile range (IQR): 13.1–25.2 s). The longest episode had a duration of 63 s, only halted through repositioning of the lower limbs by the examiner.

The rhythmic activities involved muscles across hip, knee, and ankle joints and both legs (cf. Supplementary Table 2). Within a given example, rhythm-cycle frequencies did not differ between muscles (Supplementary Table 3). Across examples and muscles, the mean rhythm-cycle frequency amounted to $0.35 \pm 0.01$ Hz, ranging from $0.18 \pm 0.01$ Hz to $0.50 \pm 0.01$ Hz. No significant interaction effect on the rhythm-cycle frequency existed between example and muscle, $F(78;461) = 0.371$, $p = 1.000$, $\eta_p^2 = 0.059$. Muscle was not a significant factor for rhythm-cycle frequency, $F(9;461) = 0.987$, $p = 0.450$, $\eta_p^2 = 0.019$, but example was, $F(10;461) = 205.472$, $p < 0.001$, $\eta_p^2 = 0.817$.

Despite the examination days spanning a period of three months, the different geneses of the rhythmic activities, and the various rhythm-cycle frequencies, the sequence of muscle recruitment was highly robust. The electromyographic bursts appeared with a modest, yet consistent onset lag between muscles (Fig. 2a). Statistical analysis revealed no differences in the sequence of muscle recruitment between examples. No significant interaction existed between example and muscle, $F(65;1) = 23.743$, $p = 0.162$, $\eta_p^2 = 0.999$. Example was not a significant factor for the sequence of muscle recruitment, $F(9;1) = 95.185$, $p = 0.079$, $\eta_p^2 = 0.999$, but muscle was, $F(8;1) = 1112.986$, $p = 0.023$, $\eta_p^2 = 1.000$. Across examples, activity first occurred in the left lower leg muscles, corresponding to activity in the L4, L5 (tibialis anterior) and S1, S2 (triceps surae) spinal cord segments[23], followed by a spread to bilateral muscles and a complex migration of activity along the lumbar and upper sacral spinal cord (Fig. 2b).

We identified these observed phenomena as a very rare form of spinal myoclonus, documented in only six other individuals with complete SCI in the literature[12–14], based on the co-activation patterns, the range of rhythm-cycle frequencies, and the possible role of the subject's hip pathology[12,13,15] (Supplementary Fig. 3).

### Muscle activation patterns do not differ between spinal myoclonus and muscle spasms
In seven examples, spinal myoclonus followed lower-limb manipulations by the examiner that elicited either seconds-long proprioceptive activation (through a single cycle of passive hip-and-knee or ankle flexion-extension movement) or cutaneous activation from the foot sole (cf. Supplementary Table 2). No examples were found following a tendon tap or a brisk manual dorsiflexion to elicit an ankle clonus (cf. Supplementary Figs. 1 and 2), suggesting that spinal myoclonus was not readily evoked by brief proprioceptive volleys to the spinal cord.

The passive movements as well as plantar stimulation always elicited muscle spasms[24,25] (Supplementary Fig. 1) that had consistent electromyographic patterns when evoked repeatedly during the same neurological examination (Supplementary Fig. 4), irrespective of whether a myoclonus followed or not (Fig. 3a). The muscle spasms were characterized by a spread to multiple ipsilateral and contralateral muscles, resulting in complex muscle activation patterns outlasting the duration of the manipulation. Thus, the initially recruited spinal networks were always those underlying the generation of muscle spasms[26,27]. Spinal myoclonus evolved from these muscle spasms after a silent period or a period of tonic activity (Fig. 3a(i), b) and never involved muscles other than those already recruited into the spasms. Noticeably, the electromyographic patterns of the myoclonus bursts closely resembled that of the spasms, indicative of common pattern generating networks (Fig. 3c). Indeed, muscle activation patterns did not differ statistically between muscle spasms and spinal myoclonus bursts; no significant interaction effect existed between the type of activity and muscle on the respective integrated electromyographic activities, $F(9;120) = 1.097$, $p = 0.370$, $\eta_p^2 = 0.076$.

### The rhythmic nature of spinal myoclonus
The major difference between spinal myoclonus and muscle spasms, i.e., its rhythmic nature, compelled us to propose that spinal myoclonus bursts might be repeatedly triggered muscle spasms. The rhythmic

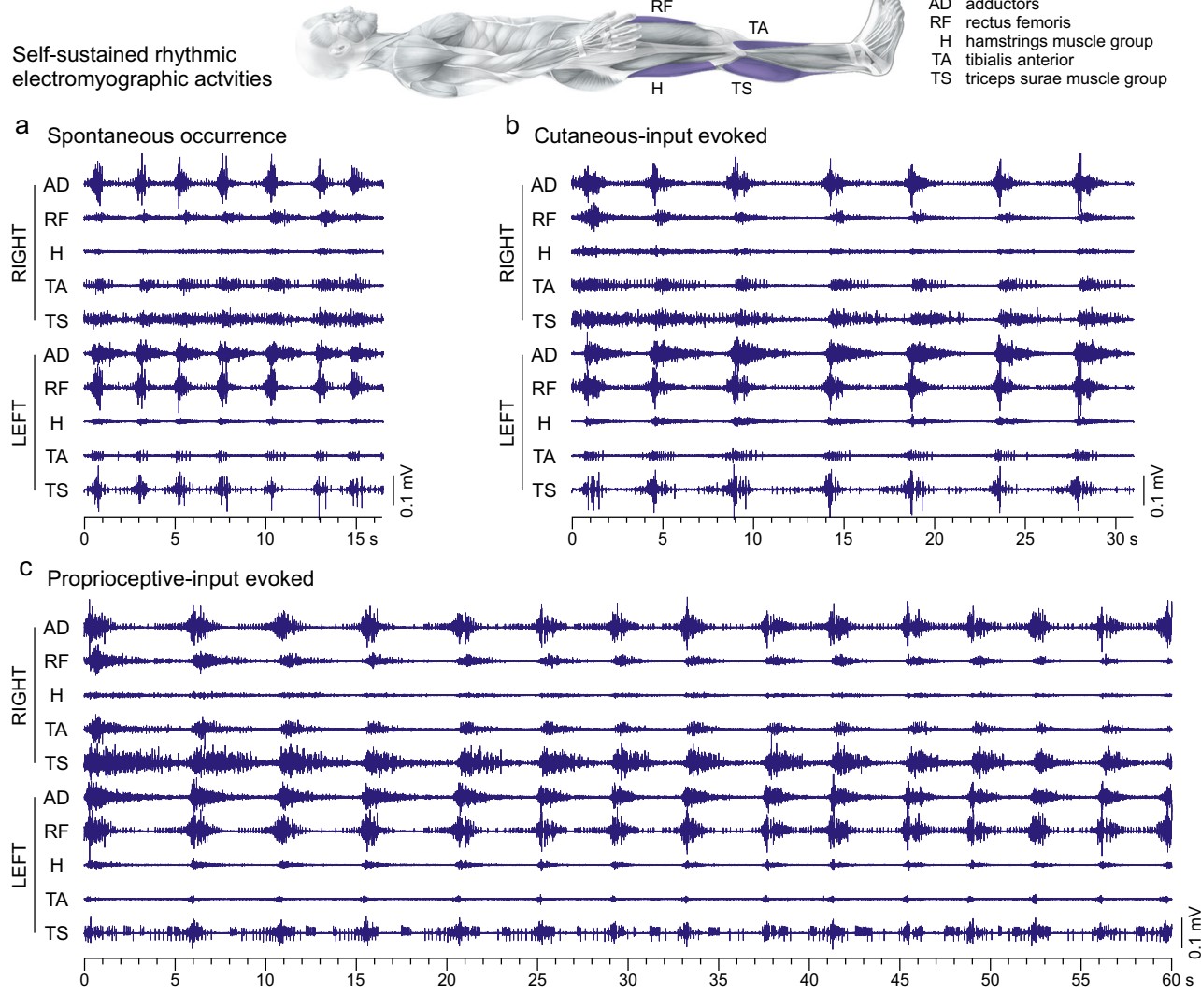

**Fig. 1 | Self-sustained rhythmic patterns of electromyographic activity in paralyzed lower limbs recorded in the supine position. a** Spontaneous rhythmic activity that occurred after a period of 3.9 min of continuous electromyographic (EMG) recordings during which no manipulations of the lower limbs were performed by the examiner, example 9 (cf. Supplementary Table 2). **b** Rhythmic activity that occurred immediately following cutaneous-input evoked spasms (not shown) by right plantar stimulation with a blunt rod, example 4. **c** Rhythmic activity that occurred immediately following spasms (not shown) evoked by one cycle of slow passive hip and knee flexion-extension movement of the right lower limb, example 2. The EMG activities of the rhythmic phenomena are shown for the total durations they had lasted with the same time and EMG amplitude scaling in (**a**), (**b**), and (**c**). Body image adapted from Ipsi- and Contralateral Oligo- and Polysynaptic Reflexes in Humans Revealed by Low-Frequency Epidural Electrical Stimulation of the Lumbar Spinal Cord, Hofstoetter US, Danner SM, Freundl B, Binder H, Lackner P, Minassian K, Brain Sciences, 11, 112, 2021[53], MDPI, © 2021 by the authors.

nature of spinal myoclonus would require a sustained excitatory drive as well as self-limiting mechanisms in spasm generation. The silent periods in-between consecutive bursts could have been caused by active inhibitory mechanisms curtailing an ongoing burst and delaying the onset of the succeeding burst[27] or could have been related to refractoriness or fatigue following each bursting event[28]. If the latter were true, the durations of silent periods should increase with the burst durations and magnitudes. Spontaneous cycle-to-cycle variations within the myoclonus episodes allowed us to study whether such relationships existed (Fig. 4a). Throughout the 94 cases of rhythmically active muscles of the eleven examples, the burst duration was not statistically correlated with the interburst duration in 89.4% (Fig. 4b(i)). The integral of the electromyographic activity of a burst was not correlated with the interburst duration in 87.3% (Fig. 4b(ii)).

The excitatory drive promoting the occurrence of spinal myoclonus as well as sustaining its rhythmic behavior could have been provided by the subject's hip pathology, as suggested previously[13-15]. Such facilitatory effect of a tonic excitatory drive was supported by

observations under ongoing EES with intensities subthreshold to evoke muscle activity (Fig. 5a). Here, each repetition of passive left hip-and-knee flexion-extension movements induced spasms that evolved into spinal myoclonus. Notably, tonic stimulation with incremental, yet subthreshold intensities did not change the muscle activation pattern of spinal myoclonus (Fig. 5b), but rather advanced the onset of the first burst[27] (Fig. 5c).

An interplay between an ongoing tonic excitatory drive from the hip, muscle spasm generation, and succeeding inhibition may present an alternative explanation for the rhythmic nature of spinal myoclonus, without requiring the involvement of a dedicated rhythm generating circuit of a locomotor CPG (Supplementary Fig. 5).

**Epidural electrical stimulation generates rhythmic activity distinct from spinal myoclonus**

EES at 30–90 Hz and with intensities above the threshold to evoke muscle responses initiated and maintained rhythmic electromyographic activities in the paralyzed lower limbs[16,17]. We found six

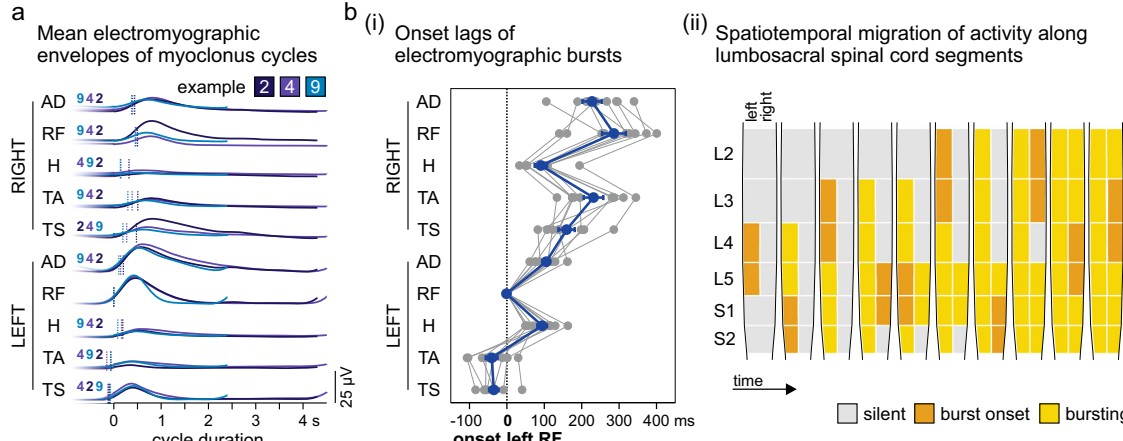

**Fig. 2 | The self-sustained rhythmic activities exhibit highly robust spatio-temporal patterns of electromyographic activity in the lower limbs.** The consistent occurrence of rhythmic activities in the left rectus femoris (RF) in ten of the eleven examples (cf. Supplementary Table 2) allowed the analysis of the sequence of muscle recruitment through the relative onset lags of electromyographic (EMG) bursts. **a** Envelopes of the EMG activities per muscle averaged across the available rhythm cycles of examples 2, 4, and 9 (cf. Fig. 1, Supplementary Table 2), and aligned with respect to the onset of the EMG burst in the left RF ('0' on the x-axis).

The EMG burst onsets are marked by vertical lines with the sequence given by the example numbers. **b** (i) Relative onset lags of rhythmic bursts occurring in the various muscles. Gray lines, individual results of the ten examples; blue line, mean values (±SE) across examples. (ii) Time course of progressive spread of the self-sustained rhythmic activity along the lumbosacral spinal cord segments based on the segmental innervations of the recruited muscles[23]. AD adductors, H hamstrings muscle group, L lumbar, S sacral, TA tibialis anterior, TS triceps surae muscle group. Source data are provided in the Source Data file.

examples of EES-induced rhythmic multi-muscle activation patterns that lasted for a minimum of 10 s with unchanged stimulation parameters (Fig. 6, Supplementary Fig. 6).

In great contrast to the pattern of spinal myoclonus that was consistent within an examination session and over a period of three months, the different examples of rhythmic activities evoked by EES displayed different patterns. One pattern was a fast rhythmic activity (rhythm cycle frequencies of 1.01 Hz in Fig. 6a(i) and 0.78 Hz and 0.92 Hz in the examples in Supplementary Figs. 6a(i) and (ii)) with a single phase of near-synchronized output across multiple muscles and a strong activation of tibialis anterior, interrupted by brief silent phases (Fig. 6a(i)). We also found, at lower cycle frequencies (0.45 Hz in Fig. 6a(ii) and 0.45 Hz and 0.58 Hz in Supplementary Figs. 6b(i) and (ii)), more complex patterns reflecting key elements of a locomotor CPG output (Fig. 6a(ii), b(ii)), i.e., right-left alternations between homologous muscles (cf. RTS and LTS) as well as reciprocal activation of extensors and flexors acting on the same joint (cf. RTA and RTS)[6]. These locomotor-like electromyographic activities resembled the component fictive patterns found in marmoset monkeys in a remarkable way (Supplementary Fig. 7). As in the marmoset monkey, the full locomotor pattern was not observed.

Rhythmic electromyographic activities with different patterns collected under the same recording conditions suggested that the fast rhythmic synchronous bursting reflected the recurrent recruitment of a flexor-dominant synergy (Figs. 6c(i), 7a(i) and b(i)). This flexion-like phase persisted within the more complex rhythmic patterns, where it alternated with an extension-like phase characterized by triceps surae activity (Figs. 6c(ii), 7a(ii) and b(ii)). The recruitment pattern in each of these two alternating phases was highly robust from cycle to cycle (Fig. 7b). The two phases were seen across different recording sessions (cf. Fig. 6). While flexion-like phases could exist without the requirement of extension-like phases in the synchronous bursting patterns (Figs. 6a(i), 7a(i)), the extension-like phases only occurred in alternation with the flexion-like phases as part of the locomotor-like patterns (Figs. 6a(ii), 7a(ii)).

The extension- as well as the flexion-like phases of the epidurally induced rhythmic activities differed from the multi-muscle patterns of spinal myoclonus bursts both in terms of the onset lags of the

electromyographic bursts between muscles as well as of the muscle activation patterns (Supplementary Fig. 8).

Notably, the rhythm-cycle frequencies of the synchronous bursting patterns were significantly higher than those of the locomotor-like patterns across the EES examples, indicating that changes in the speed of rhythmic activity additionally involved changes in the coordination of muscle activity (Supplementary Fig. 9). Both EES-induced types of rhythmic activities had faster rhythm-cycle frequencies than spinal myoclonus, with only a single EES-induced example overlapping with the spinal myoclonus range (Supplementary Fig. 9).

A striking phenomenon within the electromyographic traces during ongoing EES is captured in Fig. 7a(ii). In the following, we will present data to substantiate that this phenomenon was in fact a motor deletion in analogy to those described as spontaneous errors in the CPG operation in reduced animal models[9,19,29].

## Spontaneous errors in epidural electrical stimulation-induced rhythmic activities are flexor deletions

We identified 12 examples of electromyographic activities generated by EES that presented with transient phases of an absent tibialis anterior burst accompanied by sustained activity in the other ipsilateral muscles during otherwise robust rhythmicity (recordings EES 3, EES 4; Figs. 7a(ii), 8a). In six examples, EES induced activity in the right lower limb only, and in the remaining cases, muscles of the left lower limb were additionally recruited. In all 12 examples, the right rectus femoris displayed a double-bursting pattern during alternating flexion- and extension-like phases of the regular rhythmicity and tonic activity during the interrupted rhythmicity and was hence used to distinguish these events (Fig. 8a). We further divided the phases of tonic activity in the right rectus femoris into six divisions of equal duration and calculated the integrated electromyographic activity separately for the first division ($Del_1$) and the later ones, and averaged the values of the final five divisions ($Del_{2-6}$). We found a significant phase (flexion-like, extension-like, $Del_1$, $Del_{2-6}$) × muscle interaction effect on the integrated electromyographic activity, $F(12;810) = 23.852$, $p < 0.001$, $\eta_p^2 = 0.259$ (Fig. 8b). Both, phase, $F(3;810) = 83.872$, $p < 0.001$, $\eta_p^2 = 0.237$, and muscle, $F(4;810) = 29.959$, $p < 0.001$, $\eta_p^2 = 0.129$, were significant factors. Post-hoc Bonferroni-corrected pairwise comparisons revealed significant differences between the muscle activation

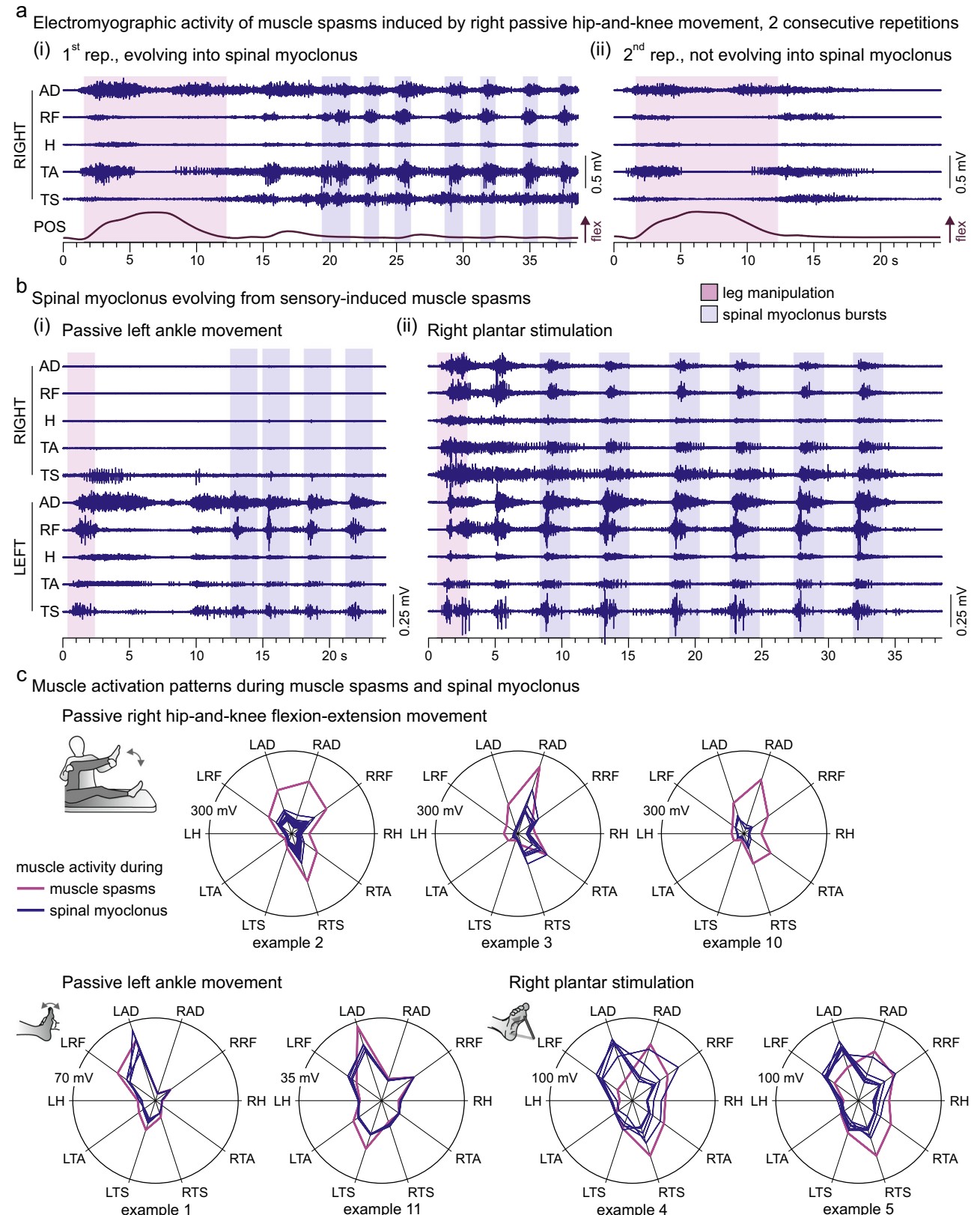

**a** Electromyographic activity of muscle spasms induced by right passive hip-and-knee movement, 2 consecutive repetitions

(i) 1$^{st}$ rep., evolving into spinal myoclonus

(ii) 2$^{nd}$ rep., not evolving into spinal myoclonus

**b** Spinal myoclonus evolving from sensory-induced muscle spasms

(i) Passive left ankle movement

(ii) Right plantar stimulation

leg manipulation
spinal myoclonus bursts

**c** Muscle activation patterns during muscle spasms and spinal myoclonus

Passive right hip-and-knee flexion-extension movement

example 2   example 3   example 10

muscle activity during
— muscle spasms
— spinal myoclonus

Passive left ankle movement

example 1   example 11

Right plantar stimulation

example 4   example 5

patterns of the flexion-like phases and all other phases, all $p < 0.001$, as well as between the extension-like phases and Del$_{2-6}$, $p = 0.012$. Notably, the extension-like phases did not differ from Del$_1$, indicating that the phases of interrupted rhythmicity started as extension-like phases. Results are further detailed in Supplementary Table 4. In all 12 examples, increasing tibialis anterior activity coincided with the termination

of the tonic activity in the other muscles, and the resumed rhythmicity always started with a flexion-like phase. We propose that the phases of interrupted rhythmicity were motor deletions and that in fact, all 12 cases found here were flexor deletions[9] (Supplementary Fig. 10).

The flexor deletions did not change the cycle frequency when rhythmic activity resumed. Mean durations of the pre-deletion cycles

**Fig. 3 | Spinal myoclonus patterns resemble repeatedly triggered muscle spasms. a** Muscle spasms evoked by imposed flexion-extension movement of the right lower limb (POS, knee position). The electromyographic (EMG) patterns of these muscle spasms (magenta backgrounds in (i) and (ii)) were consistent when the manipulation was repeated (see also Supplementary Fig. 4). **b** Spinal myoclonus evolved from the muscle spasms in response to the manipulations by the examiner. The multi-muscle EMG patterns of the myoclonus bursts (blue backgrounds) show resemblance to those of the immediately evoked spasms (magenta background); (i), example 1 (cf. Supplementary Table 2); (ii), example 5. **c** Polar plots show muscle activation patterns of spasms (magenta lines) and myoclonus bursts (blue lines) of all seven examples of spinal myoclonus that followed lower-limb manipulations by the examiner. Radial axes are muscles and polar coordinates are mean integrated EMG activities. Muscle activation patterns of muscle spasms and spinal myoclonus were not statistically different in any of the available examples. AD adductors, H hamstrings muscle group, L left; RF rectus femoris, R right, rep. repetition, TA tibialis anterior, TS triceps surae muscle group. Source data are provided in the Source Data file.

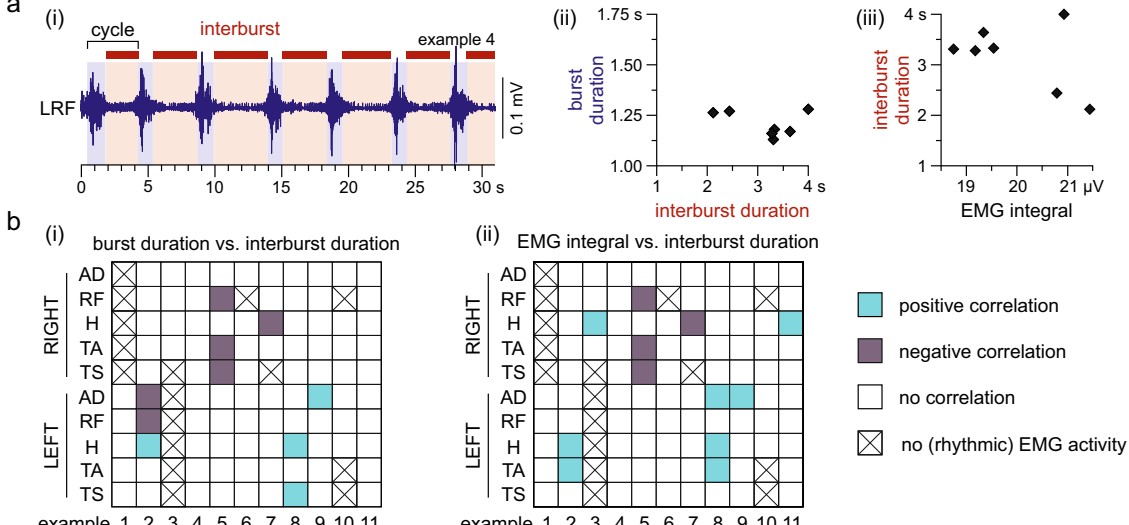

**Fig. 4 | The repetitive nature of spinal myoclonus is not due to neural circuit refractoriness. a** (i) Exemplary recording of rhythmic electromyographic (EMG) activity of the left (L) rectus femoris (RF) indicates cycle-to-cycle variations in interburst durations (red backgrounds and horizontal bars) and burst durations (blue backgrounds), example 4 (cf. Fig. 1b, Supplementary Table 2). In this example, there was no correlation between (ii) the burst durations and the interburst durations, Pearson's $r = -0.091$, $p = 0.864$; and (iii) the integrated EMG activity of the burst and interburst durations, $r = -0.092$, $p = 0.863$. **b** The relationships between (i) burst durations and interburst durations and (ii) integrated EMG activity of bursts and interburst durations were analyzed by Pearson's product moment correlations, separately for the 94 available cases of rhythmically active muscles of the eleven spinal myoclonus examples (cf. Supplementary Table 2). White squares indicate no significant correlations, blue squares significant positive correlations, and gray squares significant negative correlations (all $p < 0.05$); crosses are non-rhythmically active or inactive muscles (not included in the analyses). AD adductors, H hamstrings muscle group, TA tibialis anterior, TS triceps surae muscle group. All statistical tests were two-sided.

were 1.30 ± 0.09 s and of the post-deletion cycles 1.20 ± 0.09 s, with no statistical difference found, paired Student's $t$ test, $t(11) = -1.317$, $p = 0.215$, $r = 0.380$. Finally, we investigated whether the deletions were non-resetting, i.e., whether the rhythmic activity re-emerged after an integer number of missing cycles[9,19]. Figure 8c shows that the majority of the deletions had durations that lined up noticeably well with twice the duration of the respective mean rhythm cycles. Using statistical analysis for the classification of motor deletions in mice[9] and cats[19], we found that four of the 12 examples were indeed non-resetting.

In the six available examples of bilateral lower-limb muscles recruited by EES, weaker activity was generated on the left side, with less robust rhythmicity. The bursting across contralateral muscles was highly synchronized with the right tibialis anterior activity (Supplementary Fig. 11). During the unilateral flexor deletions, the activity across contralateral muscles was absent or reduced (Supplementary Fig. 12).

**There is no interdependence between the occurrence of spinal myoclonus and epidural electrical stimulation-induced rhythmic activity**

To investigate whether there was a link between the occurrence of spinal myoclonus and the rhythmic activity generated by EES, we re-examined data from a previous study[18] that included ten participants with chronic motor-complete SCI who had been implanted with epidural electrodes and assessed neurologically using the same protocols

as applied here[21]. Rhythmic lower-limb activity had been induced by EES in seven of these previous study participants, while none had demonstrated spinal myoclonus (Supplementary Fig. 13). This observation provides further support for the idea that the two types of rhythmic activity are indpendent motor phenomena.

## Discussion

Spinal myoclonus and EES-induced rhythmic patterns of electromyographic activity in the lower limbs in individuals with clinically complete SCI have been regarded as independent lines of indirect evidence for the existence of a CPG in humans[6,13,14,16,17,30]. Here, the observation of both types of rhythmic motor activity in a single subject allowed their direct comparison and revealed important differences in their rhythm and patterns. The rhythm-cycle frequencies of spinal myoclonus observed here were concordant with previous reports, but lower than those of the EES examples. The observation of motor deletions during otherwise robust rhythmic activity occurred only in the EES examples, but not in spinal myoclonus. Spinal myoclonus presented with consistent multi-muscle activation patterns. The patterns of EES-induced rhythmic activity yielded variability and differed from spinal myoclonus. These findings compelled us to suggest that spinal myoclonus engages a subset of pattern-formation circuits, while EES had activated both intrinsically rhythm-generating circuits as well as essential parts of the pattern-formation circuits of the locomotor CPG.

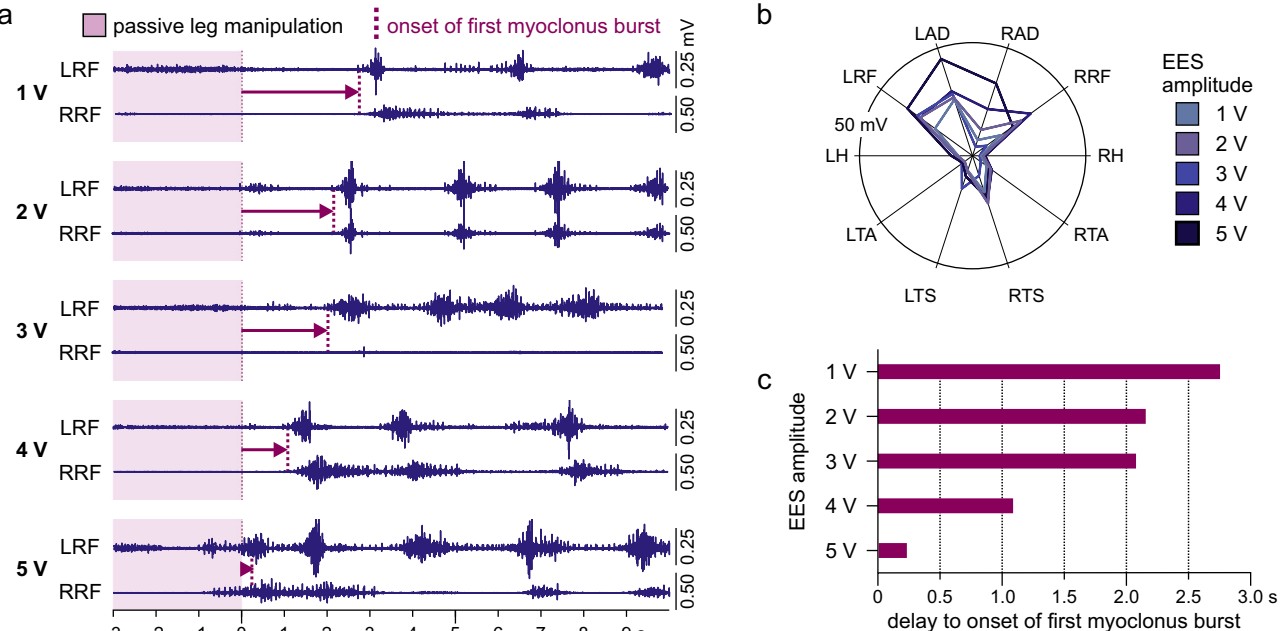

**Fig. 5 | Subthreshold excitatory drive provided by epidural stimulation promotes the onset of sensory-evoked spinal myoclonus. a** Myoclonus patterns evolving in the left (L) and right (R) rectus femoris (RF) from a single cycle of imposed flexion-extension movements of the left lower limb (magenta background showing the final extension phase) under ongoing subthreshold epidural electrical stimulation (EES). Arrows mark the times between the end of the passive leg movement and the onset of the first burst of rhythmicity (dashed lines, identified by a thresholding technique, see Methods). Active electrode contacts: 1+2−, stimulation frequency: 38.5 Hz, stimulation amplitudes: 1–5 V as indicated, pulse width: 210 μs. **b** Polar plots show similar muscle activation patterns of spinal myoclonus bursts recorded during ongoing EES with incremental, yet subthreshold amplitudes. Radial axes are muscles and polar coordinates are mean integrated electromyographic activities. **c** Bars highlight the decrease in delay between the cessation of passive leg manipulation and the onset of spinal myoclonus with incremental EES amplitudes. AD adductors, H hamstrings muscle group, TA tibialis anterior, TS triceps surae muscle group. Source data are provided in the Source Data file.

The spinal myoclonus observed here closely resembled a specific form described in a very limited number of individuals with chronic, clinically complete SCI[12–14] (Supplementary Figs. 1 and 3). This form appeared as synchronous discharges in multiple lower-limb muscles bilaterally at rhythm-cycle frequencies of 0.3–0.6 Hz[12,13], with out-of-phase bursting being an exception[14]. The involuntary rhythmic contractions were smooth[13] and did not resemble the brief, rapid jerks ascribed to other forms of spinal myoclonus[31,32]. This Bussel-Calancie type of spinal myoclonus was proposed to be due to a partial release of a spinal locomotor CPG, based on its self-sustained nature and stereotyped patterns, its sensitivity to hip-related proprioceptive inputs[13,14], cf.[33,34], and its interaction with flexor reflexes[12], cf.[30,35]. Yet, "nothing about the limited leg movements caused by these contraction patterns resembled stepping movements of the legs, except for the highly reproducible period and rate of contractions"[13].

Here, the intriguing similarity of the multi-muscle activation patterns of the myoclonus bursts with those of muscle spasms led us to propose an alternative explanation. The electromyographic patterns of muscle spasms always spanned multiple lower-limb muscles. The sensory-evoked spinal myoclonus always evolved from spasms and never recruited additional muscles. The spinal myoclonus bursts resembled the electromyographic activity of spasms, although the latter generally appeared with a longer duration (cf. Fig. 3, Supplementary Fig. 4). Statistically, the multi-muscle electromyographic patterns did not differ between myoclonus bursts and muscle spasms. A close link between these motor phenomena can be also deduced from previous studies. High levels of spasticity were a common factor across the individuals who had developed the Bussel-Calancie type of spinal myoclonus[12,13]. It was also previously noticed that spinal myoclonus could evolve from strong muscle spasms induced by hip extension when the study participants moved from sitting to lying position[13]. Muscle spasms are one of the manifestations of spasticity in chronic SCI[36]. They are involuntary, prolonged muscle contractions that can outlast the sensory inputs that had triggered them by many seconds, but do not have a recurring, rhythmic behavior[24,37] (Supplementary Figs. 1 and 13). Spasms following a SCI have traditionally been attributed to decreased inhibition of sensory transmission and increased expression of plateau potentials in motoneurons[38,39]. However, a contribution of spinal interneuronal circuits has long been hypothesized, in part because stretch at one joint induces stereotyped spasms in multiple non-stretched muscles[24,25,40] (Supplementary Fig. 4b). An enhancement in excitability of interneurons within the multi-segmental, spasm-generating circuits is likely involved[36]. Some studies have suggested that the neural substrate of muscle spasms may overlap with spinal pattern-formation circuits, because of their mixed-synergy co-activation patterns and the interaction with proprioceptive inputs, specifically the entrainment of muscle spasms by imposed oscillatory hip movements[24,41]. A study in a mouse model of chronic SCI suggested that muscle spasms are generated by persistent activity in excitatory interneuron populations, while a delayed activity in inhibitory neural circuits curtails the muscle spasms[27]. Another study in mice identified a specific class of locomotor-related excitatory interneurons, which receive extensive direct sensory afferent input and have commissural and propriospinal connections, as candidates for first-order interneurons in sensory-evoked spasms[26]. These V3 interneurons are associated with pattern formation, but do not intrinsically oscillate or initiate locomotion by themselves[26]. We propose that spinal myoclonus taps into the pattern-formation circuits of muscle spasms. The candidate circuits likely involve first-order interneurons that had become highly excitable under the specific pathophysiological conditions, resulting in quasi 'hard-wired' downstream excitatory actions across the midline and upon other spinal cord segments, as suggested by the consistent multi-muscle activation across the different spinal myoclonus examples (see also Fig. 2 in Calancie[13]). These

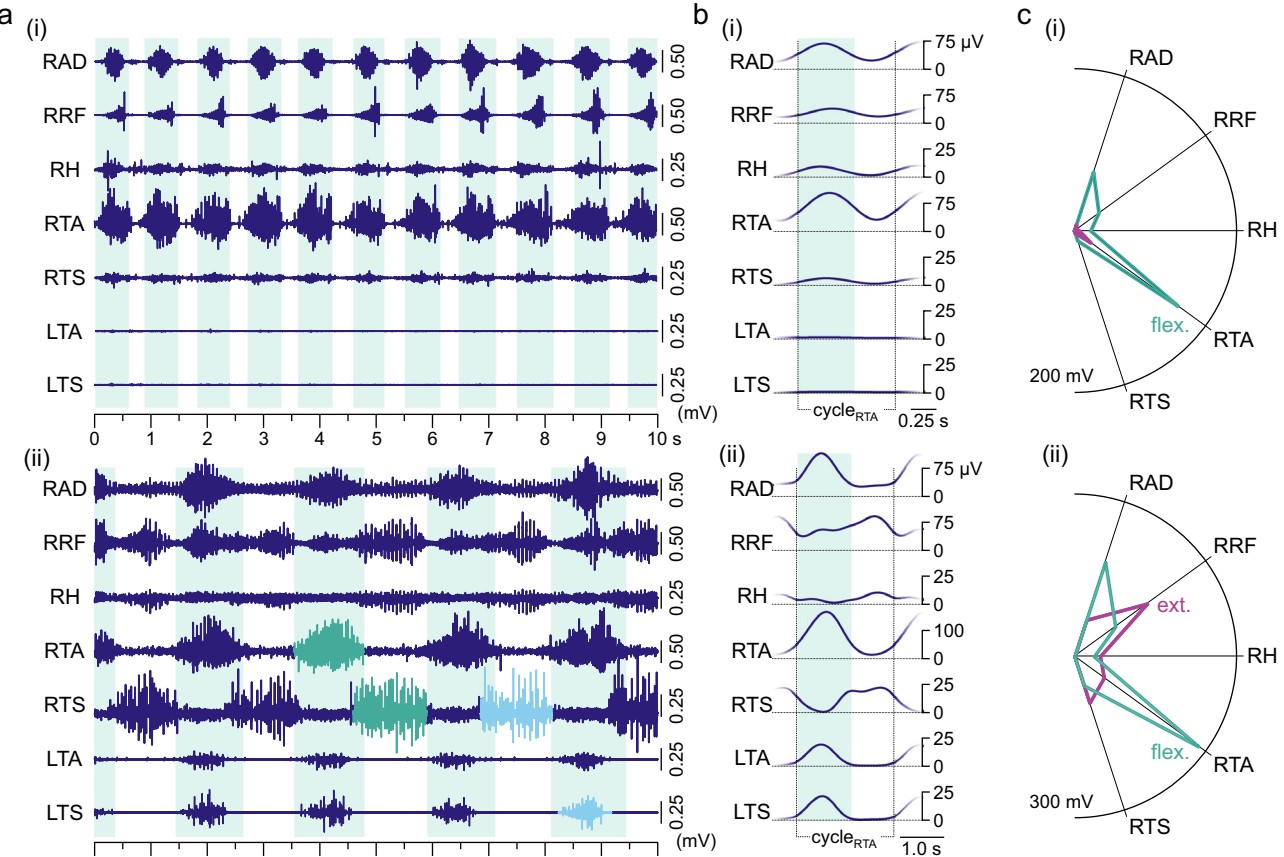

**Fig. 6 | Rhythmic patterns of electromyographic activity in the paralyzed lower limbs evoked by tonic epidural stimulation.** The lumbar spinal cord is capable of producing stereotyped rhythmic motor outputs with different multi-muscle activation patterns and a range of cycle frequencies in response to epidural electrical stimulation (EES). **a** (i) Fast rhythmic activity with a cycle frequency of 1.01 Hz and alternation between single phases of synchronous activity in multiple muscles and brief silent phases. (ii) Rhythmic activity with a cycle frequency of 0.45 Hz showing clear alternation between two distinct phases of muscle activation. This pattern exhibits essential elements that define CPG-associated locomotion in reduced animal models[6], i.e., reciprocal activity in antagonistic muscles (green bursts) and in some homologous muscles of both legs (light blue bursts). Further complexity is seen in the double-bursting pattern of the bifunctional rectus femoris (RF; hip-flexor, knee-extensor). The backgrounds mark phases of dominant activation of the monofunctional flexor muscle tibialis anterior (TA) of the right lower limb. EES was applied throughout the entire time windows displayed, (i) recording EES 4, active contacts: 0+1−, stimulation frequency: 29.4 Hz, stimulation amplitude: 7 V, pulse width: 210 μs; (ii) recording EES 3, 0−3+, 29.4 Hz, 6 V, 210 μs. Same location of the epidural lead, with the four in-line electrodes, labeled as 0 to 3 from rostral to caudal, located at the T12 vertebral level. **b** Mean electromyographic (EMG) envelopes averaged from all rhythmic cycles displayed in a. **c** Mean EMG integrals of the rhythmic activity, shown separately for the phases with a dominant activation of the right TA (flex., flexion-like phase) and for the phases in-between (silent phases in (i) and ext., extension-like phases in (ii)). Note the strong activation of the right TA that showed little activation during spinal myoclonus. Radial axes of the polar plots are muscles and polar coordinates are mean integrated EMG activities. AD, adductors; CPG, central pattern generator; cycle_{RTA}, one cycle of rhythmic activity starting from the onset of right TA bursts; H hamstrings muscle group; L left; R right; TS triceps surae muscle group. Source data are provided in the Source Data file.

pathways further receive extensive proprioceptive input from the hip, knee, and ankle as well as cutaneous input from the foot. The major distinction of spinal myoclonus from muscle spasms would then be its rhythmic nature.

The low rhythm-cycle frequencies, together with the fact that motor deletions (see below) were not observed during spinal myoclonus may suggest an alternative rhythm-generating mechanism to that involved in the EES-induced rhythmic activities. The rhythmic nature of spinal myoclonus may have resulted from the interaction of an aberrant, ongoing excitatory drive below the SCI with a self-limiting mechanism in spasm generation. All but one of the individuals presenting with the Bussel-Calancie type of spinal myoclonus reported in the literature had a combination of a SCI and an additional pathology below the lesion, including hip pathologies in three cases[13,14]. It has been suggested that these pathologies may have generated a tonic, probably nociceptive afferent inflow to the lumbar spinal cord, acting as a driving input to support the sustained rhythmic neural activity[13]. Their direct treatment successfully suppressed spinal myoclonus[13–15].

In the participant of the present study, short-lasting analgesia of the left hip reduced the repetitive bursting of spinal myoclonus, but not the muscle spasms. Long-term control was only achieved after a total hip replacement surgery (see Methods). Animal experiments suggested the activation of a broad spectrum of inhibitory interneurons during muscle spasms that are involved in the delay of their onset as well as in their termination[27]. Our data implied that the silent periods within spinal myoclonus episodes were not related to a refractory behavior following each burst (Fig. 4). An alternative explanation could be a self-limiting mechanism through the recruitment of inhibitory circuits. We have further shown that the excitatory input to the lumbar spinal cord provided by subthreshold EES could promote the onset of spinal myoclonus (Fig. 5). This observation supports a complex interaction between excitatory and inhibitory circuits in triggering spinal myoclonus bursts. Spinal myoclonus may hence be a form of repeatedly triggered muscle spasms by a sustained nociceptive afferent drive without necessarily requiring the activation of an intrinsically rhythm-generating network of a locomotor CPG (Supplementary Fig. 5).

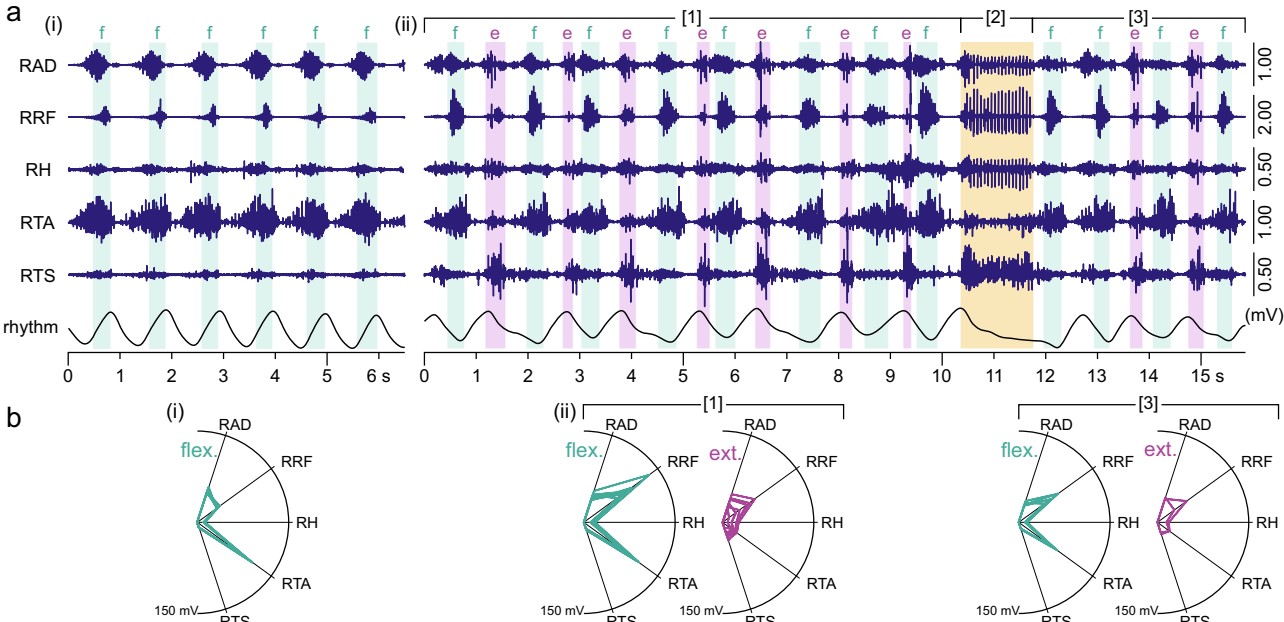

**Fig. 7 | Changes in rhythm and pattern of electromyographic activity evoked by epidural stimulation during the same recording. a** Electromyographic (EMG) activity with (i) robust periodicity and synchronous bursts in multiple muscles of the right lower limb changes to (ii) a locomotor-like pattern with slower rhythmicity (rhythm; inclinometer recordings from the knee). Different phases of activity are shown in relation to the on- and offsets of EMG bursts in the bifunctional rectus femoris (RF; hip-flexor, knee-extensor) identified by a thresholding technique. Two distinct phases were determined, a flexion-like (f) and an extension-like (e) phase. The periodicity in the locomotor-like EMG pattern [1] was interrupted by a period of reduced activity in tibialis anterior (TA) and continuous firing in the other muscles [2]. Following a sharp termination of this motor event, the rhythmicity resumed [3] with two consecutive flexion-like phases before the regular pattern continued. **b** Polar plots of muscle activation within time periods given by on- and offsets of RF bursts show distinct and robust recruitment patterns for the flexion-like (flex.) and the extension-like (ext.) phases. Separate polar plots are shown for the locomotor-like rhythmic EMG activity before [1] and after [3] the period of interrupted periodicity. Radial axes are muscles and polar coordinates are mean integrated EMG activities. Recording EES 4, active contacts: 0+1−, stimulation frequency: 29.4 Hz; stimulation amplitude: a(i) 7 V, a(ii) 8 V, pulse width: 210 μs. AD adductors; EES epidural electrical stimulation, H hamstrings muscle group, R right, TS triceps surae muscle group. Source data are provided in the Source Data file.

Alternatively, the rhythmogenesis of spinal myoclonus and EES-induced rhythmic activity could have emerged in separate generators or through different modes of operation of spinal neural circuits. In mammals and other vertebrates, CPGs can produce distinct rhythmic behaviors involving a common set of muscles, not just locomotion, but also various forms of scratch and fast paw shake[42]. The different rhythms can be generated by separate lumbar neural circuits[43], by different CPGs with largely shared rhythmgenerating components[44,45], and by specialized control mechanisms realized by reconfigurations of the rhythm-generating circuits[46]. It should be noted, however, that spinal myoclonus bears little resemblance to the fast and unilaterally expressed rhythmic behaviors of scratch and paw shake.

Lumbar EES at 30–90 Hz induced rhythmic electromyographic activity in the paralyzed lower limbs, as previously observed[16–18]. The lumbar spinal cord below a clinically complete SCI and under EES may be the human model that most closely fulfills the criteria of an isolated spinal cord required for the demonstration of CPGs[6]. Proprioceptive feedback from the lower limbs is partially blocked by collision with antidromic action potentials traveling in the electrically stimulated primary afferents, causing a functional deafferentation[6,47]. The orthodromic action potentials within the stimulated primary afferents of the lumbar and upper sacral posterior roots[17,23,47–49] trans-synaptically recruit spinal motoneurons and first-order interneurons[49–51], with the latter subsequently recruiting downstream inhibitory and excitatory circuits[16,20,52,53]. Elements of the CPG could be accessed via direct and indirect central projections of the primary afferents[8,33,54–56]. Here, several observations suggested that EES had recruited CPG elements that were not activated during spinal myoclonus (Supplementary Fig. 14). Regarding pattern formation, EES induced both synchronous bursting

as well as component locomotor patterns. In analogy to the findings in the marmoset monkey, the latter finding suggests a partial activation of the CPG circuits generating component fictive patterns that, when combined, would resemble a true locomotor pattern[11]. Regarding rhythm generation, we observed spontaneous errors in the EES-induced rhythmic patterns, which we propose to be flexor deletions. Motor deletions are transient phases of absent activity in one set of motor pools during otherwise robust fictive locomotion, often accompanied by sustained activity in antagonist motor pools[57]. They have been associated with transient failures in CPG operation under conditions of reduced sensory feedback[19,57]. In fictive locomotion preparations, both extensor and flexor deletions can occur. In the adult decerebrate cat, extensor deletions with unperturbed rhythmic flexor bursting were described[19], while there were no clear examples of flexor deletions with unaffected rhythmic extensor bursting, cf.[9]. In the isolated neonatal mouse spinal cord, flexor deletions were always accompanied by sustained firing of ipsilateral extensor motoneurons[9]. The motor deletions described here resembled the flexor deletions during fictive locomotion in the isolated mouse spinal cord in a remarkable way[9] (Supplementary Fig. 10). Here, the lower limbs were not manipulated during the deletions, nor were the stimulation parameters changed. Following a deletion, the rhythmic electromyographic activity resumed with the pre-deletion rhythm-cycle frequency. In six of the 12 deletion examples, EES had generated unilateral electromyographic activity only, making it unlikely that rhythmic activity in the contralateral hemicord would have entrained ipsilateral rhythmicity following the deletions. We hence propose that we have observed spontaneous motor deletions during rhythmic activity generated by the spinal cord, previously only reported in animal studies. When regarded as a fundamental characteristic associated

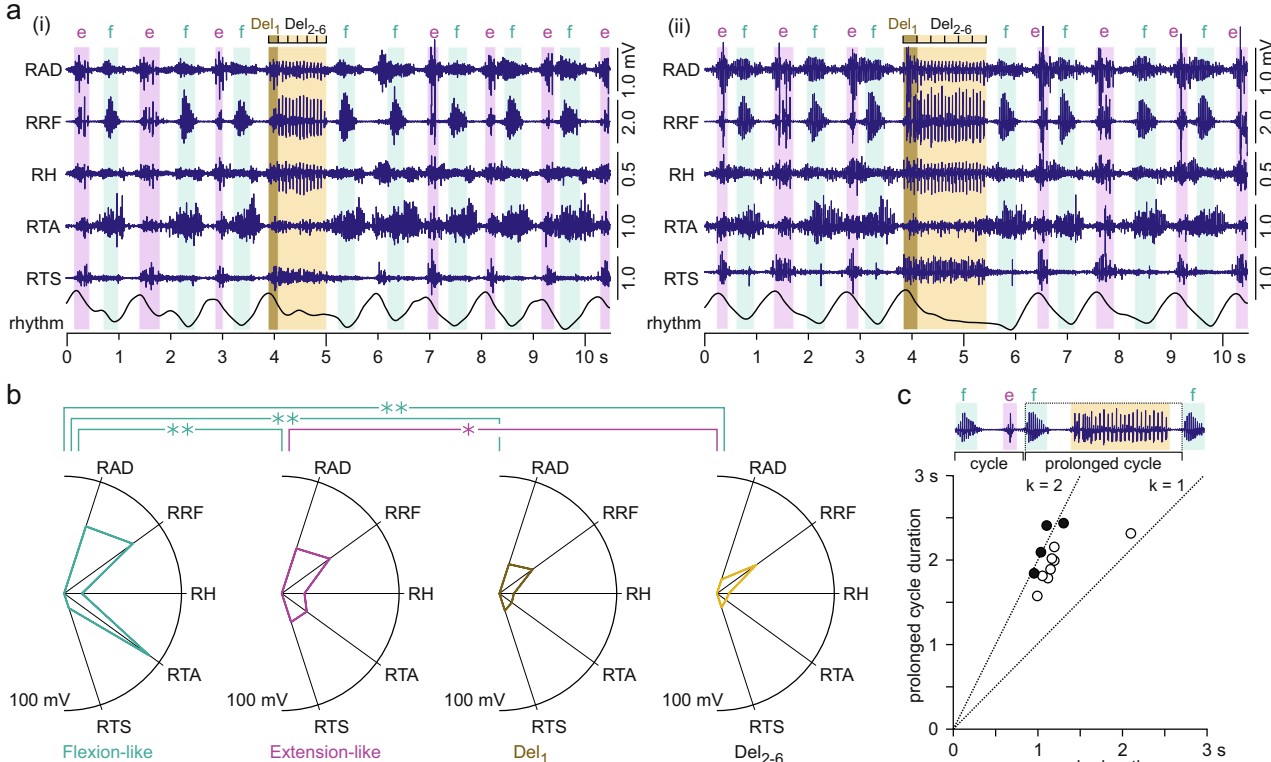

**Fig. 8 | Motor deletions of regular rhythmic electromyographic activity.**
**a** Examples of interrupted rhythmicity of electromyographic (EMG) activity generated by epidural electrical stimulation (EES) of the lumbar spinal cord. On- and offsets of rectus femoris (RF) activity, identified by a thresholding technique, were used to determine flexion-like (f) and extension-like (e) phases based on its double-bursting pattern and to determine phases containing motor deletions (yellow backgrounds). For further analysis, the deletion phases were divided into six time windows of equal durations ($Del_1$, $Del_{2-6}$). EES was applied with unchanged parameter settings throughout the displayed time windows, recording EES 4, active contacts: 0+1−, stimulation frequency: 29.4 Hz, pulse width: 210 µs, and stimulation amplitude: (i) 8 V and (ii) 9 V. **b** Polar plots of muscle activation for flexion-like and extension-like phases of the regular rhythmic activity, for the initial phase of deletion, $Del_1$, and the average of the next five time windows, $Del_{2-6}$. Radial axes are muscles and polar coordinates are mean integrated EMG activities per phase as indicated, averaged across the $n = 12$ examples of rhythmic activities containing motor deletions. A linear mixed model revealed statistical differences between phases, with brackets and asterisks denoting significant results of the post-hoc Bonferroni-corrected pairwise comparisons (*, $p < 0.05$; **, $p < 0.001$). **c** Scatter plot illustrates mean durations of regular rhythm cycles (x-axis) relative to the prolonged cycle durations containing a motor deletion (y-axis), defined as the time between the offset of the last rhythm cycle before the motor deletion to the onset of the first rhythm cycle post-motor deletion. Black circles are examples where rhythmic activity resumed after an integer number of missing cycles, revealed by a statistical analysis using formula (1) in the Methods section (all $p < 0.05$), identifying the motor deletions as being non-resetting[9,19]. Inserted lines have slopes of $k = 1$ (ratio of mean regular rhythm-cycle duration to prolonged cycle duration: 1:1) and $k = 2$ (ratio 1:2). AD adductors, DEL deletion, H hamstrings muscle group, R right, TA tibialis anterior, TS triceps surae muscle group. All statistical tests were two-sided. Source data are provided in the Source Data file.

with fictive locomotion in reduced animal models, the motor deletions investigated here are adding an important piece of evidence for the existence of a CPG for locomotion in humans[6,56].

During fictive locomotion in reduced animal models, resetting and non-resetting motor deletions exist and have led to the formulation of a two-layer conceptual scheme of the mammalian CPG[19,22,58]. This CPG model proposes a separation of rhythm generation, implemented by a layer of intrinsically rhythmic half-centers, and the distribution of the generated excitatory drive to different motor pools by a pattern-formation layer. Thereby, the terminology of layers does not refer to bands of anatomically contiguous neurons, but rather to functional levels of specialized yet deeply intertwined neuronal circuits. A disturbance of the rhythm-generating layer would directly affect the oscillatory drive to the downstream network for an arbitrary time period and hence result in a corresponding phase shift relative to the pre-deletion rhythmicity (resetting deletion). The maintenance of the phase of rhythmic activity, on the other hand, would suggest a perturbation occurring in the pattern formation layer only (non-resetting deletion), while separate neuronal circuits—those constituting the rhythm-generating layer—continue to compose the timing of the rhythmic activity without disruption. We identified both resetting

and non-resetting deletions using the same statistical method as in animal studies[9,19]. Non-resetting deletions, in particular, strongly support the existence of a rhythm-generating layer in the human spinal cord, i.e., neural networks that intrinsically generate the locomotor rhythm independently of the networks responsible for pattern formation[59]. Interestingly, the two-layer organization of the CPG was first proposed to explain the complex activation patterns of a bifunctional muscle group (posterior biceps-semitendinosus) during fictive locomotion in thalamic cats[60]. Afferent stimulation altered the activity of the muscle in either the flexor or extensor phase without resetting the rhythm. Here, the bifunctional rectus femoris also clearly demonstrated a double-bursting pattern during EES-induced locomotor-like activity.

The unilateral motor deletions in the right tibialis anterior not only resulted in a sustained firing of the other ipsilateral muscles, but also in periods of absent or reduced activity in the contralateral lower limb (Supplementary Fig. 12). This behavior contrasts with findings in animal studies, where motor deletions do not perturb the rhythmic activity on the contralateral side[9]. The predominant expression of EES-induced rhythmic activity in the right lower limb of the participant in the present study was likely a result of an asymmetric effect of the

stimulation[18]. One consequence could have been a lower level of engagement of the circuit elements residing within the left hemicord. The lower-amplitude activity in the left lower-limb muscles with less robust rhythmicity occurred predominantly synchronized with the right tibialis anterior bursts (Supplementary Fig. 11). The right tibialis anterior and the left lower-limb muscles may have thus been driven by the flexor burst generator through shared pattern-formation elements within the right hemicord. Failure in their operation would simultaneously result in unilateral flexor deletion and absent or reduced activity in the left lower-limb muscles.

The data derived from the participant of the present study hint at the activation of an asymmetric, flexor-dominant CPG organization by EES. First, rhythmic electromyographic patterns composed of the flexion-like phases could exist without alternation with extension-like phases (Figs. 6a(i), 7a(i), Supplementary Fig. 6a), but the reverse example was not found. Second, we found flexor deletions only. Data in the neonatal mouse suggested a fundamental asymmetry of the flexor and extensor portions of the hindlimb locomotor CPG[9]. A concept was proposed, in which only the flexor half-center of the rhythm-generating layer is intrinsically rhythmic. The tonically active extensor half-center exhibits rhythmic activity only when receiving rhythmic inhibition from the flexor half-center[9,59]. Asymmetric models of locomotor rhythmogenesis had been already suggested earlier, where intrinsically rhythm-generating modules would have excitatory projections only to the flexor pattern formation half-centers[61,62]. Structurally, flexor and extensor half-centers may include different compositions of inter-neuron classes[59,61,63]. Our observation of an asymmetric CPG organization should be considered as only one mode of operation of the human locomotor networks. Spatially restricted optogenetic activation of excitatory spinal interneurons demonstrated that both flexor and extensor bursting can exist independently, at least in an isolated neonatal mouse spinal cord model[64].

Finally, our data implicated that various elements of the CPG were recruited by EES in a speed-dependent manner. Locomotor-like rhythmic activities and synchronous flexor bursting patterns had distinct ranges of rhythm-cycle frequencies (Supplementary Fig. 9). The functional underpinning of the synchronous bursting is unclear. Yet, the two distinct patterns share a resemblance to the different gait types (walk, trot, gallop, bound) expressed at different speeds of locomotion in quadrupedal mammals[59,65,66].

Spinal myoclonus and EES-induced rhythmic activities, both expressed in the same individual with a clinically complete SCI, revealed that the human lumbar spinal cord harbors different mechanisms for generating rhythmicity, shaping bursts of activity and coordinating them across multiple muscles. The observation of motor deletions may be the best indirect evidence in humans to date for the existence of dedicated rhythm generating spinal circuits. This finding, together with the flexor-dominant operation—at least under EES—, is adding fundamental insights into the organization of the human CPG for locomotion. Our data of component locomotor patterns (Supplementary Fig. 7) and flexor deletions (Supplementary Fig. 10) resembled findings that were derived from invasive procedures in the marmoset monkey[11] and the isolated mouse spinal cord[9] in a remarkable way. The juxtaposition of the organization of human and animal CPGs provides an important context for appreciating the translational value of experimental investigations for promoting function after SCI. Likewise, our results emphasize the importance of CPGs as primary targets for cutting-edge interventions to augment neurorehabilitation outcomes following SCI[67,68]. In the coming years, new generations of transcriptomic analysis[69,70] will be applied to identify neuronal cell types of the CPG at the gene expression level[2,59] in the human spinal cord. The present study and future physiological investigations will be essential in bridging the gap between knowledge of the genetic architecture and conceptual models of the human CPG.

## Methods
The study participant gave written informed consent for all interventions and procedures reported here as well as for data processing and publication, in compliance with the CARE guidelines and the Declaration of Helsinki principles. Retrospective data analysis was approved by the Ethics Committee of the City of Vienna (EK-17-059-VK).

### Patient information
The male participant of this study was aged 23.4 years at the time of the first electromyographic data collection reported here. He had sustained multiple trauma in a traffic accident eleven months earlier, including a compression fracture-dislocation of the T3 vertebra, an impression fracture-dislocation of the T4 vertebra, a fracture of the spinal process of the T5 vertebra, and a luxation of the left hip joint. A total loss of motor and sensory function below the injury was diagnosed one week following the spinal stabilization surgery. Spasticity emerged about two months after the accident, with severe extensor spasms bilaterally and flexor spasms in the left leg. Rehabilitation during the first six months post-injury was compromised by the severe spasms as well as the limited mobility of the left hip. Daily doses of 100 mg baclofen and 12 mg tizanidine were administered with little effect on spasticity. At the end of this rehabilitation period, there was no recovery of sensory or motor function, while spasticity had further aggravated. Two months later, the subject was referred to a clinical program for the treatment of lower-extremity spasticity by EES of the lumbar spinal cord[20]. At that time, he had discontinued anti-spasticity mediation for 4 weeks and indicated no changes in the occurrence of spasms. At admission, the Barthel Index, used to measure performance in activities of daily living, amounted to 55 (max: 100). Examination according to the International Standards for Neurological Classification of Spinal Cord Injury determined a sensory and motor complete SCI classified as AIS A according to the American Spinal Cord Injury Association (ASIA) Impairment Scale (Supplementary Table 1). Neurological level of injury was T3, with the zone of partial preservation extending caudally to T7. Lower limb motor score (total) was 0. Examination of lower limb spasticity showed increased hypertonia during hip flexion and abduction, knee flexion and extension, as well as ankle dorsi- and plantar flexion, and confirmed severe extensor spasms bilaterally and flexor spasms in the left leg. Achilles reflexes were enhanced and developed into sustained clonus. Plantar stimulation with a blunt rod evoked withdrawal-like responses in the left but not the right leg. A standardized electromyography-based neurological examination of lower-limb spasticity[21] was conducted multiple times before and after the implantation with an epidural lead and thereafter, with on and off conditions to assess the effects of stimulation. Data analyzed and reported here were taken from these examinations.

Following the examinations reported here, the epidural lead was removed because of unsatisfying effects on the severe spasms. Short lasting analgesia of the left hip with 10 ml Lidocaine (Xyloneural) reduced the repetitive bursting but not the spasms evoked by passive movements or plantar stimulation. Two weeks following the last recordings of the present study the subject was implanted with a drug infusion pump (Medtronic Inc., Minneapolis, Minn., USA) initially set to deliver 800 µg baclofen per day and later reduced to 600 µg per day. Two months thereafter the subject underwent a total left hip replacement surgery. In a follow-up ten months later, no spasms or rhythmic electromyographic activities were detected in the neurological assessment of lower-limb spasticity.

### Electromyographic recordings
All electromyographic recordings were conducted with the subject in the supine position. Pairs of silver–silver chloride surface electrodes (Intec Medizinitechnik GmbH, Klagenfurt, Austria) were placed bilaterally over the adductors, rectus femoris, the hamstrings muscle group, tibialis anterior, and the triceps surae muscle group with a

longitudinal alignment and an inter-electrode distance of 3 cm[23,53]. A common ground electrode was placed over the iliac crest. Abrasive paste was used for skin preparation to reduce electrode resistance below 5 kΩ. Electromyographic signals were amplified (Grass Instruments, Quincy, MA, USA) with a gain of 2000, filtered to a bandwidth of 30–700 Hz, and digitized at 2002 samples per second and channel using a Codas analog-to-digital converter system (Dataq Instruments, Akron, OH, USA). Inclinometer (Accustar, Lucas Sensing Systems, Phoenix, AZ, USA) data of knee and ankle positions were acquired and used here to document the timing of passive or induced movements.

### Electromyography-based examination of lower-limb spasticity

The electromyography-based examinations were routinely performed at the Neurological Center, Clinic Penzing, both to assess lower-limb spasticity as well as to evaluate the effects of EES. With the subject lying supine, a standardized series of manual reflex testing was performed while electromyographic activity was continuously collected[20,21,71]. The examinations included passive unilateral hip and knee flexion-extension movements performed by an examiner (3 s each for flexion, holding the position, and extension), passive dorsiflexion-plantar flexion of the ankle (2 s each for flexion, holding the position, extension), stretch reflex testing with taps applied to the patellar and Achilles tendons with a reflex hammer, the attempt to elicit a patellar clonus by a quick manual downward movement of the patella or an Achilles clonus by a brisk dorsiflexion of the ankle, and plantar reflexes by stroking the foot sole with a blunt rod in a manner analogous to that used to elicit a Babinski reflex. Tendon taps were applied ten times per tendon and side, and all other tests were repeated three times per side.

### Epidural electrical stimulation

An epidural lead (Model 3487A; Medtronic) was placed with a percutaneous approach through a Tuohy needle into the posterior epidural space of the T12 vertebra under local anesthesia. The lead was externalized and—on the assessment days—connected to a test stimulator (3625-G, Medtronic). The lead carried four cylindrical electrodes arranged linearly, each with a diameter of 1.3 mm and a length of 3 mm, with an inter-electrode spacing of 6 mm. Electrodes were labeled from 0 to 3 from rostral to caudal. Different bipolar electrode setups were tested, by setting one electrode as cathode (−) and another as anode (+). Spinal mapping[23,72] performed with the lowest available stimulation frequency of 5.2 Hz and electromyographic recordings identified an effective position over the lumbar spinal cord with an asymmetrical, right-sided stimulation bias. Bipolar wide-field stimulation with the most rostral electrode set as cathode (0–3+) evoked posterior root-muscle reflexes[17,23,73] in the adductors and rectus femoris bilaterally at threshold, and additionally in bilateral hamstrings and the right triceps surae muscle group with increasing stimulation amplitudes. Bipolar wide-field stimulation with the most caudal electrode set as cathode (0+3−) evoked posterior root-muscle reflexes in the adductors, rectus femoris, and the hamstrings muscle group bilaterally at threshold, and in the right tibialis anterior and triceps surae muscle group with increasing stimulation amplitudes.

On the assessment days, stimulation was applied with various bipolar electrode configurations, frequencies between 5.2 Hz and 120 Hz, and amplitudes ranging from 1 to 10 V while the subject remained relaxed in the supine position or the electromyography-based neurological examination was performed. The rationale was to investigate the effects of EES on spasticity and to identify settings for achieving an antispasticity effect[20].

### Data analysis and reproducibility

We studied recordings acquired during nine electromyography-based neurological examinations, one (neurological examination 1 in Supplementary Table 2) conducted prior to, and the others conducted after the implantation of the epidural lead. The clinical examinations used standardized protocols and were not randomized. The examiners were not blinded during experiments. All recordings were derived from the same individual with SCI. The participant presented with a rare form of spinal myoclonus (prevalence <1% in individuals with chronic SCI[13]), expressed as self-sustained rhythmic activity in the lower limbs, and had an implanted epidural electrode lead. No statistical method was used to predetermine sample size. The examinations were carried out over a three-month period, allowing for the collection of multiple data sets and the verification of the reproducibility of the results obtained. The examinations are referred to in chronological order. We considered sections within the recordings containing rhythmic bursts of electromyographic activity across multiple lower-limb muscles, either occurring spontaneously, in response to passive leg manipulation (six examinations, prior to implantation or in EES-off condition; cf. Supplementary Table 2)[12–14], or induced by tonic EES (examinations EES 1–4; example 2 in Supplementary Table 2 and EES 2 derived from the same neurological examination)[16–18]. No data were excluded from analyses. Analyses were performed using Matlab 2017b (The MathWorks, Inc., Natick, MA, USA) and IBM SPSS Statistics 27.0 for Windows (IBM Corporation, Armonk, NY, USA).

### Self-sustained rhythmic electromyographic activity in the lower-limbs

We screened the available recordings for the presence of self-sustained rhythmic electromyographic activities lasting for a minimum of 10 s and identified eleven examples (Supplementary Table 2). The respective sections were extracted from the electromyographic recordings for further analysis.

To identify the on- and offsets of the rhythmic bursts, the electromyographic signals of the studied muscle groups of both legs were rectified and low-pass filtered at 2 Hz using a 2nd order Butterworth filter. The onset of a burst was determined separately for each muscle as the time when the data of the low-pass filtered electromyographic envelope exceeded the local minimum preceding the burst by 25% of the difference between the peak value of the burst and this local minimum. The offset of a burst was defined accordingly. Data derived from a given muscle were excluded from further analysis if no rhythmic bursts were detected or if the local maxima of the rectified, unfiltered electromyographic signals were below 25 μV. Burst durations were calculated as the times between the respective on- and offsets, interburst durations as the times between burst offsets and the onsets of the successive bursts, and cycle durations as the times between the onsets of consecutive bursts. For Figs. 2a, 6b(i) and (ii), the filtered electromyographic envelopes of each muscle were interpolated to 2000 data points per rhythm cycle, and mean envelopes were obtained by averaging across the available cycles.

Mean cycle durations (mean$_{cycle-duration}$) per muscle and example were determined, and corresponding cycle frequencies (Hz) were calculated as $1/\text{mean}_{cycle-duration}$. Comparisons of cycle frequencies of the rhythmically active muscles within a given example were done with separate Friedman's tests (cf. Supplementary Table 2), and across examples, by running a linear mixed model with muscle (left and right adductors, rectus femoris, hamstrings, tibialis anterior, and triceps surae) and example as fixed factors.

For each rhythmically active muscle and example, separate Pearson's product moment correlations were run to investigate the relationships between the burst durations and interburst durations, and between the integrated electromyographic activity during bursts and interburst durations (cf. Fig. 4). Linear regressions were calculated to investigate whether one scalar would predict the outcome of another.

In ten of the eleven examples, rhythmic bursts of electromyographic activity consistently occurred in the left rectus femoris (cf. Supplementary Table 2) and were used to calculate the relative onset lags of the rhythmic bursts occurring in the other muscles studied (cf. Fig. 2b). A linear mixed model with muscle (all but left rectus femoris)

and example as fixed factors was run to test for potential differences in the pattern of muscle recruitment. Finally, mean relative onset lags per muscle were obtained by averaging over the ten examples.

Muscle activation patterns were further investigated by calculating the integrals of the electromyographic activity per muscle for each burst of the self-sustained rhythmic outputs as well as, separately, for the muscle spasms induced by manipulation of the lower limbs by the examiner and were illustrated by polar plots (cf. Fig. 3). The respective muscle activation patterns were compared by running a linear mixed model with the type of activity (self-sustained rhythmic output, muscle spasm) and muscle as fixed factors.

### Rhythmic electromyographic activity induced by epidural electrical lumbar spinal cord stimulation

We screened the available recordings for the presence of EES-induced rhythmic electromyographic activities lasting for a minimum of 10 s with unchanged stimulation parameters and found six examples for further analysis. On- and offsets of the rhythmic electromyographic bursts induced by EES as well as cycle durations, cycle frequencies and mean electromyographic envelopes per muscle were calculated in the same fashion as in the case of the self-sustained rhythmic outputs. Distinct phases of the rhythmic patterns were identified based on the electromyographic recordings from tibialis anterior, the only monofunctional flexor muscle studied here[18]. Muscle activation patterns were calculated as the integral of the electromyographic activity per muscle, separately for flexion-like and extension-like phases of rhythmic activity. Mean values were obtained by averaging over the respectively available phases in a given example.

### Motor deletions during rhythmic electromyographic activity induced by epidural electrical lumbar spinal cord stimulation

We encountered 12 examples of otherwise robust rhythmic electromyographic activities with phases of absent (expected) tibialis anterior bursts, i.e., motor deletions[9,19], accompanied by prolonged activity in the other muscles. In all examples detected, rectus femoris of the right lower limb displayed a double-bursting pattern during the regular rhythmicity, one burst each during the flexion-like and extension-like phases, and tonic activity during the phases of absent tibialis anterior bursts. We hence used the on- and offsets of the respective electromyographic activity recorded from rectus femoris to distinguish between flexion-like and extension-like phases as well as the phases of interrupted rhythmicity, i.e., the motor deletions.

We applied the statistical analysis used in animal studies[9,19] to test whether the motor deletions were non-resetting (i.e., resulting in no change in cycle period when rhythmic activity resumed) or resetting. To this end, the duration of the phases containing a motor deletion (time between the offset of the last rhythm cycle before the motor deletion to the onset of the first rhythm cycle post-motor deletion) was compared with the durations of five (ten cases) or four (two cases) immediately preceding cycles of regular rhythmicity. We calculated the running average $\bar{Y}_2$ and variance $s_2$ of $X$-cycle intervals with sample size $n_2 = 5$ ($n_2 = 4$, respectively) before the motor deletion, with $X$ being the number of hypothesized cycles during the deletion. The duration of the motor deletion $Y_1$ was then compared with $\bar{Y}_2$ by calculating the $t$ statistic according to formula (1):

$$t = \frac{Y_1 - \bar{Y}_2 - (\mu_1 - \mu_2)}{s_2 \bullet \sqrt{\frac{n_2 + 1}{n_2}}} \tag{1}$$

where $\mu_1 - \mu_2$ is the hypothesized difference ($\mu_1 - \mu_2 = 0$). A significance level of $p < 0.05$ with $n_2 - 1$ degrees of freedom was used to identify significant differences and hence resetting motor deletions.

Muscle activation patterns during motor deletions were analyzed separately for the initial division and the later divisions of these phases. To this end, we divided each motor deletion phase into six divisions of equal duration and calculated the integrated electromyographic activity per division. We obtained a value for the initial phase of the deletion ($Del_1$), and calculated an average value for the final five divisions ($Del_{2-6}$). Muscle activation patterns during flexion-like, extension-like, as well as during $Del_1$ and $Del_{2-6}$ across the 12 available examples were then compared by running a linear mixed model with phase and muscle as fixed factors.

### Statistics

Statistical analyses were performed using IBM SPSS Statistics 27.0 for Windows (IBM Corporation, Armonk, NY, USA).

Assumptions of normality were tested using Shapiro-Wilk tests, and if necessary, data were transformed (reciprocal transformation). Mean values were compared using Student's t-tests or by fitting linear mixed models. If model assumptions were not met despite data transformation, their respective nonparametric equivalents were used. All post-hoc pairwise comparisons were Bonferroni-corrected to adjust for multiple comparisons. α-errors of $p < 0.05$ (two-sided) were considered significant. Effect sizes were reported by the partial eta-squared ($\eta_p^2$) for linear mixed models, by Kendall's W for Friedman's tests, and else by the correlation coefficient $r$. Descriptive statistics are reported as mean ± SE for normally distributed data and else as median and IQR.

### Reporting summary

Further information on research design is available in the Nature Portfolio Reporting Summary linked to this article.

## Data availability

The authors declare that the data supporting the findings of this study are available within the paper and its supplementary information files. Source data are provided with this paper.

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

## Acknowledgements

This study was supported by the Austrian Science Fund (FWF), proj.nr. I 3837-B34, U.S.H.

## Author contributions

Conceptualization: K.M. and U.S.H.; Methodology: K.M., A.B., and U.S.H.; Software: A.B and U.S.H.; Analysis: K.M., A.B., and U.S.H.; Data Curation: P.L., H.B., B.F. and U.S.H.; Data Interpretation: K.M., U.S.H., P.L., and H.B; Writing—Original Draft: K.M. and U.S.H.; Review & Editing: all authors; Visualization: U.S.H.; Supervision: U.S.H., B.F., P.L., and H.B.; Funding Acquisition: U.S.H.

## Competing interests

The authors declare no competing interests.
