## [Peer Review File · Nature Communications]

Rare phenomena of central rhythm and pattern generation in
a case of complete spinal cord injuryREVIEWER COMMENTS

Reviewer #1 (Remarks to the Author):

Minassian, Hofstoetter and colleagues characterized patterns of rhythmic motor activities in the legs of an individual with a complete thoracic spinal cord injury (SCI). Although this has been done by others (e.g. Bussel, Calancie, Nadeau), the present study performed a more detailed analysis and used electrical epidural stimulation (EES) of the lumbar cord to compare with the other different evoked patterns. There are some novel components and careful analyses like these in people with SCI can provide important insights on human CPG organization. However, the paper is not always easy to follow because some jargon is not explicitly defined and some results are presented without context. If adequately revised and better vulgarized (accessible to the non-expert), the paper could make a strong addition to the literature.

Major comments

1. Neurophysiological and clinical terms need to be clearly defined. What are the differences between spinal myoclonus, ankle clonus and muscle spasms? Do they share common CPG circuits or not? The term deletion needs to be clearly defined (see minor comments, line 59). Other terms that need definition include 'anterior-dominant synergy' (line 262), flexor-biased (line 262) and extensor-biased (line 263). That whole section starting on line 293 is difficult to follow.
2. Context and appropriate references need to be provided for some results, such as the relationship between burst/interburst duration and cycle period. Explain how this relates specifically to rhythm and/or pattern generation.
3. From a neurophysiological perspective, the discussion is hard to follow, in part because the different rhythmic activities (spinal myoclonus, ankle clonus, spasms, locomotor-like activity) are not clearly defined.

Minor comments

Line 34-35. There is some confusion here. In cats, fictive locomotion in all preparations can only be demonstrated with a neuromuscular blockade (curarization). In spinal preparations, pharmacology is required in addition to dorsal root or dorsal column stimulation. In the spinal preparations, the animal is also decerebrated. Sentences need to better describe the preparations.

Line 37. I do not think it has been demonstrated which types of receptors are present on CPG neurons because CPG neurons have not been identified.

Line 48. Study by Nadeau et al. 2010 should also be cited here.

Line 57. Change 'existed' to 'exist'.

Line 59. Deletions are not interruptions of ongoing rhythmicity. They are an absence of activity, often accompanied by sustained activity in antagonist motor pools. Deletions can be resetting (change in timing of rhythmicity) or non-resetting (maintaining the timing of rhythmicity).

Line 73. It is not clear what new theories of spinal rhythm and pattern generations are formulated in the present study.

Figure 1. It is not clear why the three rhythmic activities do not have the same length of time (15 s, 30 s and 60 s). The timescale is the same, but having the same length would help compare the frequency and characteristics of the different activities.

Lines 84-85. How are the different rhythmic activities shown in Figure 1 clearly different from ankle clonus in extended data Figure 1? All show synchronous activity in different motor pools. Here again, using the same length of time would facilitate comparisons.

Line 85. Please define 'quasi-stable'.

Line 86. ... that lasted for a minimum (of) 10 s ...

Line 103. Muscle was not a significant factor for what?

Line 129. Why are these very rare forms of spinal myoclonus? How are the forms of spinal myoclonus different from ankle clonus?

Line 135. The term 'proprioceptive inflow' is not clear. Define 'slow' passive movements. Slow is a relative term.

The color scheme is impossible to follow for colorblind people (reviewer included). Magenta and turquoise are not colors that colorblind people can differentiate from other colors. Please use a colorblind friendly palette.

Figure 4. The relationships between burst/interburst duration and cycle period have been studied extensively in fictive locomotor and scratch preparations in decerebrate cats (Yakovenko et al. 2005 J Neurophysiol; Frigon and Gossard 2009 J Physiol, 2010 J Neurosci). These studies show that dominance can change depending on the preparation or with descending and sensory inputs.

Line 226. What is the invariable pattern of myoclonus?

Line 379. Invariable patterns? I do not understand how a pattern can be invariable.

Lines 380-381. How are muscle spasms not rhythmic?

Lines 397-399. It is not clear how muscle spasms (rhythmic or not?) bypass the activation of a rhythm-generating network.

Lines 409-410. State the several observations. Do you mean that locomotor-like activity and spinal myoclonus are generated by different CPGs or by shared networks with different mechanisms?

Line 420. What is meant by 'characteristic' interruptions?

Lines 421-422. This is not the definition of a deletion.

Line 442. Perret and Cabelguen were the first to propose the two-layered hypothesis.

Perret, C., Cabelguen, J.M., Orsal, D., 1988. Analysis of the pattern of activity in "knee flexor" motoneurons during locomotion in the cat. In: Gurfinkle, V.S., Ioffe, M.E., Massion, J., Roll, J.P. (Eds.), *Stance and Motion: Facts and Concepts*. Plenum Press, New York, pp. 133-141

Reviewer #2 (Remarks to the Author):

The manuscript by Dr. Minassian et al. reports on findings from an individual with a thoracic SCI who displays a rare form of rhythmic spinal myoclonus and has an epidural stimulator to control spasticity. By comparing EMG patterns during myoclonus (spontaneous and evoked) and epidural electrical stimulation, the authors conclude that myoclonic and locomotor-like rhythmic patterns are generated by two distinct spinal mechanisms. This is significant because it goes against prior suggestions that myoclonus is generated by central pattern generating circuits. Therefore, it must be properly and strongly supported by data. The authors also report motor deletions during activity evoked by EES which represents the first evidence in humans and suggests a separation of rhythm in pattern, following the same logic as has been shown in cat and rodent locomotion. The majority of the paper is based on data from a single individual. This should be kept in mind but the observations (and analyses) from this individual contradict certain prior notions and lend support

to others, which makes the study quite interesting.

1. The muscle pattern activation in myoclonus (with EES, except for the 5V) in Fig 5B (and some of the examples in Fig 3C) look remarkably similar to purple (phases in between TA activations) in Fig 6Cii, 7B, and 8B. Although it is clear that the EES is recruiting the TA-dominant synergy during the 'on phase', is it possible that the extensor-dominated synergy is common to both myoclonus and EES-evoked CPG-like activity? If this is the case, it does not necessarily suggest distinct networks but a common rhythmic network with elements that are additionally capable of rhythmicity when independently recruited (see Hagglund et al 2013, Proc Natl Acad Sci, 110:11589-94). If the myoclonus and extensor-phasing of the EES-evoked activity are not similar, data should be shown to support this conclusion. Although the activation patterns look very similar, the timing element may be entirely different but the EES extensor-phase activity would need to be analyzed similarly to Fig 2A and B to be able to assess.

2. Interestingly, the EES-evoked patterns look largely synchronous and, judging by examples displayed alone, it seems that alternation in any opposing muscles only occurs when evoked activity is at lower frequencies. If there is a frequency-dependent effect (frequency of motor activity during EES, not EES itself), that could be parallel to rodent work demonstrating distinct underlying left-right circuitries activated in a speed-dependent manner and related to gait.

3. There is reference to analysis of a prior study but what analysis was carried out is not clear, beyond examples in Ext Data Fig 2. Please make clear in the text (and legends) exactly when the other study is being referred to and when the transition back to the single patient occurs.

4. The observed deletions are quite compelling and are exciting potential confirmation of similarity of mechanism in humans and quadrupedal animals. Making the resetting vs non-resetting call is difficult. It is impressive how well these line up with $k=2$. I would agree that 4 are non-resetting but it seems to be at least 4 and possibly more are non-resetting, if the variability window of undisturbed cycles is considered.

5. In the 6 examples of deletions with bilateral hindlimb activity, what did the contralateral limb look like? This would provide additional information that can be paralleled to (or contrasted with) animal work.

6. In some places, it escapes that this is all based on one individual so conclusions should be guarded. For example, in lines 453+, is this in general or just in this patient who has unusual pathology?

Minor

- Line 28: "present" to "represent"
- Lines 73-74: I recommend toning down "new theories". Perhaps state what the findings suggest.
- Lines 103 and 110: "no significant" to "not a significant"
- Fig 2A – the onset lines are difficult to see the differences between the colors.
- It is not clear what the 'rhythm' is aligned to in Fig 7A.

Reviewer #3 (Remarks to the Author):

Summary

This submission is a case study describing findings from a series of neurological examinations of an adult male who sustained a neurologically-complete traumatic spinal cord injury (SCI). The subject presented with severe spasticity, and the treatment plan included implantation of an epidural spinal electrode; it was thought that electrical stimulation over the lumbar spinal cord enlargement might ameliorate his spasticity. Over the course of multiple ($n=11$) clinic visits, a standardized protocol of limb manipulation and sensory inputs was applied to evaluate spasticity. Several of these visits preceded electrode implantation, while the remaining evaluations were carried out after the electrode was placed. All evaluations included continuous recordings of EMG from multiple lower limb muscles bilaterally.

It appears that a decision by the authors to publish this work was arrived at only after all evaluations were concluded. Nevertheless, I have no concerns about subject consent or related (e.g. confidentiality) issues.

What was noteworthy about this case is that the subject experienced multiple instances of rhythmic lower limb movements that – while usually emerging from common flexor- or extensor-biased spasms typical of a person with SCI in response to the sensory inputs being delivered – persisted well after cessation of the inputs, and contained elements consistent with a central pattern generator (CPG) for locomotion. Such activity has been reported previously, but must still be considered rare in humans. It's also interesting to see that – like several earlier publications – a major driver of this subject's extreme spasticity leading to these novel movement patterns seemed to be pathology in his hip, ultimately treated surgically through a complete hip replacement.

This is not just another example of severe spasticity resulting in spontaneous leg movements in a human. Rather, what makes this case study unique is how these spontaneous movements were impacted by the delivery of epidural stimulation, shifting the pattern of leg movements without stimulation from one reflecting spinal myoclonus (bilateral; all muscles contracting more or less simultaneously; high reproducibility in rate of contraction) to one more like locomotion (alternation between flexors and extensors; reciprocity between the legs) when the same sensory inputs were delivered while epidural stimulation was 'On'.

Based on the ability to contrast spontaneous leg movements with epidural stimulation turned 'on' vs 'off', the authors propose that the myoclonus-like movements are a consequence of repetitive spasms driven through one 'layer' of spinal circuitry, whereas the locomotion-like movements (seen only during epidural stimulation) reflect activity within a 'higher' level of spinal circuitry that includes more elements of the spinal cord's CPG.

Another finding that receives significant attention is the sometime occurrence of motor deletions when epidural stimulation is 'on' – brief interruptions of EMG from certain muscles during periods of rhythmic movements that otherwise show high reproducibility within and across muscles over multiple movement cycles. These deletions have been previously described in animal preparations, but according to the authors have never – until now – been reported in humans. These deletions are taken as further evidence for the existence of CPG circuitry for locomotion in humans.

The authors do a nice job of placing their findings in the context of previous descriptions of such leg movements in humans, and in reduced animal preparations. The submission is very well written, data analysis is thorough, and accompanying Figures are both well-designed and highly complementary to the text. Despite some questions I have concerning several of the major conclusions (see below), publication of this work would advance the field by adding to the understanding of how the human spinal cord circuitry for rhythmic motor output – like locomotion – is organized. This knowledge in turn could be useful for ongoing efforts to restore locomotion in appropriate individuals with SCI through combinations of epidural stimulation and/or pharmacologic agents.

Critique

A. Big things

I like this paper. However, I'm not entirely convinced about some of the statements made, and would ask that the authors consider the following comments, should they be given the opportunity to revise this work.

1. 'Spasms as a reflection of pattern generation.' I don't disagree that the same muscles that were recruited into a typical spasm also participated in the rhythmic movements you describe, but what about the triggering factors? In my experience virtually all spasms I see (and the examples you include in the early portions of several of your figures are consistent with my experience) have a critical factor that does not seem to be considered in your hypothesis: activation history. That is, once a spasm has been triggered and run its course, the probability that an identical trigger will lead to an identical spasm immediately after the initial spasm has ceased is virtually nil. Different subjects vary dramatically, of course, in the minimum interval before which spasms can be

(re)elicited, but there is always some interval at play. In marked contrast, the rhythmic movements you describe are, by your own definition, sustained for at least 10 seconds, and may continue for a minute or more. In other descriptions from the literature these movements can persist indefinitely. So: why are some of these spasms so difficult to elicit on a continual basis, while other examples of – if I understand you correctly – what you consider to be the same spinal output recur with almost clock-like regularity? In the absence of an obvious answer, I find this conclusion of yours to be not terribly compelling.

2. Are the 'motor deletions' described in this subject indeed a 'first-in-human' report of the same? In your citation #14, Calancie et al. reported that the ongoing, rhythmic and reciprocal leg movements described in that paper were completely shut down during episodes of bladder emptying (description only; no Figure was included to illustrate), and were either shut down or heavily attenuated during neck flexion (dural stretch?) and applied plantar flexion of the toes (illustrations were provided). While the neck and toe manipulations could be argued to reflect additional proprioceptive (??) inputs, the same cannot be said for bladder emptying. Is it possible that the episodes of motor deletions you describe could also have been a consequence of bladder emptying, or some other manifestation of autonomic-related (i.e. not readily apparent to the investigators) activity within the CNS?

B. Little(r) stuff (in order of appearance)

Figure 3, caption for 'C': Muscle activation patterns during spasms and ...

Pg 8, line 203: I don't like the use of 'phase-advanced' (here & elsewhere) to describe your findings. In response to a relatively discreet input (onset of limb manipulation), activity occurs at an earlier latency under conditions of higher EES voltage. Since there has not yet been any cyclic activity occurring within which the 'phase' can shift, how can there be a phase 'advance'? You're likely already sympathetic to this argument, whereby for the same concept you use the language "... highlight the decrease in delay between the cessation ..." in the legend for Figure 5 (pg 9, line 214).

More on Figure 5.

- In the records adjacent to the '2V' series, there is clear activity within both the L & R RF muscles immediately after the offset of leg manipulation. Granted this activity is much smaller in magnitude than subsequent cycles, but are you OK with disregarding it completely?

- In the records adjacent to the '5V' series, the onset of activity in both RF muscles occurs well before leg manipulation has ended. This is not accurately reflected in Figure 5A (arrow placement) or 5C (horizontal line position).

Extended data Figure 2: titles for B (iii) and B (iv) should be B (i) and B (ii), for consistency

- also, it seems from this Figure, and elsewhere in the records that stimulus voltage played a much greater role in influencing motor output than did stimulation rate; could you comment on this? Did you have any particular reason for manipulating stimulus rate over such a wide range?

Methods; pg 30, line 717, typo in 'spams' (spasms)

Methods; pg 32, line 765: you've got cathode (should be +) and anode (should be -) mixed up

Blair Calancie

Reviewer #4 (Remarks to the Author):

This manuscript 'Spinal myoclonus, epidural electrical stimulation, and motor deletions: Rare phenomena of central pattern generation in complete spinal cord injury' is a study that investigates an individual with complete SCI with a rare form of self-sustained rhythmic spinal myoclonus in the legs as well as rhythmic activities induced by spinal cord stimulation. Several findings are interesting and represent new facts for the field of spinal cord injury and neurophysiology.

According to the website, Nat. Comm. rejects about 60-80% of all submissions without peer review based on impact, methodological advances, and interest in interdisciplinary readership, and the rest of the manuscripts expected to meet these criteria. To briefly summarize, from a methodological standpoint, all conclusions in this study were made based on analysis of electromyographic patterns and would require future validation using a combination of techniques. This study was performed on one subject and the validity of presented findings requires future study with analysis of data from several subjects. In one tested subject no specific assessment was conducted to exclude the role of residual fibers, particularly when tested with EES. Finally, it is not clear how presented in this study findings conceptually change what we already know about CPG and most importantly how they would advance the field. I also have to acknowledge that outside of criteria set by the journal, this work demonstrates important observations for the field of spinal cord neurophysiology.

Nat. Comm. expects reviewers to make their decision based on criteria for publication: quality of the data; level of support for the conclusions; potential significance of the results, and I will focus my evaluation mainly on these topics.

The quality of the data is excellent as authors using well established in their group assessment based on EMG analysis. The second criteria, the level of support for the conclusions is not adequately met, all conclusions in this work are based on one type of indirect analysis (EMG) performed just in one subject. Main claims of this work are primarily coming from discussions of previous research and new findings are only indirectly supporting these claims. The third criteria, potential significance of the results is also not evident. Conceptualization of CPG and evidence of CPG in humans were broadly discussed. Conceptual separation of CPG on pattern generator and pattern formation is largely hypothetical and wasn't specifically linked to any of identified circuitries or their specific location. It is hard to say if presented results adding to what we already know about the CPG organization. From presented evidence, the impact of these results on previously established concepts of CPG from animal and human studies is questionable. It is particularly unclear how these results change and advance already known concepts? It is also a question how this report will move the field forward, at least authors did not provide justifications or links to the future steps. Again, these key criteria are set by Nat. Comm. Journal, and I strongly believe that this work could be a good fit for J. Neurophysiology or J. of Neurotrauma as an important case report.

Another key point, that provided justifications, can't support the proposition that results of this work can be generated only on one unique subject and can't be carefully collected from other cases of combined injuries. Combined SCI and hip injuries are not rare and could be selected for comparison across several subjects and that would be critical for future assessment of the results variation.

The key conclusions of this work, like 'that myoclonus taps into spinal circuits generating muscle spasms rather than reflecting CPG activity as previously suggested' represent rather focal interest and also requires future confirmation. In several places authors emphasize how specifically are discussed in manuscript findings and, unfortunately, no clear justification of importance of these findings are provided to help project these results to a wider population or translation to improve understanding and treatment strategies. The main conclusion "These findings argue strongly for the activation of neuronal networks in the human spinal cord that generate the locomotor rhythm independently from elements responsible for pattern formation" is questionable and for described above reasons I can't agree that this conclusion can be completely justified with provided in this study results.

In several places authors consider that subject has isolated lumbar spinal cord and further discuss that "the lumbar spinal cord below a clinically complete SCI and under EES may be the human model that most closely fulfills the criteria of an "isolated spinal cord" required for the demonstration of CPGs." At the same time, most of the recent result demonstrate that patients with motor or motor and sensory complete SCI have residual connectivity and based on assessment and/or EES effect to facilitate volitional movements, can be considered as incomplete. From presented data it is not clear if tested subject was carefully evaluated and if he demonstrated any signs of incomplete SCI. If he was considered as anatomically complete, that

would require detailed analysis of performed evaluation that should be provided. Considering the complex trauma in this subject, EES effect to facilitate volitional control could be performed on unaffected leg with implanted lead or could be done later after joint replacement with non-invasive transcutaneous stimulation or with percutaneous trial EES. Missing assessment of the role of residual connectivity, unfortunately, has an impact on the key conclusions of this study.

Outside of this critique, I must emphasize that this work is well-designed and as a case study definitely important for the field of spinal cord neurophysiology. There are no concerns that this work would represent a good case for future discussion and for targeted audience, although, it is hard to see this study as interdisciplinary with interest from a broad audience. Considering all mentioned concerns, I have to defer the decision on this manuscript to the Editor, since the key criteria set by the journal are not met. In case of future consideration, it would be very helpful for reviewers and authors to understand justification and how that meet with preselection criteria set by journal.

Minor points:

Authors should cite relevant works at the end of the sentence: "The electromyographic patterns largely involved synchronous discharges across muscles. In spite of the lack of a locomotor pattern, it was suggested that this type of self-sustained rhythmic activity was due to a partial release of a CPG."

From provided information it is hard to determine the exact parameters of EES and, particularly, more details on stimulation frequency would be helpful. Authors mentioned different numbers across manuscript, that is confusing:

In results: "EES at 30–90 Hz and with intensities above the threshold to evoke muscle responses initiated and maintained rhythmic electromyographic activities in the paralyzed lower limbs^{16,17}"

In discussion: "Lumbar EES at ~30 Hz induced rhythmic electromyographic activity in the paralyzed lower limbs, as observed earlier^{16–18}"

In methods: "On the assessment days, stimulation was applied with various bipolar electrode set-ups, frequencies between 5 Hz and 120 Hz"

In Fig. 5: Active electrode contacts: 1+2-, stimulation frequency: 38.5 Hz, stimulation amplitudes: 1–5 V as indicated, pulse width: 210 μ s

In Fig. 6-8: Active contacts: 0+1-, stimulation frequency: 29.4 Hz, pulse width: 210 μ s.

The algorithm for parameters selection would be very helpful. Also, except data on Fig. 1 other results collected with quite high stimulation intensity (up to 8-9V). Since other studies with EES commonly use lower intensity demonstrating efficacy in facilitation of rhythmic activity or volitional control below the injury, some justification for high amplitude would be also helpful.

Diagram with a timeline could help to illustrate the main milestones of this study.

We are grateful for the opportunity to revise our manuscript and wish to thank the reviewers for their encouraging feedback. Their insightful comments and suggestions for further data analyses have helped us to further improve the manuscript by supporting our results and sharpening our conclusions. Below, we respond point-by-point to each of the questions raised.

REVIEWER COMMENTS

Reviewer #1 (Remarks to the Author):

Minassian, Hofstoetter and colleagues characterized patterns of rhythmic motor activities in the legs of an individual with a complete thoracic spinal cord injury (SCI). Although this has been done by others (e.g. Bussel, Calancie, Nadeau), the present study performed a more detailed analysis and used electrical epidural stimulation (EES) of the lumbar cord to compare with the other different evoked patterns. There are some novel components and careful analyses like these in people with SCI can provide important insights on human CPG organization. However, the paper is not always easy to follow because some jargon is not explicitly defined and some results are presented without context. If adequately revised and better vulgarized (accessible to the non-expert), the paper could make a strong addition to the literature.

Major comments

1. Neurophysiological and clinical terms need to be clearly defined. What are the differences between spinal myoclonus, ankle clonus and muscle spasms? Do they share common CPG circuits or not?

Thank you very much for this important comment. We have prepared additional Extended Data Figures to clarify the neurophysiological and clinical terms.

In Extended Data Fig. 1, we provide an overview of the terminologies and differences between Achilles clonus, muscle spasms, and spinal myoclonus. We also highlight whether these activities are of rhythmic nature and elaborate on whether they share common CPG circuits or not.

In Extended Data Figs. 2 and 3, we provide further details on Achilles clonus and spinal myoclonus, discuss the prevailing theories of the underlying mechanisms, and compare our data with those in the literature.

Specifically, in Extended Data Fig. 2, we discuss in detail the prevailing theory that Achilles clonus relies on the stretch reflex circuitry rather than a central oscillator and list the appropriate literature. For instance, we discuss that a partial block of large-diameter afferent fibers from the calf muscles by compression of the leg with a cuff (without affecting conduction in the efferent motor fibers) abolishes the Achilles clonus. Before the Achilles clonus disappears, there is a progressive reduction in its oscillating frequency that is associated with the prolonged compression and the reduction of the conduction property of the proprioceptive fibers.

In Extended Data Fig. 3, we are placing our findings of spinal myoclonus in the context of previous descriptions in the literature and concomitantly respond to your feedback on Line 129 in the original submission (“Line 129. Why are these very rare forms of spinal myoclonus?”).

The term deletion needs to be clearly defined (see minor comments, line 59). Other terms that need definition include ‘anterior-dominant synergy’ (line 262), flexor-biased (line 262) and extensor-biased (line 263). That whole section starting on line 293 is difficult to follow.

The term deletion is now clearly defined in the introduction as well as the discussion in all passages where you had noticed inaccuracies in our description.

We have also simplified the terminology used to refer to different phases of the rhythmic activities evoked by epidural electrical stimulation and are now using “flexion-like” and “extension-like” instead consistently throughout the manuscript. Finally, we have edited the indicated section to improve readability and comprehensibility.

2. Context and appropriate references need to be provided for some results, such as the relationship between burst/interburst duration and cycle period. Explain how this relates specifically to rhythm and/or pattern generation.

Thank you very much for this comment. The main point of this section was to investigate whether burst duration or intensity were indicative of the interburst duration/silent phase (which would suggest a refractory behavior). We have now decided to exclude the additional data on the relationships between burst/interburst duration and cycle period as these were out of context, were not further discussed, and a similar analysis was not carried out for the epidural stimulation data.

3. From a neurophysiological perspective, the discussion is hard to follow, in part because the different rhythmic activities (spinal myoclonus, ankle clonus, spasms, locomotor-like activity) are not clearly defined.

Thank you very much for your feedback. Following your advice, we have prepared additional Extended Data Figures to define the terminologies and illustrate the associated electromyographic activities: Achilles clonus, Extended Data Figs. 1 and 2; spinal myoclonus, Extended Data Figs. 1 and 3; muscle spasms, Extended Data Figs. 1 and 4; and locomotor-like activity, Extended Data Fig. 7. Also, we have carefully edited parts of the discussion to increase readability and comprehensibility.

Minor comments

Line 34-35. There is some confusion here. In cats, fictive locomotion in all preparations can only be demonstrated with a neuromuscular blockade (curarization). In spinal preparations, pharmacology is required in addition to dorsal root or dorsal column stimulation. In the spinal preparations, the animal is also decerebrated. Sentences need to better describe the preparations.

Thank you very much for making us aware of these inaccuracies, which resulted from several iterations of shortening the text. The paragraph now reads as follows:

“Fictive locomotion was expressed, *inter alia*, in the immobilized (neuromuscular block by curarization), acute spinal cat pretreated with L-Dopa through electrical dorsal root or dorsal column stimulation¹, in the immobilized decerebrate cat through electrical stimulation of brainstem structures², or in the isolated neonatal mouse spinal cord through the administration of excitatory amino acids and serotonin³. Such explicit evidence is sparse in non-human primates. Attempts to elicit fictive locomotion by L-Dopa and dorsal column stimulation failed in acutely spinalized and immobilized macaque monkeys⁴. In adult marmoset monkeys, electrical stimulation of the brainstem in the decerebrate preparation or administration of different pharmacological agents (clonidine, NMDA, serotonin) following spinalization generated various rhythmic activities that presented ‘component fictive patterns’ with alternating as well as synchronous bursts in flexor and extensor nerves⁵. If combined, the component fictive patterns obtained with different experimental conditions would resemble a true fictive locomotor pattern. Yet, full fictive locomotion was not evoked by any single condition.”

Line 37. I do not think it has been demonstrated which types of receptors are present on CPG neurons because CPG neurons have not been identified.

Thank you very much for your comment. Recent molecular-genetic advances have allowed the identification of CPG-constituting spinal neuron types and the investigation of their receptor expressions. However, we understand that the sentence would be difficult to appreciate in this context. We have therefore rephrased the whole sentence: “Fictive locomotion was expressed, *inter alia*, or in the isolated neonatal mouse spinal cord through the administration of excitatory amino acids and serotonin³.” (instead of “...or in the isolated neonatal mouse spinal cord through neuropharmacological activation of receptors expressed by CPG neurons³.”)

Line 48. Study by Nadeau et al. 2010 should also be cited here.

Thank you for this comment. We have added the work of Nadeau et al. 2010.

Line 57. Change ‘existed’ to ‘exist’.

Done.

Line 59. Deletions are not interruptions of ongoing rhythmicity. They are an absence of activity, often accompanied by sustained activity in antagonist motor pools. Deletions can be resetting (change in timing of rhythmicity) or non-resetting (maintaining the timing of rhythmicity).

We apologize for these inaccuracies. We have re-phrased the paragraph:

“Assuming CPGs exist in the human spinal cord, fundamental characteristics of their operation observed in animal experiments should also be detected in humans. Notable motor phenomena during otherwise robust fictive locomotion are so-called motor deletions, reflecting spontaneous errors in CPG operation. Motor deletions are an absence of activity in a set of synergistic motor pools during a time

period when it would normally occur^{1,6}. This failure of providing rhythmic drive can be accompanied by a failure to inactivate the antagonistic set of motoneurons, resulting in their sustained firing³.”

Resetting and non-resetting types of motor deletions are mentioned later in the discussion.

Line 73. It is not clear what new theories of spinal rhythm and pattern generations are formulated in the present study.

Thank you very much for this comment. We have re-phrased the final paragraph of the introduction and now clearly state our hypotheses:

“...We observed muscle spasms, self-sustained spinal myoclonus, EES-induced rhythmic activity, and, first-in-human, spontaneous motor deletions, all in the same subject. Based on the ability to directly compare these motor phenomena, we propose that the generation of spinal myoclonus is closely linked to circuits underlying muscle spasms. Following the logic from animal studies^{3,5,7}, component locomotor patterns evoked by EES support the activation of the CPG, and the specific type of motor deletions detected here indicates a separation of rhythm generation and pattern formation in the human lumbar spinal cord with a flexor-dominant operation.”

Figure 1. It is not clear why the three rhythmic activities do not have the same length of time (15 s, 30 s and 60 s). The timescale is the same, but having the same length would help compare the frequency and characteristics of the different activities.

In Fig. 1, we chose to show different examples of myoclonus activity for the entire durations they had lasted (here, varying from 15-60 s).

To facilitate comparisons between examples, we showed them in the same time scaling as well as EMG amplitude scaling. This allows the comparison of both the EMG patterns (distribution of EMG activity across muscles per cycle) and the rhythm cycle frequencies (cycles per second). We believe that, to help compare the frequency, we need to display the different examples with the same time scale. For instance, within the duration 10 s, having the same displayed length in the figure, four, three, and two bursts can be seen in the examples (i)–(iii), respectively, representing the range of spinal myoclonus frequencies observed.

In the legend of Fig. 1, we are now stating: “EMG activities of the rhythmic phenomena are shown for the entire durations they had lasted with the same time and EMG amplitude scaling in a, b, and c.”

Lines 84-85. How are the different rhythmic activities shown in Figure 1 clearly different from ankle clonus in extended data Figure 1? All show synchronous activity in different motor pools. Here again, using the same length of time would facilitate comparisons.

Thank you very much for making us aware of this potential source of misunderstanding. We have completely revised Extended Data Fig. 1. Further clarification is provided in Extended Data Figs. 2 and 3.

Line 85. Please define 'quasi-stable'.

This term was indeed unnecessary. We have deleted it throughout the manuscript.

Line 86. ... that lasted for a minimum (of) 10 s ...

Done, thanks.

Line 103. Muscle was not a significant factor for what?

We added "for cycle frequency".

Line 129. Why are these very rare forms of spinal myoclonus? How are the forms of spinal myoclonus different from ankle clonus?

We now explain in detail why we call this observation a rare phenomenon in Extended Data Fig. 3. Briefly, different types of rhythmic behaviors occur in individuals with upper motoneuron disorders that are collectively called spinal myoclonus. They are mainly classified into spinal segmental myoclonus or propriospinal myoclonus. We discuss why neither of these types of myoclonus can adequately account for the self-sustained rhythmic electromyographic activities as described in the present study. Rather, our observations closely resemble the six cases in complete spinal cord injury independently described by Bussel, Calancie, and Nadeau. Remarkably, Prof. Blair Calancie stated in his article from 2006 that "... *The findings from 6 subjects presented herein emerged from electrophysiologic studies of many hundreds of subjects with chronic SCI ...*"⁸. Compared to the large number of spinal segmental myoclonus or propriospinal myoclonus described in the literature, the "Bussel-Calancie" type can indeed be considered a rare form.

Additionally, we elaborate on the differences between ankle clonus and spinal myoclonus in Extended Data Figs. 1, 2, and 3. Briefly, ankle clonus follows a brisk stretch of the calf muscles and is expressed as fast oscillating motion of the foot around the ankle at a cycle frequency of 3–8 Hz. The prevailing theory is that of a self-perpetuating reactivation of the stretch reflex pathway. On the other hand, the Bussel-Calancie type of spinal myoclonus is a much slower rhythmic motor behavior (0.3–0.6 Hz) with electromyographic activities showing clear waxing and waning phases with prolonged phases of inactivity.

Line 135. The term 'proprioceptive inflow' is not clear. Define 'slow' passive movements. Slow is a relative term.

Thank you for this comment. Here, we wanted to stress that the seconds-long passive movements could result in a spinal myoclonus, while no case was found following a brisk stretch. Similarly, spasms are not readily evoked by a brief muscle stretch, but rather require prolonged sensory stimulation. The sentence now reads as follows:

“In seven examples, spinal myoclonus followed lower-limb manipulations by the examiner that elicited either seconds-long proprioceptive activation (through a single cycle of passive hip-and-knee or ankle flexion-extension movement) or cutaneous activation from the foot sole (cf. Extended Data Table 2).”

The color scheme is impossible to follow for colorblind people (reviewer included). Magenta and turquoise are not colors that colorblind people can differentiate from other colors. Please use a colorblind friendly palette.

Thank you very much for this important comment and we apologize for not having been more considerate in the first place. We have added labels to the relevant parts within the figures and legends, and a differentiation between colors is no longer required to follow the illustrations.

Figure 4. The relationships between burst/interburst duration and cycle period have been studied extensively in fictive locomotor and scratch preparations in decerebrate cats (Yakovenko et al. 2005 J Neurophysiol; Frigon and Gossard 2009 J Physiol, 2010 J Neurosci). These studies show that dominance can change depending on the preparation or with descending and sensory inputs.

We agree with the reviewer. As explained in our response to your comment above, we have removed the analysis of relationships between burst/interburst duration and cycle period.

Line 226. What is the invariable pattern of myoclonus?

Thank you for noticing. We had demonstrated that the pattern of spinal myoclonus (i.e., the distribution of activity across muscles within a cycle of rhythmic activity) did not show statistical differences between the different examples. By “invariable pattern” we meant that the pattern was consistent across the identified examples of spinal myoclonus.

We have improved the sentence which now reads as follows:

“In great contrast to the pattern of spinal myoclonus that was consistent within an examination session and over a period of three months, the different examples of rhythmic activities evoked by EES displayed different patterns.”

Line 379. Invariable patterns? I do not understand how a pattern can be invariable.

We made the same mistake as above. We now state:

“..., as suggested by the consistent multi-muscle activation patterns over the different spinal myoclonus examples (see also Figure 2 in ⁸).”

Lines 380-381. How are muscle spasms not rhythmic?

Our understanding is that muscle spasms⁹ are one of the different manifestations of spinal spasticity following upper motor neuron lesions. These manifestations include increased muscle tone (resulting in velocity-sensitive, abnormal activation of muscles to an externally imposed stretch, increased resistance against movements), clonus triggered by brisk stretch (displaying oscillation frequencies of 3-8 Hz), and muscle spasms that are continuous abnormal muscle activations triggered by proprioceptive or cutaneous inputs.

Our understanding of a rhythmic behavior is that of a muscle activity that shows a series of consecutive contractions and relaxations with consistent repetition rate.

For instance, Achilles clonus is comprised of a series of fast repetitive beats with brief phases of (stretch reflex-like) EMG activities interrupted by phases of inactivity, in response to a single perturbation.

The spinal myoclonus described here is (a much slower) rhythmic behavior with EMG bursts showing waxing and waning of activity followed by phases of inactivity.

A muscle spasm, on the contrary, is a sustained activation (several seconds) of muscles in response to a single perturbation.

We now clarify the terminologies and the underlying mechanisms in Extended Data Figs. 1–4.

Lines 397-399. It is not clear how muscle spasms (rhythmic or not?) bypass the activation of a rhythm-generating network.

Recent molecular genetic advances have made the mouse spinal cord the most important model system to decipher the mammalian CPG networks for limbed locomotion^{10–12}. The CPG-constituting neurons in mice are conceptually classified into a rhythm generating layer and separate pattern formation layers. These rhythm generating layer provides the rhythmic drive to the pattern formation layers that contain neurons in left-right and flexor-extensor controlling circuits¹³. The rhythm generating network hence shapes the rhythmic locomotor behavior with frequencies consistent with the pace of slow and fast gait types. The sustained nature of muscle spasms (several seconds) would therefore not involve the rhythm generating layer.

On the other hand, the involvement of multiple muscles in spasms (e.g., stretch at one joint induces stereotyped muscle spasms in multiple non-stretched muscles, even on the other side of the body^{14,15}) suggests the integration of circuits projecting to motoneurons in different spinal cord segments and those located on both sides, i.e., neurons of the pattern formation layer.

The only strong indication for the activation of a rhythm-generating network in the present manuscript is the presence of non-resetting motor deletions in the otherwise robust rhythmic activities induced by epidural electrical stimulation.

Lines 409-410. State the several observations. Do you mean that locomotor-like activity and spinal myoclonus are generated by different CPGs or by shared networks with different mechanisms?

Thank you for this comment. We have now prepared Extended Data Fig. 14 illustrating the different elements that our data suggest to be involved in the generation of spinal myoclonus as well as epidural electrical stimulation-induced rhythmic activities. Briefly, spinal myoclonus showed resemblance to muscle spasms, but essential differences to epidural electrical stimulation (EES)-induced activities, both in terms of rhythm and pattern. Data provided no clear indication for the activity of rhythm generating elements in spinal myoclonus. Spinal myoclonus patterns did not yield left-right or flexor-extensor alternations. At the same time, the rhythmic multi-muscle patterns induced by EES hint at a flexor-biased activation of the rhythm-generating circuits, with the distribution of activity to different muscles through a downstream pattern formation layer. We have also revised the respective section in the discussion.

Line 420. What is meant by ‘characteristic’ interruptions?

We are sorry for the inaccuracy – the sentence now reads as follows:

“Regarding rhythm generation, we observed spontaneous errors in the EES-induced rhythmic patterns, which we propose to be flexor deletions.”

Lines 421-422. This is not the definition of a deletion.

This passage now reads as follows:

“Motor deletions are transient phases of absent activity in one set of motor pools during otherwise robust fictive locomotion, often accompanied by sustained activity in antagonist motor pools¹⁶.”

Line 442. Perret and Cabelguen were the first to propose the two-layered hypothesis.

Perret, C., Cabelguen, J.M., Orsal, D., 1988. Analysis of the pattern of activity in “knee flexor” motoneurons during locomotion in the cat. In: Gurfinkle, V.S., Ioffe, M.E., Massion, J., Roll, J.P. (Eds.), *Stance and Motion: Facts and Concepts*. Plenum Press, New York, pp. 133–141

Thank you very much for this comment. We are now referring to the work of Perret and Cabelguen in the discussion:

“Interestingly, the two-layer organization of the CPG was first suggested to explain the complex activation patterns of a bifunctional muscle group (posterior biceps-semitendinosus) during fictive locomotion in thalamic cats¹⁷. Afferent stimulation altered the activity of the muscle in either the flexor or extensor phase without resetting the rhythm. Here, the bifunctional rectus femoris also clearly demonstrated a double-bursting pattern during EES-induced locomotor-like activity.”

Reviewer #2 (Remarks to the Author):

The manuscript by Dr. Minassian et al. reports on findings from an individual with a thoracic SCI who displays a rare form of rhythmic spinal myoclonus and has an epidural stimulator to control spasticity. By comparing EMG patterns during myoclonus (spontaneous and evoked) and epidural electrical stimulation, the authors conclude that myoclonic and locomotor-like rhythmic patterns are generated by two distinct spinal mechanisms. This is significant because it goes against prior suggestions that myoclonus is generated by central pattern generating circuits. Therefore, it must be properly and strongly supported by data. The authors also report motor deletions during activity evoked by EES which represents the first evidence in humans and suggests a separation of rhythm in pattern, following the same logic as has been shown in cat and rodent locomotion. The majority of the paper is based on data from a single individual. This should be kept in mind but the observations (and analyses) from this individual contradict certain prior notions and lend support to others, which makes the study quite interesting.

We are grateful for this encouraging feedback and the further analyses suggested.

We agree that the proposition that spinal myoclonus might be more related to muscle spasms than locomotor-like activity needs to be supported by proper and strong data.

We have added a new first paragraph to the discussion to directly clarify the observed differences between spinal myoclonus and EES-induced rhythmic activity:

“Spinal myoclonus and EES-induced rhythmic patterns of electromyographic activity in the lower limbs in individuals with clinically complete SCI have been regarded as independent lines of indirect evidence for the existence of a CPG in humans^{8,18-22}. Here, the observation of both types of rhythmic motor activity in a single subject allowed their direct comparison and revealed important differences in their rhythm and patterns. The rhythm-cycle frequencies of spinal myoclonus observed here were concordant with previous reports, but lower than those of the EES examples. The first-in-man observation of motor deletions during otherwise robust rhythmic activity occurred only in the EES examples, but not in spinal myoclonus. Spinal myoclonus presented with consistent multi-muscle activation patterns. The patterns of EES-induced rhythmic activity yielded variability and differed from spinal myoclonus. These findings compelled us to suggest that spinal myoclonus engages a subset of pattern-formation circuits, while EES had activated both intrinsically rhythm-generating circuits as well as large parts of the pattern-formation circuits of the CPG.”

The new analyses suggested by this reviewer and the others further strengthened our suggestion that the mechanism underlying the rhythmic patterns induced by epidural stimulation exhibit further complexities that are not seen in the spinal myoclonus patterns. For instance, as noted by this reviewer in comment 2, statistical analysis confirmed that the two major patterns seen in the EES examples (synchronous bursting and locomotor-like) had distinct ranges of rhythm-cycle frequencies. We have also now summarized the contrasting rhythms and patterns generated within episodes of spinal myoclonus and those generated by epidural stimulation in the new Extended Data Figs. 8 and 9. We have additionally prepared the new Extended Data Fig. 14 that illustrates the different elements that our data suggest to be

involved in the generation of spinal myoclonus as well as epidural electrical stimulation-induced rhythmic activities.

1. The muscle pattern activation in myoclonus (with EES, except for the 5V) in Fig 5B (and some of the examples in Fig 3C) look remarkably similar to purple (phases in between TA activations) in Fig 6Cii, 7B, and 8B. Although it is clear that the EES is recruiting the TA-dominant synergy during the 'on phase', is it possible that the extensor-dominated synergy is common to both myoclonus and EES-evoked CPG-like activity? If this is the case, it does not necessarily suggest distinct networks but a common rhythmic network with elements that are additionally capable of rhythmicity when independently recruited (see Hagglund et al 2013, Proc Natl Acad Sci, 110:11589-94). If the myoclonus and extensor-phasing of the EES-evoked activity are not similar, data should be shown to support this conclusion. Although the activation patterns look very similar, the timing element may be entirely different but the EES extensor-phase activity would need to be analyzed similarly to Fig 2A and B to be able to assess.

Thank you very much for this great comment, which motivated us to add a detailed analysis. Although the muscle activation pattern of the extension-like phases during EES-induced rhythmic activity and of spinal myoclonus look similar, we found that (i) the time lags between the bursts generated across muscles during epidural stimulation, i.e., the sequence of muscle recruitment, differed statistically between examples, in contrast to the consistent "hard-wired" myoclonus patterns; and (ii) direct comparisons between the muscle activation pattern of myoclonus and EES-induced rhythmic activity were significantly different. These new results are summarized in the new Extended Data Fig. 8. We also added a new paragraph in the main text:

"The extension-like phases of the epidurally induced rhythmic activities showed some resemblance to the multi-muscle patterns of spinal myoclonus bursts, yet, the onset lags of the electromyographic bursts between muscles as well as the muscle activation patterns indeed differed statistically (Extended Data Fig. 8)."

2. Interestingly, the EES-evoked patterns look largely synchronous and, judging by examples displayed alone, it seems that alternation in any opposing muscles only occurs when evoked activity is at lower frequencies. If there is a frequency-dependent effect (frequency of motor activity during EES, not EES itself), that could be parallel to rodent work demonstrating distinct underlying left-right circuitries activated in a speed-dependent manner and related to gait.

Thank you for noticing this effect. We have performed additional analyses of the rhythm cycle frequencies encountered during EES-induced activities with a synchronous bursting or locomotor-like pattern, defined by the timing of the bursts produced in tibialis anterior and the triceps surae muscle group (in-phase or reciprocal relation). To increase the available data set for this analysis, we also considered sections in the recordings with robust rhythmicity composed for a minimum of four complete cycles, even if they had lasted for less than 10 s (seven examples each for synchronous bursts and locomotor-like activity). As rhythmic activity was weakly expressed and less stable on the left side, we confined our

analyses to the right lower limb (see also our response to your comment #5 below). Hence, we could only investigate whether there was a speed-dependent change of intralimb coordination.

Indeed, we could confirm your assumption that the rhythm-cycle frequencies of the synchronous bursting activity and of the locomotor-like activity were statistically different. The synchronous bursting examples had a higher range of rhythm-cycle frequencies. At the same time, all but one example of spinal myoclonus had lower rhythm-cycle frequencies than the EES-induced rhythmic activities. In fact, the rhythm-cycle frequencies of spinal myoclonus would be too slow for functional gait paces. These results together with statistics are shown in Extended Data Fig. 9. We have added the following paragraph to the results section:

“Notably, the rhythm-cycle frequencies of the synchronous bursting patterns were significantly higher than those of the locomotor-like patterns across the EES examples, indicating that changes of speed of rhythmic activity additionally involved changes in the coordination of muscle activity (Extended Data Fig. 9). Both EES-induced types of rhythmic activities had faster rhythm-cycle frequencies than spinal myoclonus, with only a single EES-induced example overlapping with the spinal myoclonus range.”

3. There is reference to analysis of a prior study but what analysis was carried out is not clear, beyond examples in Ext Data Fig 2. Please make clear in the text (and legends) exactly when the other study is being referred to and when the transition back to the single patient occurs.

Thank you for this feedback. Because we had observed spinal myoclonus as well as EES-induced rhythmic activities in the single participant of our study, we wondered whether the occurrence of these two types of rhythmic activities were correlated. This question prompted us to re-investigate a previous study of ours, in which ten individuals with chronic (motor-)complete SCI were investigated, all of whom had epidural electrodes implanted at the lumbar spinal cord level. At the same time, spinal spasticity (hypertonia, Achilles clonus, sensory-evoked spasms) of all of these participants were investigated with the same protocols as in the present manuscript. Seven of the participants responded with EES-induced rhythmic activities. None of the ten participants demonstrates episodes of spinal myoclonus.

We thought that this observation – that there is no interdependence between the occurrence of spinal myoclonus and epidural electrical stimulation-induced rhythmic activity – was worth mentioning in the present manuscript. This observation supports the assumption that these two motor phenomena have different origins.

We admit that the introduction of this earlier study in the middle of the results and the transition back to the single patient was confusing.

We have now moved the entire paragraph on the re-investigation of the earlier study to the end of the results, and there is no transition back to the participant of the present study thereafter.

The whole discussion deals with the data derived from the participant of the present study. We now clearly mention in the new first paragraph of the discussion that our observations discussed were

derived from a single subject. The previous study is only mentioned briefly again in the legend of the new Extended Data Fig. 14.

4. The observed deletions are quite compelling and are exciting potential confirmation of similarity of mechanism in humans and quadrupedal animals. Making the resetting vs non-resetting call is difficult. It is impressive how well these line up with $k=2$. I would agree that 4 are non-resetting but it seems to be at least 4 and possibly more are non-resetting, if the variability window of undisturbed cycles is considered.

Thank you very much for this comment. Since the demonstration of non-resetting motor deletions is of specific significance because of the direct indication of the activity of a separate rhythm generating layer, we were conservative with describing our findings. While we agree that more example could have been classified as non-resetting, the use of previously published statistical tests revealed four clear cases. The passage now reads as follows:

“Fig. 8c shows that the majority of the deletions had durations that lined up noticeably well with twice the duration of the respective mean rhythm cycles. Using previously published statistical analysis for the classification of motor deletions in mice³ and cats⁶, we found that four of the 12 examples were indeed non-resetting.”

5. In the 6 examples of deletions with bilateral hindlimb activity, what did the contralateral limb look like? This would provide additional information that can be paralleled to (or contrasted with) animal work.

Thank you very much for noticing. We had originally concentrated on the right lower limb of the subject because all the examples identified had consistent rhythmic activities across all studied muscles of this side. This was likely due to an asymmetric location of the epidural lead. In the left lower limb, rhythmic activities were inconsistently evoked within the same examples and EMG amplitudes were 1-2 orders of magnitude lower.

We have now added Extended Data Figs. 11 and 12 that address your comment and added new paragraphs to the main text (Results and Discussion). The results are indeed contrasting the findings of animal work. A possible explanation could be that the circuitry of the left side was weakly activated, but rather entrained by the CPG of the right side. This could be one explanation for the absence of activity in the left lower limb during motor deletions on the right side.

Results: “In the six available examples of bilateral lower-limb muscles recruited by EES, weaker activity was generated on the left side, with less robust rhythmicity. The bursting across contralateral muscles was highly synchronized with the ipsilateral tibialis anterior activity (Extended Data Fig. 11). During the ipsilateral flexor deletions, the activity across contralateral muscles was absent or reduced (Extended Data Fig. 12).”

Discussion: “The unilateral motor deletions in tibialis anterior not only resulted in a sustained firing of the other ipsilateral muscles, but also in periods of absent or reduced activity in the contralateral lower

limb (Extended Data Fig. 12). This behavior contrasts the findings in animal studies where motor deletions do not perturb the rhythmic activity on the contralateral side³. The predominant expression of EES-induced rhythmic activity in the right lower limb in the participant of the present study was likely a result of an asymmetric effect of the stimulation²³. A consequence could have been a lower level of engagement of the circuit elements residing within the left hemicord. The lower-amplitude activity in the left lower limb with less robust rhythmicity occurred predominantly synchronized with the right tibialis anterior bursts (Extended Data Fig. 11). The right tibialis anterior and the left lower-limb muscles may have been driven by the flexor burst generator through shared pattern-formation elements within the right hemicord. Failure in their operation would concomitantly lead to a unilateral flexor deletion as well as absent or reduced activity in the left lower-limb muscles.”

6. In some places, it escapes that this is all based on one individual so conclusions should be guarded. For example, in lines 453+, is this in general or just in this patient who has unusual pathology?

Thank you for this comment. We are now stressing at several places across the manuscript that data were derived from a single participant. Also, we have changed the title of the manuscript to: “Rare phenomena of central rhythm and pattern generation in a case of complete spinal cord injury”.

Original lines 453+ now read as follows: “The data derived from the participant of the present study hint at the activation of an asymmetric, flexor-dominant CPG organization by EES. ...”

First (new) paragraph of the discussion: “... Here, the observation of both types of rhythmic motor activity in a single subject allowed their direct comparison and revealed important differences in their rhythm and patterns. ...”

Minor

- Line 28: “present” to “represent”

Done.

- Lines 73-74: I recommend toning down “new theories”. Perhaps state what the findings suggest.

Thank you very much for your feedback. We have re-phrased the final paragraph of the introduction and avoid the phrase “new theories”.

“We observed muscle spasms, self-sustained spinal myoclonus, EES-induced rhythmic activity, and, first-in-human, spontaneous motor deletions, all in the same subject. Based on the ability to directly compare these motor phenomena, we propose that the generation of spinal myoclonus is closely linked to circuits underlying muscle spasms. Following the logic from animal studies^{3,5,7}, component locomotor patterns evoked by EES support the activation of the CPG, and the specific type of motor deletions detected here indicates a separation of rhythm generation and pattern formation in the human lumbar spinal cord with a flexor-dominant operation.”

- Lines 103 and 110: “no significant” to “not a significant”

Done.

- Fig 2A – the onset lines are difficult to see the differences between the colors.

Thank you for this feedback. We have added the example numbers to the onset lines to indicate the sequence over the three examples.

- It is not clear what the ‘rhythm’ is aligned to in Fig 7A.

Thank you very much for making us aware of this issue. The rhythm is aligned to the onsets of the bursts in the right RF. We are now stating this in the figure legend:

“...Different phases of activity are shown in relation to the on- and offsets of EMG bursts in the bifunctional rectus femoris (RF; hip-flexor, knee-extensor) identified by a thresholding technique. Two distinct phases were determined, a flexion-like (“f”) and an extension-like (“e”) phase. ...”

Reviewer #3 (Remarks to the Author):

Summary

This submission is a case study describing findings from a series of neurological examinations of an adult male who sustained a neurologically-complete traumatic spinal cord injury (SCI). The subject presented with severe spasticity, and the treatment plan included implantation of an epidural spinal electrode; it was thought that electrical stimulation over the lumbar spinal cord enlargement might ameliorate his spasticity. Over the course of multiple (n=11) clinic visits, a standardized protocol of limb manipulation and sensory inputs was applied to evaluate spasticity. Several of these visits preceded electrode implantation, while the remaining evaluations were carried out after the electrode was placed. All evaluations included continuous recordings of EMG from multiple lower limb muscles bilaterally.

It appears that a decision by the authors to publish this work was arrived at only after all evaluations were concluded. Nevertheless, I have no concerns about subject consent or related (e.g. confidentiality) issues.

What was noteworthy about this case is that the subject experienced multiple instances of rhythmic lower limb movements that – while usually emerging from common flexor- or extensor-biased spasms typical of a person with SCI in response to the sensory inputs being delivered – persisted well after cessation of the inputs, and contained elements consistent with a central pattern generator (CPG) for locomotion. Such activity has been reported previously, but must still be considered rare in humans. It's also interesting to see that – like several earlier publications – a major driver of this subject's extreme spasticity leading to these novel movement patterns seemed to be pathology in his hip, ultimately treated surgically through a complete hip replacement.

This is not just another example of severe spasticity resulting in spontaneous leg movements in a human. Rather, what makes this case study unique is how these spontaneous movements were impacted by the delivery of epidural stimulation, shifting the pattern of leg movements without stimulation from one reflecting spinal myoclonus (bilateral; all muscles contracting more or less simultaneously; high reproducibility in rate of contraction) to one more like locomotion (alternation between flexors and extensors; reciprocity between the legs) when the same sensory inputs were delivered while epidural stimulation was 'On'.

Based on the ability to contrast spontaneous leg movements with epidural stimulation turned 'on' vs 'off', the authors propose that the myoclonus-like movements are a consequence of repetitive spasms driven through one 'layer' of spinal circuitry, whereas the locomotion-like movements (seen only during epidural stimulation) reflect activity within a 'higher' level of spinal circuitry that includes more elements of the spinal cord's CPG.

Another finding that receives significant attention is the sometime occurrence of motor deletions when epidural stimulation is 'on' – brief interruptions of EMG from certain muscles during periods of rhythmic movements that otherwise show high reproducibility within and across muscles over multiple movement cycles. These deletions have been previously described in animal preparations, but according to the

authors have never – until now – been reported in humans. These deletions are taken as further evidence for the existence of CPG circuitry for locomotion in humans.

The authors do a nice job of placing their findings in the context of previous descriptions of such leg movements in humans, and in reduced animal preparations. The submission is very well written, data analysis is thorough, and accompanying Figures are both well-designed and highly complementary to the text. Despite some questions I have concerning several of the major conclusions (see below), publication of this work would advance the field by adding to the understanding of how the human spinal cord circuitry for rhythmic motor output – like locomotion – is organized. This knowledge in turn could be useful for ongoing efforts to restore locomotion in appropriate individuals with SCI through combinations of epidural stimulation and/or pharmacologic agents.

Dear Prof. Calancie, we are more than honored to receive your valuable feedback and appreciate your considerate inputs. We enjoyed integrating them into the revised manuscript.

Critique

A. Big things

I like this paper. However, I'm not entirely convinced about some of the statements made, and would ask that the authors consider the following comments, should they be given the opportunity to revise this work.

1. 'Spasms as a reflection of pattern generation.' I don't disagree that the same muscles that were recruited into a typical spasm also participated in the rhythmic movements you describe, but what about the triggering factors? In my experience virtually all spasms I see (and the examples you include in the early portions of several of your figures are consistent with my experience) have a critical factor that does not seem to be considered in your hypothesis: activation history. That is, once a spasm has been triggered and run its course, the probability that an identical trigger will lead to an identical spasm immediately after the initial spasm has ceased is virtually nil. Different subjects vary dramatically, of course, in the minimum interval before which spasms can be (re)elicited, but there is always some interval at play. In marked contrast, the rhythmic movements you describe are, by your own definition, sustained for at least 10 seconds, and may continue for a minute or more. In other descriptions from the literature these movements can persist indefinitely. So: why are some of these spasms so difficult to elicit on a continual basis, while other examples of – if I understand you correctly – what you consider to be the same spinal output recur with almost clock-like regularity? In the absence of an obvious answer, I find this conclusion of yours to be not terribly compelling.

We are very thankful for this important comment and the chance to provide a convincing line of reasoning. There are several observations in the study that need to be put together and we agree that they were scattered throughout the manuscript. We are now summarizing the dissimilarities between spinal myoclonus and EES-induced rhythmic activities in the new Extended Data Fig. 14. The dissimilarities of rhythm and pattern between spinal myoclonus and EES-induced rhythmic activities can be also better appreciated in the new Extended Data Fig. 9.

(1) Perhaps we should first ask whether spinal myoclonus and the EES-induced rhythmic activities could be generated by the same (or largely overlapping) circuits. If not, one would have to think about an alternative explanation.

Regarding rhythm generation: The ranges of the rhythm-cycle frequency of spinal myoclonus and EES-induced rhythmic activities differed significantly, and in fact barely overlap (new Extended Data Fig. 9). The rhythm-cycle frequencies of spinal myoclonus are too low to be of significance for the pace of even slow gait. Moreover, (non-resetting) motor deletions, a very strong indication for the activation of a central rhythm generator, occurred only during EES-induced rhythmicity.

Regarding pattern formation (i.e., the distribution of activity across the multiple muscles within one rhythm cycle): Muscle activation patterns differed between spinal myoclonus and the EES-induced rhythmic activities (new Extended Data Fig. 8). In addition, myoclonus patterns were strikingly consistent across examples recorded within a session or across a period of three months. Such “hard-wired” pattern would be a highly unexpected output of a CPG. In line with this observation, it seems that the spinal myoclonus patterns in Figure 2 (a), (b), and (c) in Calancie 2006 are perfectly consistent as well.

In summary, our data do not support the theory that spinal myoclonus and the EES-induced rhythmic activities were generated by the same (or largely overlapping) circuits. Hence, we need to ask which different mechanisms could contribute to their rhythm and pattern generation.

(2) The pattern of muscle spasms (and what “identical” spasms mean)

We agree that the manifestation of muscle spasms depends on the activation history. When attempting to evoke muscle spasms in close succession, the second or third attempts will result in lower levels of activity or even fail to trigger a spasm. The interesting point is, however, that the EMG pattern across the involved muscles remain consistent with each repetition (although the amplitudes can be reduced). This can be seen in the EMG recordings shown in our Figure 3a(i) and (ii), where two very similar spasm patterns are evoked (first 15 seconds, respectively). We have also added additional data from the present subject in the new Extended Data Fig. 4, further supporting this notion. The observation that an identical trigger will lead to an identical spasm (i.e., consistent pattern) – with the trigger applied every 10 seconds – has been shown in several studies by the group of Milan Dimitrijevic in the 1990ies²⁴.

(3) The rhythmicity of spinal myoclonus

The question that needs to be addressed is how the circuitry underlying muscle spasm generation could be triggered repeatedly to result in spinal myoclonus. One possibility, of course, would be the involvement of a central rhythm generator; however, we have no clear indication for its activation (see (1) above).

An alternative explanation underlying the generation of a rhythmic motor behavior could be the interplay of a sustained driving input to the spinal cord and a self-limiting mechanism of spasms that terminates the spasm and reduces the probability of the occurrence of a succeeding spasm for a specific period of time. The source of the sustained drive is the hip pathology in our subject (as also recognized by others). This sustained drive needs to provide two actions: First, by increasing the background excitability, it has to facilitate the occurrence of spasms. Second, and in response to your main concern, it has to

concomitantly reduce the minimum interval after which spasms can be (re)elicited. We discuss two observations to support such function. First, as shown by the group of Ole Kiehn in mice, muscle spasms are generated by persistent activity in intrinsic excitatory interneuron populations, while a delayed activity in inhibitory neural circuits curtail and delay the onset of the muscle spasms²⁵. Remarkably, the duration of the silence period and the spasms can be reduced by increasing sensory stimuli (*cf.*, Figure 3 – figure supplement 1 in²⁵, <https://doi.org/10.7554/eLife.23011.019>). Second, in Fig. 5 of the present manuscript, the sub-threshold EES stimulation mimics a background tonic activity. Within increasing intensity, the silent period between the initial spasm and the occurrence of spinal myoclonus decreases.

We have added the new Extended Data Fig. 5 to explain this hypothetical mechanism. Also, we have added reasoning to the discussion substantiating the similarity of spinal myoclonus and muscle spasms and make clear that our findings *may suggest* an alternative mechanism of rhythm generation.

2. Are the ‘motor deletions’ described in this subject indeed a ‘first-in-human’ report of the same? In your citation #14, Calancie et al. reported that the ongoing, rhythmic and reciprocal leg movements described in that paper were completely shut down during episodes of bladder emptying (description only; no Figure was included to illustrate), and were either shut down or heavily attenuated during neck flexion (dural stretch?) and applied plantar flexion of the toes (illustrations were provided). While the neck and toe manipulations could be argued to reflect additional proprioceptive (??) inputs, the same cannot be said for bladder emptying. Is it possible that the episodes of motor deletions you describe could also have been a consequence of bladder emptying, or some other manifestation of autonomic-related (i.e. not readily apparent to the investigators) activity within the CNS?

Thank you for this feedback. The motor deletions described in our manuscript were not characterized by an overall reduction of activity affecting all muscles equally. In line with the definition and observations in animal studies, our motor deletions were transient phases during rhythmic pattern generation with an absence of expected bursts in tibialis anterior, the only monofunctional flexor muscle we had recorded from, while all other ipsilateral muscles fired continuously. Indeed, in all animal studies known to us, a deletion in one motor pool is always accompanied by sustained or modulated activity in other motor pools. We have added Extended Data Fig. 10 to illustrate the striking similarity of our data with flexor deletions shown in mice³.

Further, spontaneous influences from unknown sources should result in deletions of arbitrary durations (e.g., duration of bladder emptying). We believe it is remarkable how well the durations of the motor deletions described here line up with $k=2$ in Fig. 8c, i.e., have about the duration of two rhythm cycles. Indeed, using the same statistics as applied in the animal work, we showed that four of the 12 identified deletions were non-resetting. It should be mentioned that our statistical approach was rather conservative and even more examples could have been non-resetting motor deletions.

Finally, we did not observe any motor deletions during the episodes of spinal myoclonus in our subject.

B. Little(r) stuff (in order of appearance)

Figure 3, caption for 'C': Muscle activation patterns during spasms and ... Pg 8, line 203: I don't like the use of 'phase-advanced' (here & elsewhere) to describe your findings. In response to a relatively discreet input (onset of limb manipulation), activity occurs at an earlier latency under conditions of higher EES voltage. Since there has not yet been any cyclic activity occurring within which the 'phase' can shift, how can there be a phase 'advance'? You're likely already sympathetic to this argument, whereby for the same concept you use the language "... highlight the decrease in delay between the cessation ..." in the legend for Figure 5 (pg 9, line 214).

We appreciate this comment and are now avoiding the term "phase-advanced" throughout the manuscript.

More on Figure 5.

- In the records adjacent to the '2V' series, there is clear activity within both the L & R RF muscles immediately after the offset of leg manipulation. Granted this activity is much smaller in magnitude than subsequent cycles, but are you OK with disregarding it completely?
- In the records adjacent to the '5V' series, the onset of activity in both RF muscles occurs well before leg manipulation has ended. This is not accurately reflected in Figure 5A (arrow placement) or 5C (horizontal line position).

Thank you very much for these comments on Fig. 5. Indeed, the on- and offsets of the EMG bursts were automatically identified using a thresholding technique. To avoid confusion, we are no stating this in the figure legend and also refer to the methods section.

Extended data Figure 2: titles for B (iii) and B (iv) should be B (i) and B (ii), for consistency - also, it seems from this Figure, and elsewhere in the records that stimulus voltage played a much greater role in influencing motor output than did stimulation rate; could you comment on this? Did you have any particular reason for manipulating stimulus rate over such a wide range?

Thanks for making us aware of the confusion in the figure titles, which have been corrected in the revised version of this figure (now: Extended Data Fig. 6).

Regarding the applied stimulation frequencies: Stimulation frequencies normally used to achieve an antispasticity effect of EES are within a range of 50-100 Hz²⁶. Yet, the individually most effective frequencies may also be lower and are identified during an extensive trial phase²⁷. In the participant of the present study, a wide range of stimulation frequencies was tested to alleviate his severe spasms, but unfortunately, EES turned out to have unsatisfying effects, irrespective of the applied parameters.

Methods; pg 30, line 717, typo in 'spams' (spasms)

Done.

Methods; pg 32, line 765: you've got cathode (should be +) and anode (should be -) mixed up

Indeed, in electrical stimulation of neural tissue including technologies such as deep brain stimulation and epidural spinal cord stimulation, the convention is anode, + (source of positive current) and cathode, – (sink of positive current). As such, a cathode is a negatively charged electrode that attracts positively charged ions (i.e., cations). Anodes are positively charged and attract negatively charged anions, being electron acceptors or sources of positive charge (oxidation reactions).

Blair Calancie

Reviewer #4 (Remarks to the Author):

This manuscript 'Spinal myoclonus, epidural electrical stimulation, and motor deletions: Rare phenomena of central pattern generation in complete spinal cord injury' is a study that investigates an individual with complete SCI with a rare form of self-sustained rhythmic spinal myoclonus in the legs as well as rhythmic activities induced by spinal cord stimulation. Several findings are interesting and represent new facts for the field of spinal cord injury and neurophysiology.

According to the website, Nat. Comm. rejects about 60-80% of all submissions without peer review based on impact, methodological advances, and interest in interdisciplinary readership, and the rest of the manuscripts expected to meet these criteria. To briefly summarize, from a methodological standpoint, all conclusions in this study were made based on analysis of electromyographic patterns and would require future validation using a combination of techniques. This study was performed on one subject and the validity of presented findings requires future study with analysis of data from several subjects. In one tested subject no specific assessment was conducted to exclude the role of residual fibers, particularly when tested with EES. Finally, it is not clear how presented in this study findings conceptually change what we already know about CPG and most importantly how they would advance the field. I also have to acknowledge that outside of criteria set by the journal, this work demonstrates important observations for the field of spinal cord neurophysiology.

Nat. Comm. expects reviewers to make their decision based on criteria for publication: quality of the data; level of support for the conclusions; potential significance of the results, and I will focus my evaluation mainly on these topics.

The quality of the data is excellent as authors using well established in their group assessment based on EMG analysis. The second criteria, the level of support for the conclusions is not adequately met, all conclusions in this work are based on one type of indirect analysis (EMG) performed just in one subject. Main claims of this work are primarily coming from discussions of previous research and new findings are only indirectly supporting these claims. The third criteria, potential significance of the results is also not evident.

Thank you very much for acknowledging the excellent quality of the data and the importance of our observations for the field of spinal cord neurophysiology. We are also grateful for your critical comments. Below, we are responding to your feedback regarding the level of support for our conclusions and the significance of our results.

Conceptualization of CPG and evidence of CPG in humans were broadly discussed.

The evidence for the existence of CPG-like circuits in humans has remained a matter of debate. Without direct evidence (to a degree that is possible in animal models), we are still in a phase when some scientists "believe" in its existence, while others are convinced of the opposite²². We believe that the present study provides a compelling, yet missing piece of evidence: the observation of motor deletions. The observation of motor deletions and of non-resetting motor deletions specifically is probably the most

important indication to-date in humans for the existence of rhythm generating networks with pacemaker function.

Regarding the reviewer's comment that the conceptualization of a human CPG has been broadly discussed, we respectfully disagree. We are not aware of any human study that has discussed the conceptual organization of the CPG. On the other hand, the existence of non-resetting and resetting motor deletions during otherwise robust rhythmic activities represents the first evidence in humans for a very specific 'conceptualization' of the CPG, i.e., one of functionally independent rhythm generator and pattern formation circuits. This suggestion follows the same logic as has been shown in cat and rodent locomotion. Our data further hint at the activation of an asymmetric, flexor-dominant CPG organization by epidural electrical stimulation, all of which represent novel observations.

Conceptual separation of CPG on pattern generator and pattern formation is largely hypothetical and wasn't specifically linked to any of identified circuitries or their specific location.

The conceptual separation of CPG circuits into rhythm generator and pattern formation layers in the mammalian spinal cord originally followed from classical electrophysiological studies in cat, based on non-resetting and resetting motor deletions. The results of these studies are broadly accepted, although the applied electrophysiological methods were mostly of indirect nature.

Today, however, research has come much further and we do have gained knowledge on specific neuron types that are either rhythm generating or pattern formatting. Indeed, recent molecular genetic advances have revolutionized the studies of the spinal control of limbed locomotion^{10,11}. These advances have allowed the categorization of neuron classes based on specific sets of transcription factors and the recording from such identified neurons. There is now direct evidence that intrinsically oscillating interneuron classes exist (e.g., Shox2 non-V2a and Hb9 neurons). They maintain rhythmic firing during motor deletions¹³. These neurons are clear candidates for a rhythm-generating kernel of the CPG. The specific location of several CPG constituting neurons in mice is known as well (e.g., Hb9 neurons are located medially in Rexed's Lamina VIII and are concentrated in the segments L1-L3). But the conceptual separation into rhythm and pattern formation layers is not a separation by anatomical locations. The circuits are rather identified by the constituting interneuron classes, their cellular properties, their connectivity, and whether they maintain rhythmic firing during motor deletions^{13,28}.

It is hard to say if presented results adding to what we already know about the CPG organization.

As commented above, we are not aware of any specific studies that have addressed the organization of the human CPG. Following the logic from animal studies, our observations, for the first time in humans, suggest a separation between rhythm generation and pattern formation and a flexor-dominant rhythm generating layer.

From presented evidence, the impact of these results on previously established concepts of CPG from animal and human studies is questionable. It is particularly unclear how these results change and advance already known concepts?

First, the observation of motor deletions in humans is not only an additional and essential piece of evidence for the existence of a human CPG *per se*, it also provides an important clue about the organization of the human CPG, as discussed above.

Second, by comparing EMG patterns of muscle spasms, spinal myoclonus and those generated by epidural electrical stimulation for the first time in the same individual expressing all these motor behaviors, we could conclude that spinal myoclonus is more related to muscle spasms than to CPG activation. This is significant because it goes against prior suggestions that spinal myoclonus is generated by central pattern generating circuits. More generally speaking, our data clearly reveal that the spinal cord harbors various mechanisms of rhythm generation and multi-muscle pattern formation.

It is also a question how this report will move the field forward, at least authors did not provide justifications or links to the future steps.

Our group is already working on the identification of specific lumbar spinal interneuron classes in human autopsy material on the basis of transcription factors. In the coming years, molecular genetic methodologies such as single cell sequencing and spatial transcriptomic will be applied to the human spinal cord. Clues to the organization of the human CPG as provided by the present study will be important to relate structure to function.

Motivated by your important comment, we have updated the last paragraph of the Discussion to better incorporated how our study will move the field forward:

“Spinal myoclonus and EES-induced rhythmic activities, both expressed in the same individual with a clinically complete SCI, revealed that the human lumbar spinal cord harbors different mechanisms to generate rhythmicity, shape bursts of activity, and coordinate them across multiple muscles. The first-in-human observation of motor deletions may be the best indirect evidence to date for the existence of dedicated rhythm generating spinal circuits. This finding together with the flexor-dominant operation, at least under EES, is adding fundamental insights into the organization of the human CPG for locomotion. Our data of component locomotor patterns (Extended Data Fig. 7) and flexor deletions (Extended Data Fig. 10) resembled findings that were derived from invasive procedures in the marmoset monkey⁵ and the isolated mouse spinal cord³ in a remarkable way. The juxtaposition of the organization of human and animal CPGs provides important context for appreciating the translational value of experimental investigations for promoting function after SCI. Likewise, our results emphasize the importance of CPGs as primary targets for cutting-edge interventions to augment neurorehabilitation outcomes following SCI^{29,30}. In the coming years, new generations of transcriptomic analysis^{31,32} will be applied to identify neuronal cell types of the CPG at the gene expression level^{13,33} in the human spinal cord. The present study and future physiological investigations will be essential to bridge the gap between knowledge of the spinal cord’s genetic architecture and conceptual models of the CPG.”

Again, these key criteria are set by Nat. Comm. Journal, and I strongly believe that this work could be a good fit for J. Neurophysiology or J. of Neurotrauma as an important case report.

We believe that once a manuscript submitted to Nature Communications is sent out for review, it was already deemed in scope by professional editors.

Another key point, that provided justifications, can't support the proposition that results of this work can be generated only on one unique subject and can't be carefully collected from other cases of combined injuries. Combined SCI and hip injuries are not rare and could be selected for comparison across several subjects and that would be critical for future assessment of the results variation.

We respectfully disagree with the reviewer. Additional pathology below the lesion, including hip pathologies, is not sufficient for the occurrence of spinal myoclonus.

Prof. Blair Calancie stated in his article from 2006 that “... *The findings from 6 subjects presented herein emerged from electrophysiologic studies of many hundreds of subjects with chronic SCI ...*”⁸. This experience suggests that we would need to screen 100 individuals with spinal cord injury to identify 1 with spinal myoclonus. Also, relevant outcomes of the present study resulted from the attempt to control the subject's spasticity with epidural stimulation. The current study advises that epidural stimulation is not appropriate for the control of this type of spinal spasticity that also underlies the emergence of spinal myoclonus. In summary, our subject presents a very rare phenomenon (the Calancie-Bussel type of spinal myoclonus, cf. Extended Data Fig. 3) and the epidural implant perhaps makes him unique, as we deem it unlikely that future cases demonstrating spinal myoclonus will undergo testing with epidural stimulation.

The key conclusions of this work, like ‘that myoclonus taps into spinal circuits generating muscle spasms rather than reflecting CPG activity as previously suggested’ represent rather focal interest and also requires future confirmation. In several places authors emphasize how specifically are discussed in manuscript findings and, unfortunately, no clear justification of importance of these findings are provided to help project these results to a wider population or translation to improve understanding and treatment strategies. The main conclusion “These findings argue strongly for the activation of neuronal networks in the human spinal cord that generate the locomotor rhythm independently from elements responsible for pattern formation” is questionable and for described above reasons I can't agree that this conclusion can be completely justified with provided in this study results.

We believe that the question of how we as humans walk and how our motor organization relates to that of quadrupedal mammals is a very basic question and as such of broad interest. Advancing the understanding of the neural control of locomotion at the level of the spinal cord of humans will stimulate translational research and help steer preclinical research in directions that could facilitate therapy development. We have now added Extended Data Figs. 7 and 10, which illustrate how strikingly similar our data on locomotor pattern generation are to those found in the marmoset monkey⁵ and our data on motor deletions are to those of flexor deletions in mice³, respectively.

In several places authors consider that subject has isolated lumbar spinal cord and further discuss that “the lumbar spinal cord below a clinically complete SCI and under EES may be the human model that most closely fulfills the criteria of an “isolated spinal cord” required for the demonstration of CPGs.”

Thank you for this important comment. Yes, we indeed wrote “*most closely fulfills the criteria of an isolated spinal cord*”. Regarding peripheral conditions, it was recently shown (with contribution of the first author of the current manuscript) that epidural stimulation – specifically with parameters as applied in the present study – largely reduces proprioceptive feedback from the legs (through antidromic collision within the electrically stimulated afferents)³⁴. This partial cancellation of feedback input largely eliminates the potential role of rhythmic peripheral feedback to have caused the rhythmic motor outputs during epidural stimulation. In addition, the supine position of the subject eliminates axial limb load and prevents hip extension – two peripheral signals that could potentially entrain rhythmic movements. Regarding the “isolation” of the lumbar spinal cord from brain control, we are responding to your comment below.

At the same time, most of the recent results demonstrate that patients with motor or motor and sensory complete SCI have residual connectivity and based on assessment and/or EES effect to facilitate volitional movements, can be considered as incomplete. From presented data it is not clear if tested subject was carefully evaluated and if he demonstrated any signs of incomplete SCI. If he was considered as anatomically complete, that would require detailed analysis of performed evaluation that should be provided. Considering the complex trauma in this subject, EES effect to facilitate volitional control could be performed on unaffected leg with implanted lead or could be done later after joint replacement with non-invasive transcutaneous stimulation or with percutaneous trial EES. Missing assessment of the role of residual connectivity, unfortunately, has an impact on the key conclusions of this study.

Thank you for this comment. We agree that subclinical, translational influence through slow conductive systems may remain undetected by today’s established and accepted assessment methods. However, for such a system to influence ongoing activity or even generate (highly unspecific) motor activity in the legs of individuals with clinically complete SCI requires *maximum* effort through the use of muscles rostral to the lesion site, such as in reinforcement maneuvers (neck-flexion against resistance, Jendrassik’s maneuver, etc.). For such attempts to manifest in one episode of measurable EMG activity, several repetitions of maximal effort are often required²⁴. As the subject of the present study was examined in a relaxed supine position with on-going monitoring of the EMG activity in the lower limbs (baseline required before each leg manipulation), the presence of a potential anatomical “incompleteness” does not impact the key conclusions drawn in the manuscript.

Outside of this critique, I must emphasize that this work is well-designed and as a case study definitely important for the field of spinal cord neurophysiology. There are no concerns that this work would represent a good case for future discussion and for targeted audience, although, it is hard to see this study as interdisciplinary with interest from a broad audience. Considering all mentioned concerns, I have to

defer the decision on this manuscript to the Editor, since the key criteria set by the journal are not met. In case of future consideration, it would be very helpful for reviewers and authors to understand justification and how that meet with preselection criteria set by journal.

Minor points:

Authors should cite relevant works at the end of the sentence: “The electromyographic patterns largely involved synchronous discharges across muscles. In spite of the lack of a locomotor pattern, it was suggested that this type of self-sustained rhythmic activity was due to a partial release of a CPG.”

Done.

From provided information it is hard to determine the exact parameters of EES and, particularly, more details on stimulation frequency would be helpful. Authors mentioned different numbers across manuscript, that is confusing:

In results: “EES at 30–90 Hz and with intensities above the threshold to evoke muscle responses initiated and maintained rhythmic electromyographic activities in the paralyzed lower limbs^{16,17}”

In discussion: “Lumbar EES at ~30 Hz induced rhythmic electromyographic activity in the paralyzed lower limbs, as observed earlier^{16–18}”

In methods: “On the assessment days, stimulation was applied with various bipolar electrode set-ups, frequencies between 5 Hz and 120 Hz”

In Fig. 5: Active electrode contacts: 1+2-, stimulation frequency: 38.5 Hz, stimulation amplitudes: 1–5 V as indicated, pulse width: 210 μ s

In Fig. 6-8: Active contacts: 0+1-, stimulation frequency: 29.4 Hz, pulse width: 210 μ s.

The algorithm for parameters selection would be very helpful. Also, except data on Fig. 1 other results collected with quite high stimulation intensity (up to 8-9V). Since other studies with EES commonly use lower intensity demonstrating efficacy in facilitation of rhythmic activity or volitional control below the injury, some justification for high amplitude would be also helpful.

Diagram with a timeline could help to illustrate the main milestones of this study.

The purpose of supplying the subject of the present study with an EES system was to control his severe spasms. Normally, stimulation frequencies of 50-100 Hz are effective to control spinal spasticity²⁶, yet, individually, they may also be at lower ranges.²⁷ Since no satisfactory control of the subject’s severe spasms was achieved at normally applied frequency ranges, additional frequencies were also tested.

The EES examples shown and analyzed in the manuscript were selected based on specific rhythmicity and multi-muscle patterns, rather than on specific stimulation parameters.

References

1. Grillner, S. & Zangger, P. On the central generation of locomotion in the low spinal cat. *Exp. brain Res.* **34**, 241–61 (1979).
2. Hultborn, H. *et al.* How do we approach the locomotor network in the mammalian spinal cord? *Ann. N. Y. Acad. Sci.* **860**, 70–82 (1998).
3. Zhong, G., Shevtsova, N. A., Rybak, I. A. & Harris-Warrick, R. M. Neuronal activity in the isolated mouse spinal cord during spontaneous deletions in fictive locomotion: insights into locomotor central pattern generator organization. *J. Physiol.* **590**, 4735–4759 (2012).
4. Eidelberg, E., Walden, J. G. & Nguyen, L. H. Locomotor control in macaque monkeys. *Brain* **104**, 647–63 (1981).
5. Fedirchuk, B., Nielsen, J., Petersen, N. & Hultborn, H. Pharmacologically evoked fictive motor patterns in the acutely spinalized marmoset monkey (*Callithrix jacchus*). *Exp. brain Res.* **122**, 351–61 (1998).
6. Lafreniere-Roula, M. & McCrea, D. A. Deletions of Rhythmic Motoneuron Activity During Fictive Locomotion and Scratch Provide Clues to the Organization of the Mammalian Central Pattern Generator. *J. Neurophysiol.* **94**, 1120–1132 (2005).
7. McCrea, D. A. & Rybak, I. A. Organization of mammalian locomotor rhythm and pattern generation. *Brain Res. Rev.* **57**, 134–46 (2008).
8. Calancie, B. Spinal myoclonus after spinal cord injury. *J. Spinal Cord Med.* **29**, 413–24 (2006).
9. Nielsen, J. B., Crone, C. & Hultborn, H. The spinal pathophysiology of spasticity—from a basic science point of view. *Acta Physiol. (Oxf)*. **189**, 171–80 (2007).
10. Goulding, M., Lanuza, G., Sapir, T. & Narayan, S. The formation of sensorimotor circuits. *Curr. Opin. Neurobiol.* **12**, 508–515 (2002).
11. Jessell, T. M. Neuronal specification in the spinal cord: inductive signals and transcriptional codes. *Nat. Rev. Genet.* **1**, 20–29 (2000).
12. Deska-Gauthier, D. & Zhang, Y. The functional diversity of spinal interneurons and locomotor control. *Curr. Opin. Physiol.* **8**, 99–108 (2019).
13. Dougherty, K. J. & Ha, N. T. The rhythm section: an update on spinal interneurons setting the beat for mammalian locomotion. *Curr. Opin. Physiol.* **8**, 84–93 (2019).
14. Schmit, B. D. & Benz, E. N. Extensor reflexes in human spinal cord injury: activation by hip proprioceptors. *Exp. Brain Res.* **145**, 520–527 (2002).
15. Wu, M., Hornby, T. G., Hilb, J. & Schmit, B. D. Extensor spasms triggered by imposed knee extension in chronic human spinal cord injury. *Exp. Brain Res.* **162**, 239–249 (2005).
16. Duysens, J. How deletions in a model could help explain deletions in the laboratory. *J. Neurophysiol.* **95**, 562–565 (2006).
17. Perret, C., Cabelguen, J. & Orsal, D. Analysis of the Pattern of Activity in ‘Knee Flexor’ Motoneurons During Locomotion in the Cat. in *Stance and Motion: Facts and Concepts* (eds. Gurfinkel, V. S., Ioffe, M. E., Massion, J. & Roll, J. P.) 133–141 (Springer US, 1988). doi:10.1007/978-1-4899-0821-6.
18. Bussel, B., Roby-Brami, A., Rémy Nérès, O. & Yakovleff, A. Evidence for a spinal stepping generator

- in man. *Paraplegia* **34**, 91–92 (1996).
19. Nadeau, S., Jacquemin, G., Fournier, C., Lamarre, Y. & Rossignol, S. Spontaneous motor rhythms of the back and legs in a patient with a complete spinal cord transection. *Neurorehabil. Neural Repair* **24**, 377–83 (2010).
 20. Dimitrijevic, M. R., Gerasimenko, Y. & Pinter, M. M. Evidence for a spinal central pattern generator in humans. *Ann. N. Y. Acad. Sci.* **860**, 360–76 (1998).
 21. Minassian, K. *et al.* Stepping-like movements in humans with complete spinal cord injury induced by epidural stimulation of the lumbar cord: electromyographic study of compound muscle action potentials. *Spinal Cord* **42**, 401–16 (2004).
 22. Minassian, K., Hofstoetter, U. S., Dzeladini, F., Guertin, P. A. & Ijspeert, A. The Human Central Pattern Generator for Locomotion: Does It Exist and Contribute to Walking? *Neuroscientist* **23**, 649–663 (2017).
 23. Danner, S. M. *et al.* Human spinal locomotor control is based on flexibly organized burst generators. *Brain* **138**, 577–588 (2015).
 24. Sherwood, A. M., Dimitrijevic, M. R. & McKay, W. B. Evidence of subclinical brain influence in clinically complete spinal cord injury: discomplete SCI. *J. Neurol. Sci.* **110**, 90–8 (1992).
 25. Bellardita, C. *et al.* Spatiotemporal correlation of spinal network dynamics underlying spasms in chronic spinalized mice. *Elife* **6**, (2017).
 26. Pinter, M. M., Gerstenbrand, F. & Dimitrijevic, M. R. Epidural electrical stimulation of posterior structures of the human lumbosacral cord: 3. Control Of spasticity. *Spinal Cord* **38**, 524–31 (2000).
 27. Hofstoetter, U. S., Freundl, B., Binder, H. & Minassian, K. Spinal Cord Stimulation as a Neuromodulatory Intervention for Altered Motor Control Following Spinal Cord Injury. in *Advanced Technologies in Rehabilitation of Gait and Balance Disorders* (eds. Sandrini, G., Homberg, V., Saltuari, L., Smania, N. & Pedrocchi, A.) 501–521 (Springer Verlag GmbH, 2018). doi:10.1007/978-3-319-72736-3_33.
 28. Gosgnach, S. *et al.* Delineating the Diversity of Spinal Interneurons in Locomotor Circuits. *J. Neurosci.* **37**, 10835–10841 (2017).
 29. Angeli, C. A. *et al.* Recovery of Over-Ground Walking after Chronic Motor Complete Spinal Cord Injury. *N. Engl. J. Med.* **379**, 1244–1250 (2018).
 30. Radhakrishna, M. *et al.* Double-blind, placebo-controlled, randomized phase I/IIa study with buspirone/levodopa/carbidopa (Spinalon™) in subjects with complete AIS A or motor-complete AIS B spinal cord injury. *Curr. Pharm. Des.* **in press**, (2016).
 31. Mathys, H. *et al.* Single-cell transcriptomic analysis of Alzheimer’s disease. *Nature* **570**, 332–337 (2019).
 32. Chen, W.-T. *et al.* Spatial Transcriptomics and In Situ Sequencing to Study Alzheimer’s Disease. *Cell* **182**, 976–991.e19 (2020).
 33. Kiehn, O. Decoding the organization of spinal circuits that control locomotion. *Nat. Rev. Neurosci.* **17**, 224–38 (2016).
 34. Formento, E. *et al.* Electrical spinal cord stimulation must preserve proprioception to enable locomotion in humans with spinal cord injury. *Nat. Neurosci.* **21**, 1728–1741 (2018).

REVIEWER COMMENTS

Reviewer #1 (Remarks to the Author):

The authors did an excellent job addressing my main concerns. I have one main comment and a few minor ones. It is much clearer, and the data are of high quality.

It should be made clear that there exists different CPGs in the spinal cord, not just the one for hindlimb locomotion. Langlet et al. (2005) showed that lesioning different segments of the lumbar cord could abolish locomotor-like activity without affecting the fast-paw shake rhythm.

Mid-lumbar segments are needed for the expression of locomotion in chronic spinal cats. Langlet C, Leblond H, Rossignol S. *J Neurophysiol*. 2005 May;93(5):2474-88.

Several studies by myself have shown that different rhythms, such as locomotor-like and scratch, have specialized mechanisms. For example:

Evidence for specialized rhythm-generating mechanisms in the adult mammalian spinal cord. Frigon A, Gossard JP. *J Neurosci*. 2010 May 19;30(20):7061-71.

Central pattern generators of the mammalian spinal cord. Frigon A. *Neuroscientist*. 2012 Feb;18(1):56-69.

Different rhythms can be mediated by distinct CPGs or shared circuits with specialized mechanisms. The fast-paw shake and the scratch rhythms are unilateral and some bursts change synergies from extensor-like to flexor like. This is highly relevant to the present study. Rhythmic muscle spasms could be generated by a different CPG, such as the analogue of the fast-paw shake in humans.

See also work by Lev-Tov in the mouse (two CPGs in the lumbosacral cord) and shared versus specialized circuitry in the turtle by Berkowitz A.

Minor comments

In the rebuttal, I find it odd that the authors state that 'Recent molecular genetic advances have made the mouse spinal cord the most important model system to decipher the mammalian CPG networks for limbed locomotion'. Each model has its advantages and disadvantages. The mouse model is indeed important but difficult to translate to human research or even larger mammals.

Line 12. Specify hindlimb locomotion because forelimb CPGs are located at cervico-thoracic levels.

Line 18. As mentioned, while the locomotor CPG generates bilateral activity, those for scratch and fast-paw shake are unilateral. Thus, left-right alternation is not a hallmark characteristic of CPGs, only the one for locomotion.

The text for Figure 1 should explain what is meant by spontaneous, cutaneous and proprioceptive-input evoked.

I congratulate the authors on a very interesting study.

Best regards,

Alain Frigon

Reviewer #2 (Remarks to the Author):

I am satisfied with the authors' responses to my comments. The revised manuscript is far more approachable and is better positioned to be appreciated by a wider audience. The authors have done an excellent job at detailing their findings, fully explaining the relevance of their findings, and

placing them in context of both prior clinical and animal work.

Reviewer #3 (Remarks to the Author):

The manuscript has undergone extensive revision, primarily in the form of providing additional supplemental data and illustrations but also through rewording of some passages, in response to the questions and suggestions of the four reviewers. The revised manuscript is clearer, and – should it be published – will be more accessible to readers lacking a deep understanding of spinal cord neurophysiology and the literature behind the CPG.

I have no more major issues, but would still like to see the following minor points addressed, for either accuracy or clarity.

Page 14, line 311: citation '13' should be included here, as we describe multiple examples of spinal myoclonus reduction/elimination following certain treatments.

Page 40, Figure 3, caption c (i.e. within the figure itself): typo for 'spasm'

Page 44, line 889: typo for 'homologous' (I think that's what you want to say, no???)

Page 46, figure 7 caption (i.e. within the figure itself): it's a little confusing to go from section 'a', to section 'c', and then finish with section 'b'. Is this necessary? Why not 'a' to 'b' to 'c'? Alternatively, why not drop these identifiers altogether, and simply point out that the period of deletions is around 11s in the record: between that time identifier along the x-axis, and the different shading of the activity in question, it should be readily apparent to the reader what you're referring to.

Page 50. Extended Data Figure 1 legend, part 3 (Bussel-Calancie spinal myoclonus). The description accurately portrays findings from the individuals reported on by Calancie (2006) with motor-complete SCI. However, it is inaccurate regarding the 2 persons with motor-incomplete SCI, in that their movements alternated between left and right sides, showed clearly evident stepping-like movements, and showed clearly evident reciprocal activity between flexors and extensors within each leg. Using your terminology, this description must be considered a higher-level manifestation of spinal circuitry contributing to the CPG, yet you've lumped these two incomplete subjects in with the other examples of 'lower-order' spinal myoclonus.

Page 51, line 965. A suggestion: "... immediately following a brief contraction (i.e. when the muscle relaxes) results in the re-afferent activation ...

Page 57, line 1037: can you rephrase to include options other than just hip pathology? e.g. "... a hip pathology or other (possibly nociceptive) signal ..."

Page 57, line 1043: 'trespassing'? I'm not sure this is the word you want here. Maybe 'exceeding', or 'triggering', or 'crossing', or ???

Page 60, line 1070: same issue with using 'homologues' (as above)

Page 71, Ext Data Figure 14. Caption typo, right side, box '3', "rhythmic"

Original critique, cathode vs anode. I honestly don't know what I was thinking. I liken that booboo to signaling a left-hand turn when you're driving, but you then turn your car to the right. Fortunately my error didn't lead to a crash. Please forgive me ...

Finally, thanks for your kind words. More important, thanks for this careful work, maybe pulling something good out of your subject's misfortune.

Blair.

Reviewer #4 (Remarks to the Author):

The authors did a great job addressing many comments and revising the manuscript, which was significantly improved after revision. However, several key weaknesses of this study still need to be covered. For obvious reasons, it is hard to consider results based on $n=1$ valid regardless of how rare the observed phenomenon is. Another concern is that with resubmitted version, most of the statements of this work primarily come from discussions of previous research. Accordingly, authors may consider reorganizing this study as a case report or a review. A line of other major and minor concerns is listed below.

Enrollment and type of the study:

As indicated in the methods, this is a retrospective study. It's, however unclear how surgical implantation for spasticity was performed. Was it performed under HDE or as a part of another clinical research study? If implantation was done under HDE, then how it was aligned with the multiple visits and EMG collections before and after implantation that apparently was a part of the research study and not a therapy. The statement that this is a retrospective study appears only at one place at the very end of the manuscript and for clarity, may need to be stated earlier.

$n=1$:

The critical limitation of this work is that this is a study with a single participant and another key limitation is that the entire analysis is performed based on only one indirect measurement (EMG). With all respect to the authors, multiple conclusions and extensive interpretations of the results from $n=1$ cannot provide a solid platform for future steps. This study indeed demonstrates a detailed analysis of the evoked EMG patterns, focusing on previously described in animals electrophysiological phenomena, and, as other reviewers also mentioned, this study has novel components. On the other hand, even EMG analysis could include a more detail assessment of evoked components providing important information on different components of the spinal circuitry. Unfortunately, further interpretation of presented results and conclusions is limited by the single participant. The repeatability of this study is rather a question, considering that it is based on a rare phenomenon. The manuscript should be either further extended, including more subjects, or presented as a case report with acknowledgment of all limitations and alternatives.

Structure of the manuscript:

Another general concern is that with all new additions, extensive discussions, and at the same time, limited experimental information from one subject, the line of arguments in this work may probably fit better with a prospective review. In the revised version discussion is disproportionately long and takes almost half of the manuscript. At the same time, along with new figures, it well covers previous findings and evaluates different mechanisms related to the human CPG and described phenomena. This type of review would be very important and timely.

The key concern is that $n=1$ could support previous findings as a case observation but not vice versa. Because of it, in its current shape, this study cites multiple findings leading to hypothetical considerations and extensive discussions, however, without clarity on results. Interpretations of previous animal works lead to new hypothetical assumptions, while it should be clearly supported or not supported by results, which is, however, hard to expect with $n=1$. With the approach taken, authors sometimes go beyond the context of their results, which is rather confusing.

The logic of this study is somehow reversed and uses analysis and interpretations of previously published results to support findings generated on one subject. Several strong statements, like "our data clearly reveal that the spinal cord harbors various mechanisms of rhythm generation and multi-muscle pattern formation," incline on the importance of the results, missing significance as a key component of importance. Authors should consider presenting their findings either as a case report and discussing all pros and cons and potential difficulties with the repeatability of this study or reorganizing it as a review. The currently taken approach may rather confuse the readers and mislead further researchers.

Although authors have changed the title by adding the word 'case', the new title doesn't refer to the 'case study' and emphasizes the 'case of complete spinal cord injury' (New title: Rare phenomena of central rhythm and pattern generation in a case of complete spinal cord injury). Also, stating in the title 'complete' SCI is misleading as besides clinical evaluation, no information

was presented regarding a complete or isolated injury.

Rare phenomenon:

The arguments regarding the difficulties of selecting the right subjects are confusing for several reasons. Compared to the cited studies performed decades ago, the current network of SCI patients involved in multiple clinical trials provides unique opportunities to select and recruit new subjects. Particularly it is critical to consider that to accept or reject the hypothesis of this study; authors use only EMG recording during patterns evoked by maneuvers or EES in clinically complete SCI subjects. This type of data can be requested and analyzed through collaboration with several centers who already generated large data sets of EMG records with and without EES.

As it was stated by the authors, this type of myoclonus is related to a specific combination of spinal cord injury and orthopedic injury. However, apparently, this specific combination is not the only requirement. What makes this combination unique is still unclear from provided background and should be elaborated in more detail. Variability in the hip injury makes this experimental design even vaguer. With the understanding that authors are demonstrating a rare phenomenon, from an experimental standpoint, it is hard to consider that the main source of the sustained drive related to the main findings is the pathology (in this case, hip injury). This means that authors entirely rely on uncontrollable experiment variables that would change dramatically depending on multiple factors and over time, even in one tested subject. This needs to be clearly addressed because built on this background argumentation significantly decreases the enthusiasm for collected data and their interpretation.

Authors expect that "sustained drive needs to provide two actions: First, by increasing the background excitability, it has to facilitate the occurrence of spasms. Second, it has to concomitantly reduce the minimum interval after which spasms can be (re)elicited." The level of excitability could be related to EES and manipulated by EES settings used during the experiment that is a critical factor to consider. Then, authors further emphasize that "in Fig. 5, the sub-threshold EES stimulation mimics a background tonic activity. Within increasing intensity, the silent period between the initial spasm and the occurrence of spinal myoclonus decreases." This suggests that the level of excitability could be provided by EES. These observations raise the question of what makes hip pathology critical and irreplaceable for this study. As multiple other not apparent factors could be involved, experimental design should be carefully evaluated, and experimental settings must include objective measurements.

Experimental setup and instrumental assessment:

Authors indicate that initiation of spinal myoclonus was performed with lower-limb manipulations by the examiner. It is unclear how these manipulations were measured and what type of instrumental assessment was involved in measuring the force and other parameters required to provide the same influence each time to induce the required response. Task performed even by a trained examiner cannot be consistent and instrumental assessment must be involved.

Assessment of the residual connectivity:

Missing assessment of the residual connectivity across the injury has a critical impact on the key conclusions of this study. This type of assessment should be considered and implemented if authors consider extending this study:

- 1) With all respect, I can't agree that "subclinical, translational influence through slow conductive systems may remain undetected by today's established methods". In fact, most of the currently known cases of clinically complete SCI clearly demonstrated discomplete injury when tested with EES and able to initiate and control motor activity below the injury. Not performing an assessment of residual connectivity significantly decreases the value of described experiments.
- 2) The statement "However, for such a system to influence ongoing activity or even generate (highly unspecific) motor activity in the legs of individuals with clinically complete SCI requires maximum effort through the use of muscles rostral to the lesion site, such as in reinforcement maneuvers (neck-flexion against resistance, Jendrassik's maneuver, etc.)" is not necessarily correct. Most of the subjects with clinically complete SCI already, after a few attempts, were able to generate volitional movement with EES without maximum efforts through the upper muscles. However, these efforts, along with Jendrassik's maneuver, may help in the diagnostics of discomplete injury even without EES. Both assessments could be performed within described experimental settings, and it is unlikely that these assessments would affect experimental designs.
- 3) It is also hard to agree with the view that residual subclinical activity would not affect the

results of this study and conclusions if they were built on these results. The following statement sounds contradictory to generated up to this time multiple data: "As the subject of the present study was examined in a relaxed supine position with ongoing monitoring of the EMG activity in the lower limbs (baseline required before each leg manipulation), the presence of a potential anatomical "discompleteness" does not impact the key conclusions drawn in the manuscript." Multiple results generated to this point indicating that the presence of anatomical/functional "discompleteness" has an impact on sublesional circuitry and, accordingly, on results generated by targeting sublesional spinal circuitry, particularly in the presence of EES.

4) It should also be considered that: (a) In described experiments, authors use EES that facilitates spinal circuitry to the level where the subclinical level of activity turns into clear supraspinal control over motor performance. (b) Even dormant supraspinal influence in discomplete SCI may impact the results generated on what authors call an "isolated spinal cord" which is, again, unlikely to be the case, considering current findings. (c) Finally, with an experimental setup and EES system, the role of supraspinal connectivity could be assessed as a part of the trial, as discussed earlier. Missing assessment of this important variable raises multiple secondary questions and concerns, critically impacting the interpretation of provided results.

Conceptual questions related to EES and CPG:

Across the manuscript, the authors state that rhythmic or locomotor patterns are evoked, triggered, or generated by EES. Multiple evidence from animal experiments suggests that the role of EES is rather complimentary in the facilitation of rhythmic activity. Authors should come with the argument that locomotor patterns are solely evoked by EES, and as they continuously use this terminology, this should be somehow clarified.

As the Rhythm generator is a key concept of this study, several questions may need to be further elaborated:

a) It is unclear what specifically indicates that rhythm is generated by circuitry providing a certain timing (rhythm generator) and not by modulation of the sensory input, EES, or other inputs through the pattern formation circuitry?

b) Pathological periodic muscles activity, i.e. slow synchronous or asynchronous tonic contractions a commonly seen in relation to the damage or dysfunction at the different levels of CNS. How these patterns would be differentiated from the CPG activity? How the influence of other parts of CNS was eliminated in this case?

c) Authors hypothesize that different frequency for two patterns suggest that they come from the different strictures, that cannot be a clear argument. There are multiple examples of wide range of frequencies coming from the circuitries located at the different levels of CNS.

It should be noticed that several statements related to the spinal myoclonus and the EES-induced rhythmic activities are using indirect and vague argumentations, i.e. "hard-wired" pattern or difference in ranges of the rhythm-cycle frequency.

Genetics studies as a background for this work:

The key argument in this study is that motor deletion suggests a separation between rhythm generation and pattern formation. Discussed relevant genetics studies, unfortunately, couldn't be matched with presented findings to provide more clarity on CPG organization. The main concern across this study is that most of the data are not driven and/or supported by presented results but rather driven by previously published observations and hypotheses. As all measures taken during this study are solely based on indirect EMG analysis, connection to the cellular and molecular mechanisms is rather hypothetical and could be a point of review rather than the original study. Even at the circuitry level, interpretation of the results of performed EMG analysis is rather a stretch. Providing solid evidence on the repeatability of presented findings appears to be necessary, and conclusions authors trying to make could be valuable if results are collected from several subjects. The following steps to target the circuitry/cellular level and, as the authors mentioned, classifying human lumbar spinal interneurons would be definitely highly appreciated by the field.

It is interesting that several arguments made by authors, i.e., how muscle spasms bypass the activation of a rhythm-generating network, are made based on interpretation of genetic results, which, however, not providing much clarity on the exact interaction of the layers and particularly how they are integrated during motor performance. Regarding the role of recent molecular genetic advances in mice model in relation to this study, several key points should be addressed:

- 1) mentioned genetic studies were able to identify the role of specific neurons, however, it doesn't mean that the role of the entire circuitry related to formation of rhythmic activity is understood.
- 2) the role of knocked out neurons is related to specific changes in the motor pattern gave us the understanding of their role when they eliminated but not the understanding of how they interact and compliment to other components of the circuitry in either normal state or after SCI.
- 3) Do we know if identified groups of neurons intrinsically oscillating or form a circuit generating repeated oscillation between two or more neurons, and how they are integrated in spinal cord anatomy?
- 4) Several animal studies indicate that circuitry conventionally called 'rhythm generator' could be located in upper lumbar segments, which if isolated by lower injury, could not generate rhythmic activity. Does this what author consider as well for human CPG? Does it mean that oscillating neurons presented only in these segments?
- 5) Authors may need to specify the classification with genetic findings in relation to rhythm generator and pattern formation layers with references to specific findings of this study, as it is not completely clear how presented findings are connected with genetics studies on mice and could address these questions. Authors also should make it clear, which clues provide by this work will be important for future identification of specific lumbar spinal interneuron classes in human.

Minor concerns:

In paragraph: "The extension-like phases of the epidurally induced rhythmic activities showed some resemblance to the multi-muscle patterns of spinal myoclonus bursts, yet, the onset lags of the electromyographic bursts between muscles as well as the muscle activation patterns indeed differed statistically (Extended Data Fig. 8)." 1) Need to define 'some resemblance' as it is relative term. 2) Not clear how author determine statistical significances with n=1. What statistical analysis indicates that data collected during several sessions in one subject could be projected to the large population supporting main conclusions of this study?

"These findings compelled us to suggest that spinal myoclonus engages a subset of pattern-formation circuits, while EES had activated both intrinsically rhythm-generating circuits as well as large parts of the pattern-formation circuits of the CPG." Authors should provide details on their hypothesis of how EES solely activates rhythm-generated circuitry and what evidence support this hypothesis?

Using the term 'pattern formation layer' in relation to biological model may not be entirely correct as no specific layer was identified for most of the species. This term is acceptable in simulation studies where this level is conceptually presented in the model.

Authors mentioned that "spinal spasticity (hypertonia, Achilles clonus, sensory-evoked spasms) of all of these participants were investigated with the same protocols as in the present manuscript. Seven of the participants responded with EES-induced rhythmic activities. None of the ten participants demonstrates episodes of spinal myoclonus." It is unclear if these participants were evaluated for this study? If yes, this information should be included in methods. Also, what type of analysis was performed to evaluate these participants? As this is a retrospective study, more information should be provided how this single participant was selected and what were the main limitations in selecting results collected from other subjects.

"Fig. 8c shows that the majority of the deletions had durations that lined up noticeably well with twice the duration of the respective mean rhythm cycles. Using previously published statistical analysis for the classification of motor deletions in mice³ and cats⁶, we found that four of the 12 examples were indeed non-resetting." The 'majority' is a relative term, needs to be clarified. Is it clear here what type of stat analysis was used?

Final remarks:

Unfortunately, this study was performed on one subject, and the validity of the presented findings is somewhat questionable. All conclusions in this study are solely based on indirect measurement, i.e., analysis of electromyographic patterns. Multiple factors mentioned in the manuscript and response to the reviewers' critique make this study hardly reproducible, and careful consideration of these results and their interpretation without overstating is critical and need improvement.

Considering that no assessment was conducted to exclude the impact of residual connectivity across the lesion, no instrumental assessment was implemented to objectively measure manipulations, and multiple other concerns mentioned above, largely decrease the enthusiasm for the results of this work. However, regardless of these limitations, feedback from other reviewers who could make a better judgment based on their extensive expertise in the field should definitely be taken as a lead.

Reviewer #1:

The authors did an excellent job addressing my main concerns. I have one main comment and a few minor ones. It is much clearer, and the data are of high quality.

Dear Prof. Frigon, Thank you very much for your insightful and encouraging comments.

It should be made clear that there exists different CPGs in the spinal cord, not just the one for hindlimb locomotion. Langlet et al. (2005) showed that lesioning different segments of the lumbar cord could abolish locomotor-like activity without affecting the fast-paw shake rhythm.

Mid-lumbar segments are needed for the expression of locomotion in chronic spinal cats. Langlet C, Leblond H, Rossignol S. J Neurophysiol. 2005 May;93(5):2474-88.

Several studies by myself have shown that different rhythms, such as locomotor-like and scratch, have specialized mechanisms. For example:

Evidence for specialized rhythm-generating mechanisms in the adult mammalian spinal cord. Frigon A, Gossard JP. J Neurosci. 2010 May 19;30(20):7061-71.

Central pattern generators of the mammalian spinal cord. Frigon A. Neuroscientist. 2012 Feb;18(1):56-69.

Different rhythms can be mediated by distinct CPGs or shared circuits with specialized mechanisms. The fast-paw shake and the scratch rhythms are unilateral and some bursts change synergies from extensor-like to flexor like. This is highly relevant to the present study. Rhythmic muscle spasms could be generated by a different CPG, such as the analogue of the fast-paw shake in humans.

See also work by Lev-Tov in the mouse (two CPGs in the lumbosacral cord) and shared versus specialized circuitry in the turtle by Berkowitz A.

Great comment. We are now specifying “locomotor CPG” throughout the manuscript and discuss that different CPGs have been demonstrated in mammals and other vertebrates. We have incorporated the findings of Dr. Langlet, of your group, as well as of Dr. Berkowitz into our discussion of possible rhythm-generating mechanisms of spinal myoclonus.

Regarding the work of Lev-Tov, we believe that our work is less related to the two CPGs in the rodent lumbar and sacral spinal cord, as they control different sets of motoneuron pools (hindlimbs vs. tail), in which case the existence of two anatomically separate CPGs is more likely.

We have added the following text to the discussion of possible rhythm-generating mechanisms of spinal myoclonus:

“Alternatively, the rhythmogenesis of spinal myoclonus and EES-induced rhythmic activity could have emerged in separate generators or through different modes of operation of spinal neural circuits. In mammals and other vertebrates, CPGs can produce distinct rhythmic behaviors involving a common set of muscles, not just locomotion, but also various forms of scratch and fast paw shake(Frigon, 2012). The different rhythms can be generated by separate lumbar neural circuits(Langlet et al., 2005), by different CPGs with largely shared rhythm generating components(Berkowitz & Hao, 2011; Hao & Berkowitz, 2017), and by specialized control mechanisms realized by reconfigurations of the rhythm-generating circuits(Frigon & Gossard, 2010). It should be noted, however, that spinal myoclonus shows little resemblance to the fast and unilaterally expressed rhythmic behaviors of scratch and paw shake.”

Minor comments

In the rebuttal, I find it odd that the authors state that ‘Recent molecular genetic advances have made the mouse spinal cord the most important model system to decipher the mammalian CPG networks for limbed locomotion’. Each model has its advantages and disadvantages. The mouse model is indeed important but difficult to translate to human research or even larger mammals.

Agreed. The often-assumed translational value of the mouse motor-control model of course has its limitation. Our statement was specifically addressing the identification and targeted manipulation of specific interneuron (sub-)types allowed by their developmentally expressed transcription factors in mice.

Line 12. Specify hindlimb locomotion because forelimb CPGs are located at cervico-thoracic levels.

Done.

Line 18. As mentioned, while the locomotor CPG generates bilateral activity, those for scratch and fast-paw shake are unilateral. Thus, left-right alternation is not a hallmark characteristic of CPGs, only the one for locomotion.

Agreed. We now specifically state “locomotor CPG” here in the abstract as well as at different points throughout the main text

The text for Figure 1 should explain what is meant by spontaneous, cutaneous and proprioceptive-input evoked.

I congratulate the authors on a very interesting study.

Thank you again very much. Your feedback has helped us a lot to further improve our manuscript, especially to improve precision and by placing it in a wider context of information gained from animal work.

Best regards, Alain Frigon

Reviewer #2:

I am satisfied with the authors' responses to my comments. The revised manuscript is far more approachable and is better positioned to be appreciated by a wider audience. The authors have done an excellent job at detailing their findings, fully explaining the relevance of their findings, and placing them in context of both prior clinical and animal work.

We are very thankful for this comment and are glad that we could satisfactorily respond to the reviewer's questions to the original manuscript version. We are very satisfied with the revised version as it will be much more accessible to readers lacking a deep understanding of pathological muscle activities related to upper motor neuron lesions (clonus, muscle spasms) as well as concepts behind the CPG.

Reviewer #3:

The manuscript has undergone extensive revision, primarily in the form of providing additional supplemental data and illustrations but also through rewording of some passages, in response to the questions and suggestions of the four reviewers. The revised manuscript is clearer, and – should it be published – will be more accessible to readers lacking a deep understanding of spinal cord neurophysiology and the literature behind the CPG.

I have no more major issues, but would still like to see the following minor points addressed, for either accuracy or clarity.

Page 14, line 311: citation '13' should be included here, as we describe multiple examples of spinal myoclonus reduction/elimination following certain treatments.

We have added citation 13.

Page 40, Figure 3, caption c (i.e. within the figure itself): typo for 'spasm'

Thank you so much for your thorough review, we corrected the typo.

Page 44, line 889: typo for 'homologous' (I think that's what you want to say, no???)

Yes, indeed, this was a typo – thanks for noticing.

Page 46, figure 7 caption (i.e. within the figure itself): it's a little confusing to go from section 'a', to section 'c', and then finish with section 'b'. Is this necessary? Why not 'a' to 'b' to 'c'? Alternatively, why not drop these identifiers altogether, and simply point out that the period of deletions is around 11s in the record: between that time identifier along the x-axis, and the different shading of the activity in question, it should be readily apparent to the reader what you're referring to.

We agree and have changed the figure and hope it is clearer now.

Page 50. Extended Data Figure 1 legend, part 3 (Bussel-Calancie spinal myoclonus). The description accurately portrays findings from the individuals reported on by Calancie (2006) with motor-complete SCI. However, it is inaccurate regarding the 2 persons with motor-incomplete SCI, in that their movements alternated between left and right sides, showed clearly evident stepping-like movements, and showed clearly evident reciprocal activity between flexors and extensors within each leg. Using your terminology, this description must be considered a higher-level manifestation of spinal circuitry contributing to the CPG, yet you've lumped these two incomplete subjects in with the other examples of 'lower-order' spinal myoclonus.

Sorry for this inaccuracy. We have now (i) added in this figure and throughout the text “Bussel-Calancie type of spinal myoclonus in **complete SCI** ...” when referring to the synchronous myoclonus pattern and (ii) have added the following text to the legend of Extended Data Figure 1 for further clarification:

“In the two individuals with motor-incomplete SCI, the rhythmic activity alternated between left and right sides and resulted in involuntary stepping-like movements in the supine position. Such pattern of spinal myoclonus suggests a higher-level manifestation of the spinal circuitry underlying rhythmic activity compared to the circuitry generating spinal myoclonus in individuals with complete SCI, including the subject of the present study.”

Page 51, line 965. A suggestion: “... immediately following a brief contraction (i.e. when the muscle relaxes) results in the re-afferent activation ...

Thank you very much for this suggestion, which helps to further clarify our point.

Page 57, line 1037: can you rephrase to include options other than just hip pathology? e.g. “... a hip pathology or other (possibly nociceptive) signal ...”

Done.

Page 57, line 1043: ‘trespassing’? I’m not sure this is the word you want here. Maybe ‘exceeding’, or ‘triggering’, or ‘crossing’, or ???

Thank you very much for noticing. We have replaced “trespassing” by “exceeding”.

Page 60, line 1070: same issue with using ‘homologues’ (as above)

Sorry - done.

Page 71, Ext Data Figure 14. Caption typo, right side, box ‘3’, “rhythmic”

Typo corrected.

Original critique, cathode vs anode. I honestly don’t know what I was thinking. I liken that booboo to signaling a lefthand turn when you’re driving, but you then turn your car to the right. Fortunately my error didn’t lead to a crash. Please forgive me ...

Finally, thanks for your kind words. More important, thanks for this careful work, maybe pulling something good out of your subject’s misfortune.

Thank you very much – your approval of our manuscript means a lot to us.

Blair.

Reviewer #4:

The authors did a great job addressing many comments and revising the manuscript, which was significantly improved after revision. However, several key weaknesses of this study still need to be covered. For obvious reasons, it is hard to consider results based on $n=1$ valid regardless of how rare the observed phenomenon is. Another concern is that with resubmitted version, most of the statements of this work primarily come from discussions of previous research. Accordingly, authors may consider reorganizing this study as a case report or a review. A line of other major and minor concerns is listed below.

We would like to thank the reviewer for reading our revised manuscript and acknowledging the significant improvements made. We, respectfully, want to clarify the following:

The reviewer again criticizes the validity of our results because of the $n=1$. We like to make clear that we have observed:

- 1) Spinal myoclonus, which we clearly related to previous observations from another six subjects in previous literature (Bussel et al., 1988; Calancie, 2006; Nadeau et al., 2010);
- 2) Epidural electrical stimulation induced rhythmic activities, which are clearly related to previous observations from other subjects in our previous studies (Danner et al., 2015) and earlier studies by others (Dimitrijevic et al., 1998); and
- 3) The phenomenon of motor deletions, which we clearly identified following the very same logic as in previous animal studies (Zhong et al., 2012).

Therefore, each of the motor phenomena described here have been previously reported in independent studies in humans (spinal myoclonus and EES-induced rhythmic activity) or in animals (motor deletions). The value of this report is the first-in-man observation (motor deletions) and the fact that all these various events were collectively seen in a single subject – allowing direct comparisons and interpretations for the first time.

This means that none of our results come by surprise, or are questionable observations. The reviewer's comment on the validity of our results is incomprehensible and indeed unacceptable for us.

Please note that two of the reviewers of the present manuscript identified themselves as Prof. Blair Calancie – the most qualified person to assess the validity of our results on spinal myoclonus and their interpretations –, and as Prof. Alain Frigon (trained by Serge Rossignol) to assess the validity of all CPG-related results. The third reviewer was also definitely a senior and expert in the field of spinal locomotor control, as reflected by his in-depth knowledge of the classical literature.

Increasing the n – even if it were possible – would neither refute our results nor our interpretations, but reveal a range of expected biological and physiological variability, a range that can be deduced from the $n = 7$ from our own work in Danner et al., 2015 (EES-induced rhythmic activity in complete SCI) and the $n = 4$ in Calancie et al., 2006 (spinal myoclonus in complete SCI).

As stated above, the $n = 1$ is the very strength of the present study, the fact that all these motor phenomena were observed in a single subject allowed direct comparisons and interpretations. For instance, the patterns of spinal myoclonus in the motor-complete SCI subjects in Bussel et al., 1988 and Calancie et al., 2006 were in-phase bursting, very much like in the subject of our study. This observation would directly question the locomotor capability of the human spinal cord (lack of reciprocal and left-right alterations). Here, we show that the same spinal cord has indeed also a locomotor capability (as revealed by EES), which was not reflected by the spinal myoclonus activity.

As for increasing the n, it is indeed ethically questionable to plan the implantation of epidural leads in individuals already suffering from spinal myoclonus out of scientific curiosity – knowing that EES was ineffective to control spinal myoclonus.

According to the editorial request, we framed the study as a case report.

Enrollment and type of the study:

As indicated in the methods, this is a retrospective study. It's, however unclear how surgical implantation for spasticity was performed. Was it performed under HDE or as a part of another clinical research study? If implantation was done under HDE, then how it was aligned with the multiple visits and EMG collections before and after implantation that apparently was a part of the research study and not a therapy. The statement that this is a retrospective study appears only at one place at the very end of the manuscript and for clarity, may need to be stated earlier.

All required forms for the interventions described here, which we had shared with the journal upon initial submission, were approved by the handling editors.

n=1:

The critical limitation of this work is that this is a study with a single participant and another key limitation is that the entire analysis is performed based on only one indirect measurement (EMG). With all respect to the authors, multiple conclusions and extensive interpretations of the results from n=1 cannot provide a solid platform for future steps. This study indeed demonstrates a detailed analysis of the evoked EMG patterns, focusing on previously described in animals electrophysiological phenomena, and, as other reviewers also mentioned, this study has novel components. On the other hand, even EMG analysis could include a more detail assessment of evoked components providing important information on different components of the spinal circuitry. Unfortunately, further interpretation of presented results and conclusions is limited by the single participant. The repeatability of this study is rather a question, considering that it is based on a rare phenomenon. The manuscript should be either further extended, including more subjects, or presented as a case report with acknowledgment of all limitations and alternatives.

The reviewer again criticizes the indirect assessment approach using surface electromyographic recordings. Which direct assessments is the reviewer thinking about that could be applied in humans for the purpose of the study? Even current functional imaging techniques must be considered as indirect and do not have the time resolution to map rhythmic activities as those described here. Electromyographic recordings can be clearly used for the detection of rhythmicity generated in the spinal cord and expressed through the respectively innervated myotomes as the best alternative to electroneurographic recordings from anterior roots, e.g., see their direct comparison in Frigon, 2012 (Fig. 1). All recent high-impact studies of EES in individuals with SCI used surface EMG to study generated motor outputs (New England Journal of Medicine, Lancet, Nature Medicine, etc, by groups such as of Prof. Susan Harkema and Prof. Gregoire Courtine). All earlier studies of spinal myoclonus used surface EMG.

As for the repeated critique of n=1, see our responses above.

As for the questioned repeatability of the study, see our comments below.

Structure of the manuscript:

Another general concern is that with all new additions, extensive discussions, and at the same time, limited experimental information from one subject, the line of arguments in this work may probably fit better with a prospective review. In the revised version discussion is disproportionately long and takes almost half of the manuscript. At the same time, along with new figures, it well covers previous findings and evaluates different mechanisms related to the human CPG and described phenomena. This type of review would be very important and timely. The key concern is that n=1 could support previous findings as a case observation but not vice versa. Because of it, in its current shape, this study cites multiple findings leading to hypothetical considerations and extensive discussions, however, without clarity on results. Interpretations of previous animal works lead to new hypothetical assumptions, while it should be clearly supported or not supported by results, which is, however, hard to expect with n=1. With the approach taken, authors sometimes go beyond the context of their results, which is rather confusing. The logic of this study is somehow reversed and uses analysis and interpretations of previously published results to support findings generated on one subject. Several strong statements, like “our data clearly reveal that the spinal cord harbors various mechanisms of rhythm generation and multi-muscle pattern formation,” incline on the importance of the results, missing significance as a key component of importance. Authors should consider presenting their findings either as a case report and discussing all pros and cons and potential difficulties with the repeatability of this study or reorganizing it as a review. The currently taken approach may rather confuse the readers and mislead further researchers. Although authors have changed the title by adding the word ‘case’, the new title doesn’t refer to the ‘case study’ and emphasizes the ‘case of complete spinal cord injury’ (New title: Rare phenomena of central rhythm and pattern generation in a case of complete spinal cord injury). Also, stating in the title ‘complete’ SCI is misleading as besides clinical evaluation, no information was presented regarding a complete or isolated injury.

The reviewer criticizes the “clarity on results” and a “reversed logic” of this study, by which our conclusions would not base on our own results but on interpretations of previous experimental animal results. We have clearly shown that both rhythms and patterns of spinal myoclonus and epidural electrical stimulation induced activity are distinctly different within the same subject. Rhythm-cycle frequencies of spinal myoclonus were lower. Motor deletions were only found in the EES-examples. Spinal myoclonus did not show variability in the muscle recruitment pattern across the various assessments conducted over a period of several months, whereas epidural electrical stimulation induced activity showed different patterns with higher variability, including major components of locomotor activity. These are clear-cut results that allow the assumption that “our data clearly reveal that the spinal cord harbors various mechanisms of rhythm generation and multi-muscle pattern formation”.

Rare phenomenon:

The arguments regarding the difficulties of selecting the right subjects are confusing for several reasons. Compared to the cited studies performed decades ago, the current network of SCI patients involved in multiple clinical trials provides unique opportunities to select and recruit new subjects. Particularly it is critical to consider that to accept or reject the hypothesis of this study; authors use only EMG recording during patterns evoked by maneuvers or EES in clinically complete SCI subjects. This type of data can be requested and analyzed through collaboration with several centers who already generated large data sets of EMG records with and without EES. As it was stated by the authors, this type of myoclonus is related to a specific combination of spinal cord injury and orthopedic injury. However, apparently, this specific combination is not the only requirement. What makes this combination unique is still unclear from provided background and should be elaborated in more detail.

Variability in the hip injury makes this experimental design even vaguer. With the understanding that authors are demonstrating a rare phenomenon, from an experimental standpoint, it is hard to consider that the main source of the sustained drive related to the main findings is the pathology (in this case, hip injury). This means that authors entirely rely on uncontrollable experiment variables that would change dramatically depending on multiple factors and over time, even in one tested subject. This needs to be clearly addressed because built on this background argumentation significantly decreases the enthusiasm for collected data and their interpretation. Authors expect that “sustained drive needs to provide two actions: First, by increasing the background excitability, it has to facilitate the occurrence of spasms. Second, it has to concomitantly reduce the minimum interval after which spasms can be (re)elicited.” The level of excitability could be related to EES and manipulated by EES settings used during the experiment that is a critical factor to consider. Then, authors further emphasize that “in Fig. 5, the subthreshold EES stimulation mimics a background tonic activity. Within increasing intensity, the silent period between the initial spasm and the occurrence of spinal myoclonus decreases.” This suggests that the level of excitability could be provided by EES. These observations raise the question of what makes hip pathology critical and irreplaceable for this study. As multiple other not apparent factors could be involved, experimental design should be carefully evaluated, and experimental settings must include objective measurements.

Regarding the comments: “Variability in the hip injury makes this experimental design even vaguer” and “This means that authors entirely rely on uncontrollable experiment variables”: This clearly must be a misunderstanding as we did not aim to provoke spinal myoclonus by any type of controlled experimental designs. Spinal myoclonus was expressed either spontaneously or during clinical examinations of spasticity and was only identified as such during and following the examinations. The original major complaint of the patient was intractable severe muscle spasms.

The importance of the hip pathology can be clearly deduced from the fact that short-lasting analgesia of the left hip reduced the repetitive bursting of spinal myoclonus, but not the muscle spasms. Long-term control was only achieved after a total hip replacement surgery, as stated in the methods. In previous studies of spinal myoclonus, three patients had concomitant hip pathologies as well and their direct treatment suppressed spinal myoclonus. On the other hand, no single intervention including pharmacotherapy and epidural electrical stimulation had any satisfactory effect on spinal myoclonus in the present subject. Again– in the mentioned previous studies of spinal myoclonus – standard anti-spasticity medication had also failed to suppress the spinal myoclonus episodes.

Experimental setup and instrumental assessment:

Authors indicate that initiation of spinal myoclonus was performed with lower-limb manipulations by the examiner. It is unclear how these manipulations were measured and what type of instrumental assessment was involved in measuring the force and other parameters required to provide the same influence each time to induce the required response. Task performed even by a trained examiner cannot be consistent and instrumental assessment must be involved.

This is a repeated misunderstanding by the reviewer. The clinical examinations did not aim to provoke spinal myoclonus. The lower-limb manipulations by the examiner were part of a previously described protocol to assess different clinical manifestations of spasticity. In fact, we identified only eleven examples of spinal myoclonus (that lasted for a minimum of 10 s) throughout the multiple clinical assessments before and after the implantation of the epidural lead. When they did occur, no deviations from the clinical protocol were made in the attempt to provoke further myoclonus episodes.

Assessment of the residual connectivity:

Missing assessment of the residual connectivity across the injury has a critical impact on the key conclusions of this study. This type of assessment should be considered and implemented if authors consider extending this study:

1) With all respect, I can't agree that "subclinical, translational influence through slow conductive systems may remain undetected by today's established methods". In fact, most of the currently known cases of clinically complete SCI clearly demonstrated discomplete injury when tested with EES and able to initiate and control motor activity below the injury. Not performing an assessment of residual connectivity significantly decreases the value of described experiments.

2) The statement "However, for such a system to influence ongoing activity or even generate (highly unspecific) motor activity in the legs of individuals with clinically complete SCI requires maximum effort through the use of muscles rostral to the lesion site, such as in reinforcement maneuvers (neck-flexion against resistance, Jendrassik's maneuver, etc.)" is not necessarily correct. Most of the subjects with clinically complete SCI already, after a few attempts, were able to generate volitional movement with EES without maximum efforts through the upper muscles. However, these efforts, along with Jendrassik's maneuver, may help in the diagnostics of discomplete injury even without EES. Both assessments could be performed within described experimental settings, and it is unlikely that these assessments would affect experimental designs.

3) It is also hard to agree with the view that residual subclinical activity would not affect the results of this study and conclusions if they were built on these results. The following statement sounds contradictory to generated up to this time multiple data: "As the subject of the present study was examined in a relaxed supine position with ongoing monitoring of the EMG activity in the lower limbs (baseline required before each leg manipulation), the presence of a potential anatomical "discompleteness" does not impact the key conclusions drawn in the manuscript." Multiple results generated to this point indicating that the presence of anatomical/functional "discompleteness" has an impact on sublesional circuitry and, accordingly, on results generated by targeting sublesional spinal circuitry, particularly in the presence of EES.

4) It should also be considered that: (a) In described experiments, authors use EES that facilitates spinal circuitry to the level where the subclinical level of activity turns into clear supraspinal control over motor performance. (b) Even dormant supraspinal influence in discomplete SCI may impact the results generated on what authors call an "isolated spinal cord" which is, again, unlikely to be the case, considering current findings. (c) Finally, with an experimental setup and EES system, the role of supraspinal connectivity could be assessed as a part of the trial, as discussed earlier. Missing assessment of this important variable raises multiple secondary questions and concerns, critically impacting the interpretation of provided results.

The reviewer again states his opinion that residual supraspinal connectivity across the injury would have a critical impact on the key conclusions of this study, while not providing any clues on how they would provoke or influence the observed motor phenomena. As throughout his critique, no references are given to support any of his strong opinions:

"Multiple results generated to this point indicating that the presence of anatomical/functional "discompleteness" has an impact on sublesional circuitry ...". We would appreciate if the reviewer could provide one single reference where an impact of "discompleteness" was shown in a subject lying relaxed (no volitional attempt to execute a motor task) in the supine position.

“Most of the subjects with clinically complete SCI already, after a few attempts, were able to generate volitional movement with EES without maximum efforts through the upper muscles.” We have witnessed recordings at Prof. Harkema’s lab and have directly worked with Prof. Courtine’s group for several years. Together, these two groups cover most of the SCI subjects with EES systems. We definitely disagree with the reviewer that “most of the subjects” were able to produce gross movements acutely without maximum Jendrassik-like volitional effort while EES was provided. Anyway, there was no attempt to produce any movements in the present study.

Regarding the use of EES to “facilitate spinal circuitry to the level where the subclinical level of activity turns into clear supraspinal control over motor performance”:

- (i) The eleven episodes of spinal myoclonus which were the focus of the first part of the manuscript occurred without the presence of EES.
- (ii) Only in Figure 5, EES was applied at sub-motor threshold level, yet in the absence of any attempted motor task.
- (iii) In all of the EES induced examples of rhythmic activity, stimulation was applied clearly above the muscle-activation threshold level, again in the absence of any attempted motor task.

None of these conditions are comparable to the “motor enabling” effects of EES the reviewer is likely referring to (“clear supraspinal control”), which require training or at least instruction as well as the intention of the person treated to produce activity.

Finally, there is a very nice demonstration in an individual with a clinically motor-incomplete SCI (with much more preserved descending control than in case of a “discomplete” injury) who could not replicate the pattern of spinal myoclonus with voluntary effort, cf. Fig. 4 in (Calancie, 2006).

Conceptual questions related to EES and CPG:

Across the manuscript, the authors state that rhythmic or locomotor patterns are evoked, triggered, or generated by EES. Multiple evidence from animal experiments suggests that the role of EES is rather complimentary in the facilitation of rhythmic activity. Authors should come with the argument that locomotor patterns are solely evoked by EES, and as they continuously use this terminology, this should be somehow clarified.

As the Rhythm generator is a key concept of this study, several questions may need to be further elaborated: a) It is unclear what specifically indicates that rhythm is generated by circuitry providing a certain timing (rhythm generator) and not by modulation of the sensory input, EES, or other inputs through the pattern formation circuitry? b) Pathological periodic muscles activity, i.e. slow synchronous or asynchronous tonic contractions a commonly seen in relation to the damage or dysfunction at the different levels of CNS. How these patterns would be differentiated from the CPG activity? How the influence of other parts of CNS was eliminated in this case? c) Authors hypothesize that different frequency for two patterns suggest that they come from the different strictures, that cannot be a clear argument. There are multiple examples of wide range of frequencies coming from the circuitries located at the different levels of CNS. It should be noticed that several statements related to the spinal myoclonus and the EES-induced rhythmic activities are using indirect and vague argumentations, i.e. “hard-wired” pattern or difference in ranges of the rhythm-cycle frequency

“Multiple evidence from animal experiments suggests that the role of EES is rather complimentary in the facilitation of rhythmic activity”. Again, we kindly ask the reviewer to provide a single reference of the “multiple evidence”. We can only assume that the reviewer thinks about studies

in spinal rats, that are walking with body weight support on a treadmill (proprioceptive feedback from stepping legs) under strong pharmacological (serotonergic and or dopaminergic) activators in combination with EES. No activity-enhancing drugs, nor load-related proprioceptive feedback were provided in our participant, rendering EES to the only dominating source of stimulation when turned on. It is also important to mention that the EES-induced activity immediately ceased when stopping the stimulation.

In each of the examples of EES-induced activities, the stimulation intensity was above the threshold to evoke spinal reflex responses in the lower limb muscles. We and others have repeatedly shown that EES-induced rhythmic activities in individuals with motor complete SCI result in very specific EMG signals: Each stimulation pulse within an ongoing train of stimuli triggers mono/oligosynaptic spinal reflexes, which can be clearly distinguished within the rhythmic bursts that they form (Danner et al., 2015; Gilge et al., 2004; Minassian et al., 2004, 2017).

Clear indications that the rhythm of EES-induced activities is provided by a rhythm generating circuitry in the paper are:

- Maintained rhythm cycle frequencies following motor deletion, i.e., even following seconds of no rhythmic activity and no movements in the lower limbs;
- Even more importantly, the occurrence of non-resetting motor deletions, which means that rhythmicity resumes without a phase shift after the deletion. In the literature, non-resetting motor deletions in non-mammalian vertebrates as well as rodents and cats are firmly accepted indicators that in spite of a phase of no rhythmic motor output, the information of rhythm frequency and phase must have been preserved by an ongoing rhythmically active part of the circuitry, i.e., the rhythm generating layer.

Regarding the role of phasic proprioceptive feedback in shaping the rhythmic bursts, major sources of phase transitions are load receptors detecting ground reaction forces and muscle spindle afferents detecting stretch at the hip joint. Signaling from both sources was already minimized in the supine position in which spinal myoclonus and EES-induced rhythmic activities were detected in the present subject. On top of that, tonic lumbar EES with the stimulation intensities and frequencies used here largely cancels phasic proprioceptive feedback from the lower limbs to the spinal cord as recently demonstrated (Formento et al., 2018).

We described the timing of muscle recruitment within the episodes of spinal myoclonus as quasi “hard-wired” (in quotation marks), because we had statistically demonstrated that they did not differ across spinal myoclonus examples found over a period of three months. This remarkably consistent pattern is hardly an “indirect and vague argumentation” that the underlying circuitry was not flexible as would be expected (and indeed required) from a CPG for locomotion.

All of these points were already discussed in detail in the submitted manuscript.

Genetics studies as a background for this work:

The key argument in this study is that motor deletion suggests a separation between rhythm generation and pattern formation. Discussed relevant genetics studies, unfortunately, couldn't be matched with presented findings to provide more clarity on CPG organization. The main concern across this study is that most of the data are not driven and/or supported by presented results but rather driven by previously published observations and hypotheses. As all measures taken during this study are solely based on indirect EMG analysis, connection to the cellular and molecular mechanisms is rather hypothetical and could be a point of review rather than the original study. Even at the circuitry

level, interpretation of the results of performed EMG analysis is rather a stretch. Providing solid evidence on the repeatability of presented findings appears to be necessary, and conclusions authors trying to make could be valuable if results are collected from several subjects. The following steps to target the circuitry/cellular level and, as the authors mentioned, classifying human lumbar spinal interneurons would be definitely highly appreciated by the field. It is interesting that several arguments made by authors, i.e., how muscle spasms bypass the activation of a rhythm-generating network, are made based on interpretation of genetic results, which, however, not providing much clarity on the exact interaction of the layers and particularly how they are integrated during motor performance. Regarding the role of recent molecular genetic advances in mice model in relation to this study, several key points should be addressed: 1) mentioned genetic studies were able to identify the role of specific neurons, however, it doesn't mean that the role of the entire circuitry related to formation of rhythmic activity is understood. 2) the role of knocked out neurons is related to specific changes in the motor pattern gave us the understanding of their role when they eliminated but not the understanding of how they interact and compliment to other components of the circuitry in either normal state or after SCI. 3) Do we know if identified groups of neurons intrinsically oscillating or form a circuit generating repeated oscillation between two or more neurons, and how they are integrated in spinal cord anatomy? 4) Several animal studies indicate that circuitry conventionally called 'rhythm generator' could be located in upper lumbar segments, which if isolated by lower injury, could not generate rhythmic activity. Does this what author consider as well for human CPG? Does it mean that oscillating neurons presented only in these segments? 5) Authors may need to specify the classification with genetic findings in relation to rhythm generator and pattern formation layers with references to specific findings of this study, as it is not completely clear how presented findings are connected with genetics studies on mice and could address these questions. Authors also should make it clear, which clues provide by this work will be important for future identification of specific lumbar spinal interneuron classes in human.

Motor deletions, including non-resetting ones, are clearly shown in our results and are directly compared to those recorded from the isolated mouse spinal cord, cf. Extended Data Fig. 10 of our work and (Zhong et al., 2012). The interpretation that these motor deletions suggest a separation into rhythm generation and pattern formation follows the exact same logic as in electrophysiological studies by multiple international teams in mammals and other vertebrates. It is hence unclear how the reviewer believes that we drew "hypothetic assumptions" driven by "genetic" studies. These genetic studies were solely discussed in our responses to this reviewer, but do not appear in the manuscript.

Unfortunately, the reviewer continues to make strong and unfavorable statements – which are simply wrong. For instance, "It is interesting that several arguments made by authors, i.e., how muscle spasms bypass the activation of a rhythm-generating network, are made based on interpretation of genetic results".

Again, we are not using interpretations derived from genetic studies to discuss the mechanisms underlying muscle spasms. Rather, we have defined muscle spasms and their underlying mechanisms in Extended Data Fig. 1 and the discussion section. We mention the most current theories, including (i) decreased inhibition of sensory transmission, (ii) increased expression of plateau potentials in motoneurons, (iii) contribution of spinal interneuron circuits, etc. Muscle spasms are continuous muscle activities that typically outlast the input that provoked them, but they are not recurring or rhythmic bursts – the involvement of a rhythm-generating network is not necessary and never mentioned in the literature.

Minor concerns:

In paragraph: “The extension-like phases of the epidurally induced rhythmic activities showed some resemblance to the multi-muscle patterns of spinal myoclonus bursts, yet, the onset lags of the electromyographic bursts between muscles as well as the muscle activation patterns indeed differed statistically (Extended Data Fig. 8).” 1) Need to define ‘some resemblance’ as it is relative term. 2) Not clear how author determine statistical significances with n=1. What statistical analysis indicates that data collected during several sessions in one subject could be projected to the large population supporting main conclusions of this study?

Regarding 1) Thank you for this comment. The section now reads as follows:

“The extension- as well as the flexion-like phases of the epidurally induced rhythmic activities differed from the multi-muscle patterns of spinal myoclonus bursts both in terms of the onset lags of the electromyographic bursts between muscles as well as of the muscle activation patterns (Extended Data Fig. 8).”

Regarding 2) It is of course possible to apply statistical analysis to multiple data derived from a single subject. The n then denotes the number of observations and not the size of the subject cohort.

“These findings compelled us to suggest that spinal myoclonus engages a subset of pattern-formation circuits, while EES had activated both intrinsically rhythm-generating circuits as well as large parts of the pattern-formation circuits of the CPG.” Authors should provide details on their hypothesis of how EES solely activates rhythm-generated circuitry and what evidence support this hypothesis?

We have elaborated on these differences in detail in the discussion and in the Extended Data Figures 9 and 14. Briefly, spinal myoclonus does not present motor deletions, while EES-induced rhythmic activities do, and rhythm cycle frequencies of spinal myoclonus are too slow for functional gait paces and are below those of the EES examples.

Following a feedback of one of the other reviewers, we have now added the following text to the discussion of possible mechanisms of rhythm generation in spinal myoclonus:

“Alternatively, the rhythmogenesis of spinal myoclonus and EES-induced rhythmic activity could have emerged in separate generators or through different modes of operation of spinal neural circuits. In mammals and other vertebrates, CPGs can produce distinct rhythmic behaviors involving a common set of muscles, not just locomotion, but also various forms of scratch and fast paw shake (Frigon, 2012). The different rhythms can be generated by separate lumbar neural circuits (Langlet et al., 2005), by different CPGs with largely shared rhythm generating components (Berkowitz & Hao, 2011; Hao & Berkowitz, 2017), and by specialized control mechanisms realized by reconfigurations of the rhythm-generating circuits (Frigon & Gossard, 2010). It should be noted, however, that spinal myoclonus shows little resemblance to the fast and unilaterally expressed rhythmic behaviors of scratch and paw shake.”

Using the term 'pattern formation layer' in relation to biological model may not be entirely correct as no specific layer was identified for most of the species. This term is acceptable in simulation studies where this level is conceptually presented in the model.

We agree with the reviewer that the term “layers” might indeed suggest an anatomical layer-by-layer arrangement of circuits with different functions, which is of course not the case. However, this is the terminology used in current literature. To increase clarity, we have now added:

“Thereby the terminology of layers does not denote bands of anatomically contiguous neurons but rather functional levels of specialized yet deeply intertwined neuronal circuits.”

Authors mentioned that "spinal spasticity (hypertonia, Achilles clonus, sensory-evoked spasms) of all of these participants were investigated with the same protocols as in the present manuscript. Seven of the participants responded with EES-induced rhythmic activities. None of the ten participants demonstrates episodes of spinal myoclonus." It is unclear if these participants were evaluated for this study? If yes, this information should be included in methods. Also, what type of analysis was performed to evaluate these participants? As this is a retrospective study, more information should be provided how this single participant was selected and what were the main limitations in selecting results collected from other subjects.

In the last paragraph of our results, we addressed the question whether the occurrence of EES-induced rhythmic activities would be correlated to the occurrence of spinal myoclonus. If these two types of rhythmic activities were not both to be found in a larger subject cohort, this would strengthen our assumption that they would not result from the activity of the same circuitry. To this end, we went back to a subject cohort of one of our previous publications in whom we had found multiple cases of EES-induced rhythmic activities, but not a single example of spinal myoclonus (Danner et al., 2015). The single participant of the present manuscript was selected because of the occurrence of spinal myoclonus.

“Fig. 8c shows that the majority of the deletions had durations that lined up noticeably well with twice the duration of the respective mean rhythm cycles. Using previously published statistical analysis for the classification of motor deletions in mice3 and cats6, we found that four of the 12 examples were indeed non-resetting.” The ‘majority’ is a relative term, needs to be clarified. Is it clear here what type of stat analysis was used?

This terminology (the majority) was used for better readability of the manuscript and a qualitative description of the illustration. It is immediately clarified in the following sentences by the provided statistics. The statistical analysis used was the same as in the referenced animal studies and was also described in detail in the Methods section, including the formula used.

Final remarks:

Unfortunately, this study was performed on one subject, and the validity of the presented findings is somewhat questionable. All conclusions in this study are solely based on indirect measurement, i.e., analysis of electromyographic patterns. Multiple factors mentioned in the manuscript and response to the reviewers’ critique make this study hardly reproducible, and careful consideration of these results and their interpretation without overstating is critical and need improvement. Considering that no assessment was conducted to exclude the impact of residual connectivity across the lesion, no instrumental assessment was implemented to objectively measure manipulations, and multiple other concerns mentioned above, largely decrease the enthusiasm for the results of this work. However, regardless of these limitations, feedback from other reviewers who could make a better judgment based on their extensive expertise in the field should definitely be taken as a lead.

Unfortunately, the strong and highly unfavorable comments of the reviewer appear as personal generalized opinions. The way the statements are given imply gravity (e.g., “Multiple

evidence from animal experiments suggests that...”, “Most of the subjects with clinically complete SCI...”), yet they are not backed up by a single reference to literature. The critique is especially unsatisfying because no actual improvements are suggested (e.g., the reviewer’s critique that this study is “based on indirect measurement” is not accompanied by any suggestions of which direct measurements would have been more suitable that can be actually applied in humans).

Questioning the validity of our results is a very strong critique, which we cannot accept. None of our results come by surprise, or are questionable observations. Each of the motor phenomena described here have been previously reported in independent studies in humans (spinal myoclonus and EES-induced rhythmic activity) or in animals (motor deletions). The value of this report is the first-in-man observation (motor deletions) and the fact that all these various events were collectively seen in a single subject and under the same conditions – allowing direct comparisons and interpretations for the first time.

References

- Berkowitz, A., & Hao, Z.-Z. (2011). Partly Shared Spinal Cord Networks for Locomotion and Scratching. *Integrative and Comparative Biology*, *51*(6), 890–902.
- Bussel, B., Roby-Brami, A., Azouvi, P., Biraben, A., Yakovleff, A., & Held, J. P. (1988). Myoclonus in a patient with spinal cord transection. Possible involvement of the spinal stepping generator. *Brain : A Journal of Neurology*, *111*, 1235–1245.
- Calancie, B. (2006). Spinal myoclonus after spinal cord injury. *The Journal of Spinal Cord Medicine*, *29*(4), 413–424.
- Danner, S. M., Hofstoetter, U. S., Freundl, B., Binder, H., Mayr, W., Rattay, F., & Minassian, K. (2015). Human spinal locomotor control is based on flexibly organized burst generators. *Brain*, *138*(3), 577–588.
- Dimitrijevic, M. R., Gerasimenko, Y., & Pinter, M. M. (1998). Evidence for a spinal central pattern generator in humans. *Annals of the New York Academy of Sciences*, *860*, 360–376.
- Formento, E., Minassian, K., Wagner, F., Mignardot, J. B., Le Goff-Mignardot, C. G., Rowald, A., Bloch, J., Micera, S., Capogrosso, M., & Courtine, G. (2018). Electrical spinal cord stimulation must preserve proprioception to enable locomotion in humans with spinal cord injury. *Nature Neuroscience*, *21*(12), 1728–1741.
- Frigon, A. (2012). Central pattern generators of the mammalian spinal cord. *The Neuroscientist : A Review Journal Bringing Neurobiology, Neurology and Psychiatry*, *18*(1), 56–69.
- Frigon, A., & Gossard, J.-P. (2010). Evidence for specialized rhythm-generating mechanisms in the adult mammalian spinal cord. *The Journal of Neuroscience : The Official Journal of the Society for Neuroscience*, *30*(20), 7061–7071.
- Hao, Z.-Z., & Berkowitz, A. (2017). Shared Components of Rhythm Generation for Locomotion and Scratching Exist Prior to Motoneurons. *Frontiers in Neural Circuits*, *11*.
- Jilge, B., Minassian, K., Rattay, F., Pinter, M. M., Gerstenbrand, F., Binder, H., & Dimitrijevic, M. R. (2004). Initiating extension of the lower limbs in subjects with complete spinal cord injury by epidural lumbar cord stimulation. *Experimental Brain Research*, *154*(3), 308–326.
- Langlet, C., Leblond, H., & Rossignol, S. (2005). Mid-Lumbar Segments Are Needed for the Expression

of Locomotion in Chronic Spinal Cats. *Journal of Neurophysiology*, 93(5), 2474–2488.

Minassian, K., Hofstoetter, U. S., Dzeladini, F., Guertin, P. A., & Ijspeert, A. (2017). The Human Central Pattern Generator for Locomotion: Does It Exist and Contribute to Walking? *Neuroscientist*, 23(6), 649–663.

Minassian, K., Gilge, B., Rattay, F., Pinter, M. M., Binder, H., Gerstenbrand, F., & Dimitrijevic, M. R. (2004). Stepping-like movements in humans with complete spinal cord injury induced by epidural stimulation of the lumbar cord: electromyographic study of compound muscle action potentials. *Spinal Cord*, 42(7), 401–416.

Nadeau, S., Jacquemin, G., Fournier, C., Lamarre, Y., & Rossignol, S. (2010). Spontaneous motor rhythms of the back and legs in a patient with a complete spinal cord transection. *Neurorehabilitation and Neural Repair*, 24(4), 377–383.

Zhong, G., Shevtsova, N. A., Rybak, I. A., & Harris-Warrick, R. M. (2012). Neuronal activity in the isolated mouse spinal cord during spontaneous deletions in fictive locomotion: insights into locomotor central pattern generator organization. *The Journal of Physiology*, 590(19), 4735–4759.

REVIEWERS' COMMENTS

Reviewer #1 (Remarks to the Author):

I have no further comments. I congratulate the authors on their work, which raises a lot of questions and will stimulate future research in this field.

Best,

Alain

Reviewer #3 (Remarks to the Author):

Nothing further.
Well done.

Reviewer #4 (Remarks to the Author):

I greatly appreciate the authors' feedback and all efforts invested in improving this manuscript. This is well-organized and important for the field work, and a revised version was greatly improved. The main concerns regarding this work are separate from the importance of the findings, as stated earlier. Concerns come from all conclusions and scientific projections based on $n=1$, in the retrospective case study using clinical assessment without adequate control and with only one output measured. Also, some of the previously stated questions and concerns are not covered, and a few issues mentioned below need to be addressed.

Once again, I greatly appreciate the time and effort the Authors invested in addressing all questions and their appreciation of the reviewers' efforts and time in reviewing this manuscript.

For the Authors' convenience, all comments are included in the attached file.

Reviewer #1 (Remarks to the Author):

I have no further comments. I congratulate the authors on their work, which raises a lot of questions and will stimulate future research in this field.

Best,

Alain

We are very grateful to Prof. Frigon for his insightful comments on earlier versions of our manuscript, which helped us to improve and extend the scope of the paper. His approval of our manuscript means a lot to us.

Reviewer #3 (Remarks to the Author):

Nothing further.

Well done.

Thank you very much for your comment and valuable input to improve our manuscript. We are glad to see that all of your questions have been answered to your satisfaction.

Reviewer #4 (Remarks to the Author):

I greatly appreciate the authors' feedback and all efforts invested in improving this manuscript. This is well-organized and important for the field work, and a revised version was greatly improved. The main concerns regarding this work are separate from the importance of the findings, as stated earlier. Concerns come from all conclusions and scientific projections based on $n=1$, in the retrospective case study using clinical assessment without adequate control and with only one output measured. Also, some of the previously stated questions and concerns are not covered, and a few issues mentioned below need to be addressed.

Once again, I greatly appreciate the time and effort the Authors invested in addressing all questions and their appreciation of the reviewers' efforts and time in reviewing this manuscript.

For the Authors' convenience, all comments are included in the attached file.

We appreciate this reviewer's continued thorough review of our manuscript. We are pleased that the reviewer is happy with the revisions that have been made and that he now feels that our study is an important addition to the field of research.

The reviewer's main concern remains the $n=1$ approach of our study. We are now clearly stating in the manuscript why the participant was included in our study and emphasize the rarity of his specific conditions. The participant had a spinal cord injury and hip pathology, presented with a very rare form

of spinal myoclonus (prevalence < 1% in individuals with chronic SCI¹), and had an implanted epidural electrode lead. Also, we are now clearly stating that the examinations considered for this study were carried out over a three-month period, allowing for the collection of multiple data sets and the verification of the reproducibility of the results obtained.

This information has been added to the Data analysis, statistics and reproducibility paragraph of the Methods section, also in response to an editorial request. The respective section of the paragraph now reads as follows:

“The clinical examinations used standardized protocols and were not randomized. The examiners were not blinded during experiments. All recordings were derived from the same individual with SCI. The participant presented with a rare form of spinal myoclonus (prevalence < 1% in individuals with chronic SCI¹), expressed as self-sustained rhythmic activity in the lower limbs, and had an implanted epidural electrode lead. No statistical method was used to predetermine sample size. The examinations were carried out over a three-month period, allowing for the collection of multiple data sets and the verification of the reproducibility of the results obtained.”

Additionally, the reviewer made us aware that a specific syndrome called dyscomplete spinal cord injury may have played a role in the motor phenomena described in the manuscript. We appreciate this suggestion, yet, felt that the underlying mechanisms were too hypothetical to be included. This is the text we had formulated based on the reviewer’s feedback, but decided not to add to the revised manuscript:

“Finally, individuals with clinically complete SCI presenting with spinal myoclonus may have sustained a very specific form of lesion to the descending tracts of the spinal cord, sparing some residual trans-lesional connectivity. Conduction through such descending fibers may remain undetected by standard clinical assessments. In theory, such residual connectivity could provide a background excitation to the sub-lesional spinal circuits, even in the absence of voluntary efforts to initiate a lower-limb movement. Since long-term control of spinal myoclonus was achieved only after hip surgery, as seen in a previous study¹, the leading facilitation of spinal myoclonus must have resulted from a tonic background excitation of sensory afferents associated with the hip pathology.”

1. Calancie, B. Spinal myoclonus after spinal cord injury. *J. Spinal Cord Med.* **29**, 413–24 (2006).